# Efficient and Near-Optimal Algorithm for Contextual Dueling Bandits with Offline Regression Oracles

**Aadirupa Saha**[*]
University of Illinois, Chicago
aadirupa.saha@gmail.com

**Robert E. Schapire**
Microsoft Research
schapire@microsoft.com

## Abstract

The problem of contextual dueling bandits is central to reinforcement learning with human feedback (RLHF), a widely used approach in AI alignment for incorporating human preferences into learning systems. Despite its importance, existing methods are constrained either by strong preference modeling assumptions or by applicability only to finite action spaces. Moreover, prior algorithms typically rely on online optimization oracles, which are computationally infeasible for complex function classes, limiting their practical effectiveness. In this work, we present the first fundamental theoretical study of general contextual dueling bandits over continuous action spaces. Our key contribution is a novel algorithm based on a regularized min-max optimization framework that achieves a regret bound of $\tilde{O}(\sqrt{dT})$—the first such guarantee for this general setting. By leveraging offline oracles instead of online ones, our method further improves computational efficiency. Empirical evaluations validate our theoretical findings, with our approach significantly outperforming existing baselines in terms of regret.

## 1 Introduction

Preference-based feedback is widely used in applications like online retail, prediction markets, tournament ranking, recommender systems, search optimization, robotics, and multiplayer games. Compared to ordinal or absolute rewards, it often provides more reliable insights, such as in book summarization [36], training language models like ChatGPT and LLAMA [9], and designing robotic reward functions via trajectory preferences [12]. A well-studied variant of the multi-armed bandit model [4, 21], the dueling bandit problem, addresses decision-making under pairwise preference feedback with noise and limitations. It has been a key research area for over a decade [43, 3, 29, 7, 26], aiming to identify 'good' arms by actively gathering preference feedback from chosen item pairs.

With the success of reinforcement learning with human feedback (RLHF) [12, 38, 37], a key tool for aligning language models with human values and designing AI behavior, interest in the theoretical foundations of personalized learning with preference feedback has grown, particularly in contextual dueling bandits. The fundamentals of dueling bandits and their generalizations have been widely studied in the bandit and learning theory literature [41, 3, 43, 44]. A major limitation of dueling bandits is the assumption of a fixed preference relation $\mathbf{P}$, whereas real-world preferences vary with context (e.g., user demographics, search queries, seasonal trends). Studies on non-stationary preferences often benchmark against the best single item in hindsight, as in adversarial dueling bandits [29, 18], which is often unrealistic. Some approaches mitigate this with a dynamic regret objective, competing with the best action per round [28, 19], but these require impractical knowledge of non-stationary complexity measures [10]. Moreover, such measures can be large, potentially linear in $O(T)$, and fail to leverage contextual information effectively. A development of the line of contextual dueling bandit research is given in Table 1.

In the study of contextual bandits [1, 32, 16], the primary focus has been on value or reward feedback. However, there is a notable scarcity of research concentrating on preference or relative feedback (i.e. in the dueling bandits framework). Some recent attempts address the problem of linear contextual

---

[*]corresponding author

dueling bandits for linear score-based preferences [25, 8, 34], however, due to the restrictive linearity assumption, or lack of reasonable performance guarantees, are often far from practical deployment.

Seeking inspiration from these system needs, the work of [14] first formulated the problem of *contextual dueling bandits* for any general context space and preference relations. However, their proposed algorithms are either computationally intractable that use the computationally inefficient EXP4 algorithms or yield suboptimal $O(T^{2/3})$ regret bounds. [30] resolves the above limitation of [14] by proposing a computationally efficient and statistically optimal algorithm for contextual dueling bandits, under a natural 'realizability assumption,' that assumes that all the preference relations ($\mathbf{P}_t$s) can be approximated by a (suitably large) function class $\mathcal{M}$. **(1).** One of the caveat in their result was it requires access to an *online square loss regression oracle* for $\mathcal{M}$ whose $T$-step square loss regret, $\mathcal{E}_{\text{on},\mathcal{M}}(T)$, needs to be 'small' ($o(T)$) for reasonable regret guarantee.

| Reference | Feedback Setting (Contextual) | Runtime Efficiency | Regret (Optimality) | Oracle Type/Calls |
|---|---|---|---|---|
| ILTCB [2] | Bandit (reward) | Efficient | $O\left(\sqrt{KT\log\left(T|\Pi|\right)}\right)$ ($\Pi$ : Policy classs) | Offline oracle $\widetilde{O}(\sqrt{KT/\log|\mathcal{F}|})$ calls |
| SquareCB [16] | Bandit (reward) | Inefficient | $O(\sqrt{K\mathcal{E}_{\text{off},\mathcal{M}}T})$ Optimal (up to $K$ or $\log T$ factors) | Online oracle $O(T)$ calls |
| FALCON+ [32] | Bandit (reward) | Efficient | $O\left(\sqrt{K\mathcal{E}_{\text{off},\mathcal{M}}\left(\mathcal{F},T,\frac{\delta}{\log T}\right)T}\right)$ Optimal (up to $K$ or $\log T$ factors) | Offline oracle/ $\boldsymbol{O(\log T)}$ or $\boldsymbol{O(\log\log T)}$ calls |
| SparringEXP4.P [14] | Dueling preference | Inefficient | $O\left(\sqrt{KT\log\left(T|\Pi|\right)}\right)$ Optimal (up to log factors) | N/A |
| MaxInp [25] | Dueling preference | Inefficient | $O(\sqrt{dT\log dT})$ Optimal (up to log factors) | Online $O(T)$ calls |
| MinMaxDB [30] | Dueling feedback | Efficient | $O(\sqrt{K\mathcal{E}_{\text{off},\mathcal{M}}T})$ Optimal (up to $K$ or $\log T$ factors) | Online $O(T)$ calls |
| Double-Monster (this paper) | Dueling feedback | Efficient | $O\left(K\sqrt{\mathcal{E}_{\text{off},\mathcal{M}}\left(\mathcal{F},T,\frac{\delta}{\log T}\right)T}\right)$ Optimal (up to $\sqrt{K}$ and $\log T$ factors) | Offline oracle/ $\boldsymbol{O(\log T)}$ or $\boldsymbol{O(\log\log T)}$ calls |
| Double-Monster-Inf (this paper) | Cont space dueling feedback | Efficient | $O\left(d\sqrt{T}\log T\right)$ Optimal (up to $\log T$ factors) | Offline oracle/ $\boldsymbol{O(\log T)}$ or $\boldsymbol{O(\log\log T)}$ calls |

Table 1: Developments in Contextual Dueling Bandits. $\mathcal{E}_{\text{off},\mathcal{M}}$ : Averaged regression loss of the online square loss regressor over $T$ rounds for the function class $\mathcal{F}$. $\mathcal{E}_{\text{off},\mathcal{M}}\left(\mathcal{F},T,\delta\right)$ : Expected regression loss of the offline square loss regressor upon $T$ samples for the function class $\mathcal{F}$ and confidence parameter $\delta$. $\tilde{O}(\cdot)$ hides logarithmic factors in the $O(\cdot)$ notation.

Offline regression oracles work on static datasets, where all input-output pairs are available upfront. The goal is to find the best-fitting function using the entire dataset at once. The oracle can not fit new data points outside the training distribution, making it suited for batch learning tasks.

Unlike offline regression, an online regression oracle operates in a dynamic setting where data arrives sequentially and the model must adapt in real-time, making it significantly more powerful but also substantially harder to design—especially for general function classes where achieving low regret is often infeasible. For example, no online oracle exists for the class of one-dimensional linear threshold functions due to its infinite Littlestone dimension [23, Example 3], and similarly, the class of all non-decreasing $[-1,+1]$-valued functions over $\mathbb{R}$ is not online learnable despite admitting simple offline oracles [22, Example 1]. Further **(2).** the second major limitation of the work lies in their algorithm only applies to finite action spaces of size $K$. It's not generalizable to continuous decision spaces, which is, however, a much more practical and realistic scenario to work on.

Noting these precise limitations of the prior works, the natural question to ask thus is *whether an optimal regret general (continuous) decision space contextual dueling bandit algorithm can be designed with an offline regression oracle?* Note the task of incorporating offline oracles with online learning algorithms is particularly challenging since due to the interactive nature of online learning algorithms, the data generated by the algorithm could be arbitrary (not necessarily following a specific distribution), which makes it much harder to use an offline oracle. There lies one of the main contributions of this work. Towards this, we first analyze the **(i)** *Best-Response* regret for finite $K$-armed contextual dueling bandit, and further extend the setting to **(ii)** The more practical framework of continuous decision space with potentially infinite arms.

## 1.1 Our contributions

**(1). Warm-Up: Best-Response Regret with Offline Oracles:** Our first contribution is in proposing a new efficient algorithm for contextual dueling bandits with an offline regression oracle. To state the guarantee, let $\mathcal{X}$ be a context space, let $\mathcal{A} := [K]$ be an action space of size $K$, and let $\mathcal{P} := \{\mathbf{P} \in [-1,1]^{K \times K} : P[i,j] = -P[j,i], P[i,i] = 0\}$ denote the set of *preference matrices*, which are skew-symmetric matrices with bounded entries and $0$ along the diagonal. In a stochastic contextual dueling bandit instance, the learner interacts with a distribution $\mathcal{D}$ over $\mathcal{X} \times \mathcal{P}$ via the following protocol: at each round $t$ (1) nature samples $(x_t, \mathbf{P}_t) \sim \mathcal{D}$ and reveals $x_t$ to the learner, (2) learner chooses (potentially randomly) two actions $(a_t, b_t) \in [K]^2$, (3) learner observes $o_t \sim \text{Ber}(\frac{P_t[a_t,b_t]+1}{2})$, where $\text{Ber}(\cdot)$ denotes Bernoulli random variable. Thus $o_t \in \{0,1\}$ indicating the preferred feedback between $a_t$ and $b_t$. At each round $t$, the goal of the learner is to choose a dueling action pair $(a_t, b_t)$ and minimize the performance loss against the dynamic best-action for $x_t$ over $T$ rounds (see **Objective-1**, Section 2).

For this problem, we propose an algorithm in Section 4 with the following regret performance:

**Theorem 1** (Main result: Best-Response Regret (informal)). *Under 'realizable' preference functions and assuming the existence of a 'strongest (set of) best item(s)', with probability at least $(1-\delta)$ for any $\delta \in (0,1)$, the best-response regret $(\text{BR-Reg}_T)$ of our algorithm* `Double-Monster` *(Algorithm 1)*

*after $T$ rounds is bounded by:* $O\left(K\sqrt{\mathcal{E}_{\text{off},\mathcal{M}}\left(\frac{\delta}{\log T}, T\right)T}\right)$, *for any $T \in \mathbb{N}_+$, where $\mathcal{E}_{\text{off},\mathcal{M}}(\delta, n)$*

*denotes the estimation square loss of the regression oracle when trained on $n$ iid instances, with confidence probability at least $1 - \delta$.*

We introduce the problem and the assumptions more formally in Section 2 and Section 3.

**(2) Main Results: Best-Response Regret in Continuous Decision Spaces:** We further extend the results of Theorem 5 to continuous decision space $\mathcal{K} \subset \mathbb{R}^d$, where each action $a \in \mathcal{K}$ is represented by a $d$-dimensional embedding. Assuming continuous action spaces makes the setting suitable for real world problems like recommender systems or language models where the action set is often large, and potentially infinite. In this case, the preference relation of any action pair $(\mathbf{a}, \mathbf{b})$ under $P_t$ is defined as: $P_t(\mathbf{a}, \mathbf{b}) = \sigma\big(s(x_t, \mathbf{a}) - s(x_t, \mathbf{b})\big)$, where $\sigma$ denotes the sigmoid function. The scoring function comes from a realizable function class $\Phi$, a set of functions mapping $\mathcal{X}$ to $\mathbb{R}^d$, such that for any action $\mathbf{a} \in \mathcal{K}$ and context $x \in \mathcal{X}$, we have $\mathbf{E}[s(x, \mathbf{a}) \mid x, \mathbf{a}] = \langle \phi^*(x), \mathbf{a} \rangle$ for some unknown mapping $\phi^* \in \Phi$. A detailed description is given in Section 2. We propose an algorithm for this general problem in Section 6 with near-optimal best-response regret bound (up to $\log$ factors).

**(3) Experimental evaluation:** We also report experiments to validate our theoretical analysis and runtime efficiency of the proposed methods (Section 7) and Appendix E.

## 2 Problem Setup

**Notation.** Let $[n] := \{1, 2, \ldots n\}$, for any $n \in \mathbb{N}$. Also $[n]^2 = [n] \times [n]$ denotes the cartesian product of $[n]$ with itself. We use lowercase bold letters for vectors and uppercase bold letters for matrices. $\mathbf{I}_d$ denotes the $d \times d$ identity matrix. For any vector $\mathbf{x} \in \mathbb{R}^d$, $\|\mathbf{x}\|_2$ denotes the $\ell_2$ norm of $\mathbf{x}$. $\Delta_n := \{\mathbf{p} \in [0,1]^n \mid \sum_{i=1}^n p(i) = 1, p(i) \geq 0, \forall i \in [n]\}$ denotes the $n$-simplex. $\mathbf{e}_i$ denotes the $i$-th standard basis vector in $\mathbb{R}^n$. If $\mathbf{p} \in \Delta_{n \times n}$ is a joint distribution over $[n] \times [n]$, then we denote by $\mathbf{p}^\ell$ and $\mathbf{p}^r$ respectively the left and the right marginal of $\mathbf{p}$, defined as $\mathbf{p}^\ell(i) = \sum_{j=1}^K p(i,j)$ and $\mathbf{p}^r(j) = \sum_{i=1}^K p(i,j)$. In this work, we consider the *zero-sum* representation of preference matrices:
$$\mathcal{P}_n := \{\mathbf{P} \in [-1,1]^{n \times n} \mid P[i,j] = -P[j,i], P[i,i] = 0, \ \forall i,j \in [n]\}.$$
Note any $\mathbf{P} \in \mathcal{P}$ can be viewed as a *zero-sum game*, where the two players, called row and column player, simultaneously choose two (possibly randomized) items from $[n]$, with their goal being to respectively maximize and minimize the value of the selected entry. [2]

**Setting.** We assume any arbitrary context set $\mathcal{X}$, an action space of $K$ items denoted by $\mathcal{A} := [K]$, and a function class $\mathcal{M} \subseteq \{\mathbf{M} \mid \mathbf{M} : \mathcal{X} \mapsto \mathcal{P}_K\}$, all known to the learner ahead of the game. At

---

[2]Standard dueling bandit literature represents preference matrices $\mathbf{Q} \in [0,1]^{n \times n}$, such that $Q[i,j]$ indicates the probability of item $i$ being preferred over item $j$. Here $\mathbf{Q}$ satisfies $Q[i,j] = 1 - Q[j,i]$ and $Q[i,i] = 0.5$. Note both representations are equivalent as there exists a one to one mapping $\mathbf{P} = (2\mathbf{Q} - 1) \in \mathcal{P}$ [14, 7].

each round, we assume a context-preference pair $(x_t, \mathbf{P}_t) \sim \mathcal{D}$ is drawn from a joint-distribution $\mathcal{D}$, such that $x_t \in \mathcal{X}$, and $\mathbf{P}_t \in \mathcal{P}_K$. We will denote the marginal distribution of the context $\mathcal{X}$ as $\mathcal{D}_\mathcal{X}$. The task of the learner is to select an action pair $(a_t, b_t) \in [K] \times [K]$, upon which a relative feedback $o_t \in \{0, 1\}$ is revealed according to $\mathbf{P}_t$; specifically the probability that $a_t$ is preferred over $b_t$, indicated by $o_t = 1$, is given by $\Pr(o_t = 1) = \frac{P_t[a_t, b_t] + 1}{2}$, and hence $\Pr(o_t = 0) = \frac{1 - P_t[a_t, b_t]}{2}$.

**Assumption 1** (Realizability). *Consider a function class $\mathcal{M} \subseteq \{\mathbf{M} \mid \mathbf{M} : \mathcal{X} \to \mathcal{P}_K\}$ consisting of mappings from context $\mathcal{X}$ to preference space $\mathcal{P}_K$. Realizability assumption entails $\exists \mathbf{M}^\star \in \mathcal{M}$ such that $\forall a, b \in [K]$ and any $x \in \mathcal{X}$, we have $\mathbb{E}_{(x, \mathbf{P}) \sim \mathcal{D}}[P[a, b] \mid x] = \mathbf{M}^\star(x)[a, b]$.*

**Objective (1): Best-Response Regret [30]**   Assuming the learner selects the duel $(a_t, b_t) \sim \mathbf{p}_t \in \Delta_{K \times K}$ at each round $t$, we measure the learner's performance via a notion of best response regret:
$$\text{BR-Reg}_T := \frac{1}{2} \sum_{t=1}^T \mathbb{E}_{x_t \sim \mathcal{D}_\mathcal{X}} \left[ \max_{\mathbf{q} \in \Delta_K} \left[ \mathbb{E}_{a \sim \mathbf{q}} \left[ \mathbb{E}_{(a_t, b_t) \sim \mathbf{p}_t} [\mathbf{M}^\star(x_t)[a, a_t] + \mathbf{M}^\star(x_t)[a, b_t]] \right] \right] \right],$$
where $\mathcal{D}_\mathcal{X}$ denotes the marginal distribution over $\mathcal{X}$.

**Objective (2): Best-Response Regret for Continuous Arm Spaces**   In this setting, we relax the assumed of finite $K$-armed action space and assume a general continuous (potentially infinite) decision space $\mathcal{K} \subset \mathbb{R}^d$. In order to model the preference relation for an action pair $(\mathbf{a}, \mathbf{b}) \in \mathcal{K} \times \mathcal{K}$: We further assume that given any context $x_t$ each action $\mathbf{a}_t \in \mathcal{K}$ first gets assigned to a stochastic score $s_t \in [-1, 1]$ such that $\mathbf{E}[s_t \mid x_t = x, \mathbf{a}_t = \mathbf{a}] = \langle \mathbf{a}, \phi^*(x) \rangle$, where $\phi^* : \mathcal{X} \mapsto \mathbb{R}^d$ is an unknown mapping that embeds every context to a $d$-dimensional feature space. In particular, with the *realizability assumption*, one can typically assume the learner has access to a class of functions $\Phi \subseteq \{\phi : \mathcal{X} \to \mathbb{R}^d\}$ such that $\phi^* \in \Phi$. Then at time $t$, the preference relation of any action pair $(\mathbf{a}, \mathbf{b})$ under the above realizable score setting is: $P_t(\mathbf{a}, \mathbf{b}) = \sigma(\langle \phi^*(x), \mathbf{a} - \mathbf{b} \rangle)$, where $\sigma(\cdot)$ denotes the sigmoid transformation, i.e. $(\sigma(x) = \frac{1}{1 + e^{-x}}, \forall x \in \mathbb{R})$. Similar to before, the learner's task is to select an action pair $(\mathbf{a}_t, \mathbf{b}_t) \in \mathcal{K} \times \mathcal{K}$ at time $t$, upon which a relative feedback $o_t \in \{0, 1\} \sim \text{Ber}(P_t(\mathbf{a}_t, \mathbf{b}_t))$ is revealed according to $\mathbf{P}_t$.

Under this model of continuous action space $\mathcal{K}$, one could define best-response regret as:
$$\text{BR-Reg}_T^{(cont)} := \sum_{t=1}^T \mathbb{E}_{x_t \sim \mathcal{D}_\mathcal{X}} \left[ \max_{\mathbf{a}^* \in \mathcal{K}} \langle \mathbf{a}^*, \phi^*(x_t) \rangle - \frac{\langle (\mathbf{a}_t + \mathbf{b}_t), \phi^*(x_t) \rangle}{2} \right]$$

# 3   A Primer on Regression Oracles

We now introduce the concepts of offline and online regression oracles to familiarize readers with their structural disparities and performance-related distinctions.

## 3.1   Offline Regression Oracles [32]

Consider any abstract input space $\mathcal{Z}$ and output space $\mathcal{Y}$. Assume a given dataset $D_n := \{(z_i, y_i)\}_{i=1}^n$ consists of $n$ data points, each (input, output) pair $(z_i, y_i) \overset{iid}{\sim} \mathcal{D}$ being drawn iid from a fixed underlying distribution $\mathcal{D}$ on $\mathcal{Z} \times \mathcal{Y}$. Given a general function class $\mathcal{F} \subseteq \{f \mid f : \mathcal{Z} \mapsto \mathcal{Y}\}$, a general offline regression oracle associated with $\mathcal{F}$, denoted by $\texttt{OffReg}_\mathcal{F}$ is an algorithm, operates on the given dataset $D_n$ and outputs a mapping $\hat{f} : \mathcal{Z} \mapsto \mathcal{Y}$. In learning theory, the quality of $\hat{f}$ is measured by its "out-of-sample error," i.e., its expected error on *random* and *unseen* test data.

**Assumption 2** (Guarantee of Offline Regression Oracles). *Consider a general function class $\mathcal{F} \subseteq \{f \mid f : \mathcal{Z} \mapsto \mathcal{Y}\}$. Then given a dataset $D_n := \{(z_i, y_i)\}_{i=1}^n$ that consists of $n$ data points, s.t. $(z_i, y_i) \overset{iid}{\sim} \mathcal{D}$, we assume that the output $\hat{f}_n \leftarrow \texttt{OffReg}_\mathcal{F}(D_n)$ of the offline regression oracle $\texttt{OffReg}$ satisfies that: For any $\delta \in (0, 1]$, with probability at least $1 - \delta$,*
$$\mathbb{E}_{(z, y) \sim \mathcal{D}} \left[ (\hat{f}_n(z) - y)^2 - \inf_{f \in \mathcal{F}} (f(z) - y)^2 \right] \le \mathcal{E}_{\text{off}, \mathcal{F}}(\delta, n),$$
*where $\mathcal{E}_{\text{off}, \mathcal{M}}(\delta, n)$ denotes the square loss of the regression oracle when trained on $n$-iid instances.*

**Assumption 4 under Realizability.** Moreover, if we further assume realizability, in the sense that there exists $f^* \in \mathcal{F}$ such that $f^*(z) = \mathbb{E}_{(z, y) \sim \mathcal{D}}[y \mid z]$, then it is well-known that Assumption 4 further implies: For any $\delta \in (0, 1]$, with probability at least $1 - \delta$, we have
$$\mathbb{E}_{(z, y) \sim \mathcal{D}} \left[ (\hat{f}_n(z) - f^\star(z))^2 \right] \le \mathcal{E}_{\text{off}, \mathcal{F}}(\delta, n). \tag{1}$$

**Remark 2.** *The advantage of an offline regression oracle is that it can consider all the data available for training, the training data is generated iid from some underlying distribution and therefore, can often produce a more accurate model. See Remark 13 for the advantages of offline regressors. Conversely, online regression is a real-time processing approach that involves analyzing data as it becomes available, updating the model as new data points arrive. Thus finding efficient online oracles for any arbitrary function class could be much harder compared to its offline counterpart (e.g. see [23, Example 3], [22, Example 1]) See Appendix A for a detailed discussion.*

## 4   Warm-Up: $K$-Armed General Contextual DB with Offline Oracles

This section presents our algorithm for the contextual best-response regret (**Objective-1** in Section 2).

**Double-Monster: Key Algorithmic Ideas.**   To adapt with the offline regression oracle, our algorithm runs in epochs, with a specific (and predetermined) epoch schedule, where the offline regression oracle is called only at rounds $\tau_1, \tau_2, \tau_3, \ldots$, and each set of rounds between $\tau_{m-1} + 1$ and $\tau_m$ is considered to be within the $m$-th epoch. For example, if we set $\tau_m = 2^m$, our algorithm runs in $O(\log T)$ epochs for any unknown $T$, but one can also use more complex epoch schedules, such as $\tau_m = \left\lfloor 2T^{1-2^{-m}} \right\rfloor$, which results in only $O(\log \log T)$ epochs and oracle calls.

At the beginning of each epoch $m$, the algorithm computes an estimated function $\widehat{\mathbf{M}}_m$ using the offline regression oracle: $\widehat{\mathbf{M}}_m := \arg\min_{\mathbf{M} \in \mathcal{M}} \sum_{\tau=\tau_{m-2}+1}^{\tau_{m-1}} (\mathbf{M}(x_\tau)[a_\tau, b_\tau] - o_\tau)^2$. We next assume access to an algorithm Convex-Constraint-Solver, which is a randomized algorithm that for given $x \in \mathcal{X}$, $\mathbf{M} \in \mathcal{M}$ and $\gamma \in \mathbb{R}$, outputs a joint distribution $\mathbf{p} \in \Delta_{K \times K}$, such that:

$$\mathbf{E}_{a \sim \mathbf{p}^\ell} \left[ \mathbf{M}(x)[i,a] \right] + \mathbf{E}_{b \sim \mathbf{p}^r} \left[ \mathbf{M}(x)[j,b] \right] + \frac{2}{\gamma} \frac{1}{p(i,j)} \leq \frac{5K^2}{\gamma}, \ \forall i, j \in [K] \tag{2}$$

where recall we denote by $\mathbf{p}^\ell$ and $\mathbf{p}^r$ respectively the left and the right marginal of $\mathbf{p}$, defined as $\mathbf{p}^\ell(i) = \sum_{j=1}^K p(i,j)$ and $\mathbf{p}^r(j) = \sum_{i=1}^K p(i,j)$. Note $\mathbf{p}$ being the solution of $\binom{K}{2} + K$-convex constraints, such a solver can be designed computationally efficiently using standard tools from convex programming. Moreover, in Appendix B.2 justify that a $\mathbf{p} \in \Delta_{K \times K}$ will always exist. Now, at any time $t$ in epoch $m$, given the context $x_t$ the estimated least square function estimate $\widehat{\mathbf{M}}_m$, and some tuning parameter $\gamma_m$ (exact expression is given later), we query the Convex-Constraint-Solver with the triplet $(x_t, \widehat{\mathbf{M}}_m, \gamma_m)$ and obtain $\mathbf{p}_t \in \Delta_{K \times K}$ using:

$$\mathbf{p}_t \leftarrow \text{Convex-Constraint-Solver}(x_t, \widehat{\mathbf{M}}_m, \gamma_m) \tag{3}$$

Next, the algorithm samples a duel $(a_t, b_t) \sim \mathbf{p}_t$. Noting $\mathbf{E}_{a \sim \mathbf{q}, b \sim \mathbf{p}_t} \left[ \widehat{\mathbf{M}}_m(x_t)[a,b] \right] = \mathbf{q}^\top \widehat{\mathbf{M}}_m(x_t) \mathbf{p}_t$, $\mathbf{p}_t$ can be alternatively expressed as:

$$\mathbf{q}^\top \widehat{\mathbf{M}}_m(x_t) \mathbf{p}_t^\ell + \mathbf{q}'^\top \widehat{\mathbf{M}}_m(x_t) \mathbf{p}_t^r + \frac{2}{\gamma_m} \mathbf{E}_{i \sim \mathbf{q}, j \sim \mathbf{q}'} \left[ \frac{1}{p_t(i,j)} \right] \leq \frac{5K^2}{\gamma_m}, \ \forall \mathbf{q} \text{ and } \mathbf{q}' \in \Delta_K \tag{4}$$

which implies the mixed strategy $\mathbf{p}_t$ is a 'nearly unbeatable by any pair of opponent $\mathbf{q}$ and $\mathbf{q}''$ in the zero-sum game $\widehat{\mathbf{M}}_m(x_t)$. Recall, any distribution $\mathbf{p}^* \in \Delta_K$ is a Nash Equilibrium of a symmetric zero-sum square matrix $\mathbf{P} \in \mathbb{R}^{K \times K}$ if $\forall \mathbf{q} \in \Delta_K : \mathbf{q}^\top \mathbf{P} \mathbf{p}^* \leq 0$ [11] and Nash strategies are unbeatable. However, to incorporate the additional 'explorative component' of $\mathbf{p}_t$ we need to ensure $p_t(i,j)$ has significant mass based on the quality of action $i$ and $j$. This is enforced through the term $\frac{1}{p_t(i,j)}$ in Eq. (4). In Appendix B.2, we prove that Eq. (2) always adheres to a valid solution and hence one can always find a $\mathbf{p}_t$ for Eq. (4). *The pseudocode is given in Algorithm 1 in Appendix B.*

### 4.1   Performance Guarantees of Algorithm 1.

**Useful Concepts towards proving Theorem 5.** We analyze the regret of Algorithm 1 in Theorem 5, but we need to define some concepts before that.

**Definition 1** (Policy Class). *A standard policy class $\Pi := \{\pi \mid \pi : \mathcal{X} \mapsto \Delta_K\}$ is a set containing all the mappings from context space $\mathcal{X}$ to the $K$-simplex. Further we also define by policy class $\Pi^2 := \{\pi \mid \pi : \mathcal{X} \mapsto \Delta_{K \times K}\}$ a set of all mappings from context $\mathcal{X}$ to $K \times K$-simplex.*

For any policy $\boldsymbol{\pi} \in \Pi$ we denote by $\pi(i \mid x)$ the $i$-th component of vector $\boldsymbol{\pi}(x)$, $\forall i \in [K], x \in \mathcal{X}$. Similarly, for any policy $\boldsymbol{\pi} \in \Pi^2$, we denote by $\pi(i, j \mid x)$, the $i, j$-th component of the matrix $\boldsymbol{\pi}(x)$.

**Definition 2** (Decision policy of Epoch-$m$). *At any epoch $m$ and $x \in \mathcal{X}$, the decision policy of epoch $m$ is defined as $\boldsymbol{\pi}_m \in \Pi^2$: $\boldsymbol{\pi}_m(x) \leftarrow$ Convex-Constraint-Solver$(x, \widehat{\mathbf{M}}_m, \gamma_m)$.*

Note that at any time $t$ in epoch $m$, $\boldsymbol{\pi}_m(x_t) = \mathbf{p}_t$, where $\mathbf{p}_t$ is as defined in Eq. (3). Thus $\boldsymbol{\pi}_m \in \Pi^2$. We will further denote by $\boldsymbol{\pi}_m^{\ell}, \boldsymbol{\pi}_m^r \in \Pi$ the left and right marginal policies of $\boldsymbol{\pi}_m$ defined as: $\boldsymbol{\pi}_m^{\ell}(i \mid x) = \mathbf{E}_{x \sim \mathcal{D}_{\mathcal{X}}}\left[\sum_{j=1}^{K} \boldsymbol{\pi}_m(i, j \mid x)\right]$ and $\boldsymbol{\pi}_m^r(j \mid x) = \mathbf{E}_{x \sim \mathcal{D}_{\mathcal{X}}}\left[\sum_{i=1}^{K} \boldsymbol{\pi}_m(i, j \mid x)\right]$, $\forall i, j \in [K]$

**Definition 3** (Instantaneous Regret against a Fixed Policy). *For any arbitrary preference relation $\mathbf{M} \in \mathcal{M}$, we denote the regret of policy $\boldsymbol{\pi}$ against $\boldsymbol{\pi}'$ as:*

$$\mathrm{Reg}_{\mathbf{M}}(\boldsymbol{\pi}, \boldsymbol{\pi}') := \mathbf{E}_{x \sim \mathcal{D}_{\mathcal{X}}}[\boldsymbol{\pi}(x)^{\top} \mathbf{M}(x) \boldsymbol{\pi}'(x)].$$

In particular, for $\mathbf{M} = \mathbf{M}^*$, $\mathrm{Reg}(\boldsymbol{\pi}, \boldsymbol{\pi}') = \mathrm{Reg}_{\mathbf{M}^*}(\boldsymbol{\pi}, \boldsymbol{\pi}')$ defines true regret of policy $\boldsymbol{\pi}$ against $\boldsymbol{\pi}'$ and for $\mathbf{M} = \widehat{\mathbf{M}}_m$, $\mathrm{Reg}_{\widehat{\mathbf{M}}_m}(\boldsymbol{\pi}, \boldsymbol{\pi}')$ simply denotes the instantaneous empirical regret of $\boldsymbol{\pi}$ against $\boldsymbol{\pi}'$ in epoch $m$. For simplicity, we denote $\widehat{\mathrm{Reg}}_m(\boldsymbol{\pi}, \boldsymbol{\pi}') = \mathrm{Reg}_{\widehat{\mathbf{M}}_m}(\boldsymbol{\pi}, \boldsymbol{\pi}')$.

Further since for any $\mathbf{P} \in \mathcal{P}_K$, i.e. $P[i, j] = -P[j, i]$, $\forall i, j \in [K] \times [K]$, we note that:

**Remark 3** (Properties of Policy Regret). *For any pair of policies $\boldsymbol{\pi}$ and $\boldsymbol{\pi}' \in \Pi$: (1) $\mathrm{Reg}(\boldsymbol{\pi}, \boldsymbol{\pi}') = -\mathrm{Reg}(\boldsymbol{\pi}', \boldsymbol{\pi})$, (2) $\widehat{\mathrm{Reg}}_m(\boldsymbol{\pi}, \boldsymbol{\pi}') = -\widehat{\mathrm{Reg}}_m(\boldsymbol{\pi}', \boldsymbol{\pi})$.*

**Definition 4** (Best-Response Policy). *We let $\boldsymbol{\psi}^{\star} : \Pi \times \mathcal{M} \to \Pi$ be a best-response policy, meaning, for any context $x \in \mathcal{X}$, policy $\boldsymbol{\pi} \in \Pi$ and function $\mathbf{M} \in \mathcal{M}$, $\boldsymbol{\psi}^{\star}[\boldsymbol{\pi}, \mathbf{M}](x)$ maximizes $\mathbf{E}_{a^{\star} \sim \mathbf{p}, a \sim \boldsymbol{\pi}(x)}\big[\mathbf{M}(x)[a^{\star}, a]\big] = \mathbf{p}^{\top} \mathbf{M}(x) \boldsymbol{\pi}(x)$ over $\mathbf{p} \in \Delta_K$. More precisely for any $x \in \mathcal{X}$, $\boldsymbol{\pi} \in \Pi$ and $\mathbf{M} \in \mathcal{M}$, $\boldsymbol{\psi}^{\star}[\boldsymbol{\pi}, \mathbf{M}](x) \in \Pi$ can be defined as:*

$$\boldsymbol{\psi}^{\star}[\boldsymbol{\pi}, \mathbf{M}](x) = \arg \max_{\mathbf{p} \in \Delta_K} \mathbf{p}^{\top} \mathbf{M}(x) \boldsymbol{\pi}(x)$$

Note that the due to linearity of the above objective, the corresponding argmax always occurs at one of the $K$ extreme points of $K$-simplex $\Delta_K$.

**Definition 5** (Instantaneous Best Response Policy Regret). *For any arbitrary underlying preference relation $\mathbf{M} \in \mathcal{M}$, we denote the instantaneous best-response regret of any decision policy $\boldsymbol{\pi} \in \Pi$ as:*

$$\mathrm{BReg}_{\mathbf{M}}(\boldsymbol{\pi}) := \mathbf{E}_{x \sim \mathcal{D}_{\mathcal{X}}}\left[\max_{\mathbf{p}^{\star} \in \Delta_K} \mathbf{p}^{\star \top} \mathbf{M}(x) \boldsymbol{\pi}(x)\right].$$

In particular, for $\mathbf{M} = \mathbf{M}^*$, $\mathrm{BReg}_{\mathbf{M}^*}(\boldsymbol{\pi})$ denotes the instantaneous true best-response regret of any decision policy $\boldsymbol{\pi} \in \Pi$. For simplicity we will denote $\mathrm{BReg}(\boldsymbol{\pi}) = \mathrm{BReg}_{\mathbf{M}^*}(\boldsymbol{\pi})$. Further for $\mathbf{M} = \widehat{\mathbf{M}}_m$, $\mathrm{BReg}_{\widehat{\mathbf{M}}_m}(\boldsymbol{\pi})$ simply defines the instantaneous empirical best-response regret of $\boldsymbol{\pi}$ in epoch $m$. For simplicity, we will use $\widehat{\mathrm{BReg}}_m(\boldsymbol{\pi}) = \mathrm{BReg}_{\widehat{\mathbf{M}}_m}(\boldsymbol{\pi})$.

**Remark 4.** *Note that for any $\boldsymbol{\pi} \in \Pi$, one can write*

$$\mathrm{BReg}(\boldsymbol{\pi}) = \mathrm{Reg}(\boldsymbol{\psi}^{\star}[\boldsymbol{\pi}, \mathbf{M}^{\star}], \boldsymbol{\pi}), \text{ and } \widehat{\mathrm{BReg}}_m(\boldsymbol{\pi}) = \widehat{\mathrm{Reg}}(\boldsymbol{\psi}^{\star}[\boldsymbol{\pi}, \widehat{\mathbf{M}}_m], \boldsymbol{\pi}),$$

*This implies that the instantaneous true and empirical regret of any fixed policy $\boldsymbol{\pi}$ against its corresponding best response policy, respectively yield the instantaneous true and empirical best response regret of $\boldsymbol{\pi}$.*

**Assumption 3** (Idempotent Best-Response (IBR)). *A function class $\mathcal{M} : \mathcal{X} \mapsto \mathcal{P}$ is said to satisfy Idempotent Best-Response if for all $\mathbf{M} \in \mathcal{M}$ and all $\boldsymbol{\pi} \in \Pi$, $\mathrm{BReg}_{\mathbf{M}}(\boldsymbol{\psi}^*[\boldsymbol{\pi}, \mathbf{M}]) = 0$.*

Roughly speaking, any preference matrix with 'total-ordering', where there is a specific underlying ranking of the $K$ items, satisfies the above structure: The class of random utility (RUM) based preferences [5, 33, 27], or matrices with strong stochastic transitivity (SST) [42], always satisfies the above property. However, the class 'Idempotent Best-Response' is larger, in particular, any preference matrix, which has a set of equally strong best items also satisfies the structure. The figure on the right shows the dependencies of different preference classes. We now state our main result towards analyzing Algorithm 1.

**Theorem 5** (Regret Analysis of Algorithm 1). *Consider any function class $\mathcal{M}$ that satisfies Assumption 3. Then under realizability (Assumption 1), and with an epoch schedule $\tau_1, \ldots, \tau_m$ such that $\tau_m \geq 2^m$ for $m \leq \log T$, with probability at least $(1 - \delta)$ for any $\delta \in (0, 1)$ the best-response regret* (BR-$\mathrm{Reg}_T$) *of* **Double-Monster** *after $T$ rounds is bounded by:*

$$O\left(K \sum_{m=2}^{m(T)} \sqrt{\mathcal{E}_{\mathrm{off},\mathcal{M}}(\delta/(2m^2), \tau_{m-1} - \tau_{m-2})(\tau_m - \tau_{m-1})}\right) \tag{5}$$

**Corollary 6** (Theorem 5 for Special Function Classes). *Algorithm 1 yields the following best-response regret guarantees for some of the special function classes:*

- *For finite $\mathcal{M}$, for the choice of $\delta = 1/T$, the regret of Algorithm 1 is bounded by $O(K\sqrt{T \log(|\mathcal{M}|T \log T)})$. The results follows since for finite $\mathcal{M}$, we have offline regression oracles with $\mathcal{E}_{\mathrm{off},\mathcal{M}}(\delta, n) = O(\frac{\log(|\mathcal{M}|n)}{n\delta})$ [32]. For completeness, the proof is given in Appendix C.5.*

- *For a general, potentially nonparametric function class $\mathcal{F}$ with empirical entropy is $O(\varepsilon^{-p})$, $\forall \varepsilon > 0$ for some constant $p > 0$, results of [40] and [24] gives offline regression oracles such that $\mathcal{E}_{\mathrm{off},\mathcal{M}}(\delta, n) = O(n^{-2/(2+p)} \log(1/\delta))$. Again assuming $\delta = 1/T$, this implies a regret bound of $O(KT^{\frac{1+p}{2+p}} \log T)$ of Algorithm 1 for this function class.*

- *For low dimensional linear predictors $\mathcal{F} = \{(x, a, b) \mapsto \langle \theta, \phi(x, a, b) \rangle : \theta \in \mathbb{R}^d, \|\theta\|_2 \leq 1\}$, instantiating* OffReg *as the least squared estimator [39, 13, 21] and $\delta = 1/T$, the regret guarantee of Algorithm 1 becomes $O(d\sqrt{T \log(T/d)})$.*

- *[15] gives regression error bounds for deep neural networks for $\mathcal{F} = \mathcal{G}^K$, $\mathcal{G}$ being the class of Multi-Layer Perceptrons (MLP), and $f^*(x, a, b) = g^*_{a,b}(x)$ for $x \in \mathcal{X}, a, b \in [K]$. Assume that $\mathcal{D}$ is a continuous distribution on $[-1, 1]^d$ and $g^*_{a,b}$ lie in a Sobolev ball with smoothness $\beta \in \mathbb{N}$, by Theorem 1 of [15] deep MLP-ReLU network estimator attains $\mathcal{E}_{\mathrm{off},\mathcal{M}}(n, \delta) = \widetilde{O}(n^{-\frac{\beta}{\beta+d}} \log 1/\delta)$ estimation error. Consequently, the regret bound of Algorithm 1 boils down to $\tilde{O}(KT^{\frac{\beta+2d}{2\beta+2d}})$ regret.*

## 5 Regret Analysis of **Double-Monster**: Proof Analysis of Theorem 5

Towards proving Theorem 5, we will first prove Lemma 7 that guarantees on the empirical regret performance of Algorithm 1. But it will be worth introducing a few more notations first:

**Definition 6** (Decision variance). *For any two policies $\pi_1, \pi_2 \in \Pi$ the decision variance of $\pi_1, \pi_2$ with respect to another decision policy $\tilde{\pi} \in \Pi^2$ is defined as: $\mathcal{V}_{\tilde{\pi}}(\pi_1, \pi_2) := \mathbf{E}_{x \sim \mathcal{D}_{\mathcal{X}}} \left[\sum_{a=1}^{K} \frac{\pi_1(a|x)\pi_2(a|x)}{\tilde{\pi}(a,b|x)}\right]$.*

Intuitively $\mathcal{V}_{\tilde{\pi}}(\pi_1, \pi_2)$ captures the selection-variance of $\tilde{\pi}$ for any expected duel $(a, b)$ played by the joined policy $\pi_1, \pi_2$. In particular, when the base policy $\tilde{\pi} = \pi_m$, for any epoch $m$, we will use the shorthand notation $\mathcal{V}_m(\pi_1, \pi_2)$ to represent $\mathcal{V}_{\pi_m}(\pi_1, \pi_2)$.

**Lemma 7** (Properties of decision policy $\pi_m$ in Algorithm 1). *At any epoch $m$, $\forall \pi, \pi' \in \Pi$, the decision policy of epoch $m$ $\pi_m$ satisfies:*

$$\widehat{\mathrm{BReg}}_m(\pi_m^\ell) + \widehat{\mathrm{BReg}}_m(\pi_m^r) \leq \frac{5K^2}{\gamma_m},$$

$$2\mathcal{V}_m(\pi, \pi') \leq 5K^2 + \gamma_m \widehat{\mathrm{Reg}}_m(\pi_m^\ell, \pi) + \gamma_m \widehat{\mathrm{Reg}}_m(\pi_m^r, \pi').$$

Lemma 7 establishes an important result for bounding the empirical performance of the decision policy $\pi_m$, for any epoch $m$: Precisely it shows that the empirical best response regret of $\pi_m$ is bounded by $O(K^2/\gamma_m)$ and the decision variance of any pair of policies $\pi, \pi'$ w.r.t. $\pi_m$ is bounded by empirical regret bound $\pi$ and $\pi'$ against $\pi_m$. The proof is given in Appendix B.

### 5.1 Relating Empirical Performance to True Performance through Decision Variance

We will first show how to relate the true and empirical performance of any policy in terms of their corresponding decision variance, as given below:

**Lemma 8.** *For any epoch $m$ and any two decision policies $\boldsymbol{\pi}$ and $\boldsymbol{\pi}'$,*

$$\text{Reg}(\boldsymbol{\pi}', \boldsymbol{\pi}) - \widehat{\text{Reg}}_m(\boldsymbol{\pi}', \boldsymbol{\pi}) \leq \frac{\mathcal{V}_{m-1}(\boldsymbol{\pi}', \boldsymbol{\pi})}{3\gamma_m} + \frac{3K^2}{4\gamma_m}.$$

Proof of this lemma is deferred to Appendix C.2. We note that Lemma 8 further implies:

**Corollary 9.** *For any epoch $m$, and any policy $\boldsymbol{\pi} \in \Pi$*

$$\text{BReg}(\boldsymbol{\pi}_m^\ell) + \text{BReg}(\boldsymbol{\pi}_m^r) - [\widehat{\text{BReg}}_m(\boldsymbol{\pi}_m^\ell) + \widehat{\text{BReg}}_m(\boldsymbol{\pi}_m^r)]$$

$$\leq \frac{\mathcal{V}_{m-1}(\boldsymbol{\psi}^\star[\boldsymbol{\pi}_m^\ell, \mathbf{M}^\star], \boldsymbol{\pi}_m^\ell) + \mathcal{V}_{m-1}(\boldsymbol{\psi}^\star[\boldsymbol{\pi}_m^\ell, \mathbf{M}^\star], \boldsymbol{\pi}_m^r)}{3\gamma_m} + \frac{3K^2}{2\gamma_m}.$$

### 5.2 The Recursion: Bounding Empirical Regret by True Regret and Vice-Versa

Using Corollary 9, we now prove a crucial recursive relation between empirical best-response and true best-response regret for any decision policy $\boldsymbol{\pi}$ at any epoch $m$. This lemma ensures that the estimated empirical (best-response) regret of the algorithm's decision policy $\boldsymbol{\pi}_m$ becomes more and more accurate and eventually matches its true best-response regret for large $m$.

**Lemma 10** (Epochwise Recursion). *Let $\boldsymbol{\pi}$ be any policy in $\Pi^2$. Then for all epochs $m \in \mathbb{N}_+$:*

$$\text{BReg}(\boldsymbol{\pi}^\ell) + \text{BReg}(\boldsymbol{\pi}^r) \leq 2\widehat{\text{BReg}}_m(\boldsymbol{\pi}^\ell) + 2\widehat{\text{BReg}}_m(\boldsymbol{\pi}^r) + 11K^2/\gamma_m,$$

$$\widehat{\text{BReg}}_m(\boldsymbol{\pi}^\ell) + \widehat{\text{BReg}}_m(\boldsymbol{\pi}^r) \leq 2\text{BReg}(\boldsymbol{\pi}^\ell) + 2\text{BReg}(\boldsymbol{\pi}^r) + 11K^2/\gamma_m.$$

Given Lemma 10, Theorem 5 follows by combining it with Lemma 7. Complete proof is given in Appendix C.4.

## 6 Main Algorithm: General Contextual DB for Continuous Action Spaces

In this section, we extend the results of the previous section to continuous action spaces $\mathcal{K} \subset \mathbb{R}^d$, as defined in **Objective-(2)**, Section 3). We present our algorithm in Algorithm 2, which is shown to yield $\tilde{O}(\sqrt{dT})$ contextual best-response regret as analyzed in Theorem 11. We explain the key ideas behind Algorithm 2 below in detail but it would be useful to introduce a few notations before that.

**Additional Notations:** For a set $X$, we let $\Delta(X)$ denote the set of all probability distributions over $X$. If $X$ is continuous, we typically denote $\Delta(X)$ as the set of all probability measures on the measurable space $(X, \mathcal{B})$, where $\mathcal{B}$ is the Borel $\sigma$-field on the set $X$. Further, for any $n \in \mathbb{N}_+$, we denote the $n$-simplex by $\Delta_n$. So, if $X$ is finite, $\Delta(X) = \Delta_{|X|}$. $\mathbf{I}_n$ denotes the identity matrix of dimension $n$ for any $n \in \mathbb{N}_+$. We use $\|\mathbf{x}\|$ (or $\|\mathbf{x}\|_2$) to denote the euclidean norm for $\mathbf{x} \in \mathbb{R}^d$. For any positive definite matrix $\mathbf{H} \in \mathbb{R}^{d \times d}$, we denote the induced norm on $x \in \mathbb{R}^d$ by $\|\mathbf{x}\|_{\mathbf{H}}^2 = \langle \mathbf{x}, \mathbf{H}\mathbf{x} \rangle$, and $\det(\mathbf{H})$ represents the determinant of matrix $\mathbf{H}$.

---

**Algorithm 2 Double-Monster-Inf** (for Continuous Action Spaces)

---

1: **input** Epoch schedule $0 = \tau_0 < \tau_1 < \tau_2 < \cdots$. Confidence parameter $\delta$. Tuning parameter $c, \gamma_1, \gamma_2, \ldots$
2: Arm set: $\mathcal{K} \subset \mathbb{R}^d$. An instance of $\texttt{OffReg}$ for function class $\Phi$
3: **for** epoch $m = 1, 2, \ldots$ **do**
4:     Set $\gamma_m = \left( \dfrac{\sqrt{d}}{2\sqrt{2\mathcal{E}_{\text{off}, \Phi^*}(\delta/(2m^2), \tau_{m-1} - \tau_{m-2})}} \right)$ (for epoch 1, $\gamma_1 = 1$).
5:     Compute $\phi_m \leftarrow \texttt{OffReg}\left( \{(x_\tau, \mathbf{a}_\tau, \mathbf{b}_\tau), o_\tau\}_{\tau = \tau_{m-2}+1}^{\tau_{m-1}} \right)$, i.e.

   $\phi_m = \arg\min_{\phi \in \Phi} \sum_{\tau = \tau_{m-2}+1}^{\tau_{m-1}} (\sigma(\langle \phi(x_\tau), \mathbf{a}_\tau - \mathbf{b}_\tau \rangle) - o_\tau)^2$ via the **offline least squares oracle**
6:     **for** round $t = \tau_{m-1} + 1, \cdots, \tau_m$ **do**
7:         Observe context $x_t \in \mathcal{X}$.
8:         Compute $\mathbf{p}_t \sim \texttt{Cont-CvxConstraint-Solver}(x_t, \phi_m, \gamma_m)$ i.e.:

   $$\mathbf{p}_t \leftarrow \arg\max_{q \in \Delta_{\mathcal{K} \times \mathcal{K}}} \left[ \langle \mathbf{ab_q}, \phi_m(x_t) \rangle + \tfrac{1}{\gamma_m} \log \det(\mathbf{H_q}) \right],$$

   where $\mathbf{ab_q} = \mathbb{E}_{(\mathbf{a}, \mathbf{b}) \sim \mathbf{q}}[(\mathbf{a} + \mathbf{b})/2]$, $\mathbf{H_q} = \mathbb{E}_{(\mathbf{a}, \mathbf{b}) \sim \mathbf{q}}[(\mathbf{a} - \mathbf{b})(\mathbf{a} - \mathbf{b})^\top] + c\mathbf{I}_d$.
9:         Sample $(\mathbf{a}_t, \mathbf{b}_t) \sim \mathbf{p}_t$ and observe preference feedback $o_t \sim \text{Ber}\left( \sigma(\langle \phi^*(x_t), \mathbf{a}_t - \mathbf{b}_t \rangle) \right)$.
10:     **end for**
11: **end for**

---

**Double-Monster-Inf: Key Algorithm Ideas.** The main ideas of this algorithm are similar to `Double-Monster` (Algorithm 1) which also runs in epochs with a predetermined epoch schedule $\tau_0, \tau_1, \ldots$, and tries to estimate the underlying linear score mapping $\phi^*$ with $\phi_m$ at each epoch $m$. However, the key difference from `Double-Monster` (Algorithm 1) lies in the duel-selection routine which requires using a different *convex constraint solver for continuous domains* as it is not possible to maintain distribution over pairs of actions in the continuous space. The key steps are:

At the beginning of each epoch $m$, the algorithm computes an estimated scoring function $\phi_m$ using the offline regression oracle that yields:

$$\phi_m \leftarrow \arg\min_{\phi \in \Phi} \sum_{\tau = \tau_{m-2}+1}^{\tau_{m-1}} (\sigma(\langle \phi(x_\tau), a_\tau - b_\tau \rangle) - o_\tau)^2.$$

Once $\phi_m$ is computed, at any time $t$ in epoch $m$, given the context $x_t$ the estimated least square function estimate $\phi_m$, and some tuning parameter $\gamma_m = \left( \dfrac{\sqrt{d}}{2\sqrt{2\mathcal{E}_{\text{off},\Phi^*}(\delta/(2m^2), \tau_{m-1} - \tau_{m-2})}} \right)$, we query the `Cont-CvxConstraint-Solver` with the triplet $(x_t, \phi_m, \gamma_m)$ and obtain $\mathbf{p}_t \in \Delta_{\mathcal{K} \times \mathcal{K}}$ using a continuous convex constraint solver. More precisely, `Cont-CvxConstraint-Solver` is defined as a (possibly randomized) algorithm, s.t. given any triplet $(x, \phi, \gamma)$, with $x \in \mathcal{X}, \phi \in \Phi$ and $\gamma \in \mathbb{R}$, it outputs $\mathbf{p} \in \Delta_{\mathcal{K} \times \mathcal{K}}$:

$$\mathbf{p} \leftarrow \text{argmax}_{\mathbf{q} \in \Delta(\mathcal{K} \times \mathcal{K})} \left[ \langle \mathbf{ab_q}, \phi(x) \rangle + \tfrac{1}{\gamma} \log \det(\mathbf{H_q}) \right], \quad \text{where} \tag{6}$$

$\mathbf{ab_q} = \mathbf{E}_{(\mathbf{a},\mathbf{b}) \sim \mathbf{q}}[(\mathbf{a} + \mathbf{b})/2]$, $\mathbf{H_q} = \mathbf{E}_{(\mathbf{a},\mathbf{b}) \sim \mathbf{q}}[(\mathbf{a} - \mathbf{b})(\mathbf{a} - \mathbf{b})^\top] + c\mathbf{I}_d$, $c > 0$ is a tuning parameter.

*Above is the key step of Algorithm 2 which sets it apart from our previous arm selection technique of Algorithm 1 proposed for the finite action space. Formally, $\mathbf{p} \in \Delta_{\mathcal{K} \times \mathcal{K}}$ in Eq. (6) captures a distribution over 'high-scoring' action pairs (owing to the term '$\langle \mathbf{ab_p}, \phi(x) \rangle$'), with sufficient variability (owing to the term '$\frac{1}{\gamma} \log \det(\mathbf{H_p})$').* Importantly, the tradeoff between the first and second terms of Eq. (6): the first term would like to put mass on the actions that align with $\phi_m(x_t)$ — ensuring 'exploitation' of the regressed model $\phi_m$; the second term would want to put more mass on distinct pairs of actions in $\mathcal{K}$ to induce higher variability — The challenge was to incorporate this *explore-exploit* tradeoff in the duel-selection rule $\mathbf{p}_t$ for continuous action spaces.

Next, the algorithm simply samples a duel $(a_t, b_t) \sim \mathbf{p}_t$ from the joint distribution $\mathbf{p}_t$ and the rest of the algorithm proceeds almost the same as *Algorithm* 1 except for the choice of the tuning parameters $\gamma_m$. The complete pseudocode of the algorithm is given in Algorithm 2.

Notably, our arm selection technique is inspired from [17, Alg. 2], which introduced the idea of *logdet-barrier distribution* based arm-selection for contextual bandits which represents a regularized empirical risk minimization procedure. (1) We however had to modify their proposed distribution due to the dueling (preference) nature of the feedback. (2) More importantly, our regret analysis is different as their analysis relies on the existence of an online regression oracle while we design and analyze all algorithms with (the weaker) offline oracles as motivated in Section 1 and Section 3.

## 6.1 Performance Guarantees of Algorithm 2.

**Theorem 11** (Regret Analysis of Algorithm 2). *Assume $\forall x \in \mathcal{X}, \|\phi^*(x)\| \leq D$ for some $D > 0$, and suppose we choose $c = \frac{5d}{64D^2 \gamma_{m(T)}}$. Then under realizability (Assumption 1) of the function class $\Phi$, and with an epoch schedule $\tau_1, \ldots, \tau_m$ such that $\tau_m \geq 2^m$ for $m \leq \log T$, with probability at least $(1 - \delta)$ for any $\delta \in (0, 1)$ the best-response regret of Algorithm 2, $\text{BR-Reg}_T^{(cont)}$, in $T$ rounds $\mathcal{K}$ is bounded by:* $O \left( \sum_{m=2}^{m(T)} \sqrt{d\mathcal{E}_{\text{off},\Phi^*}(\delta/(2m^2), \tau_{m-1} - \tau_{m-2})(\tau_m - \tau_{m-1})} \right)$.

Tallying the result of Theorem 11 with [17, Theorem 2], which however only applies to the simpler setting of reward feedback models, one can claim that regret guarantee of Algorithm 2 is optimal up to log factors. The proof analysis of Theorem 11 is given in Appendix D.1.

## 7 Experiments

In this section, we report the empirical regret performance of our proposed algorithm `Double-Monster-Inf`(Algorithm 2) on different environments for general continuous decision spaces. All results are averaged across 100 runs.

We set the decision set $\mathcal{K} \subset \mathbb{R}^d$ to the unit ball in dimension-$d$. We run experiments on two different problem instances (instantiated by its feature maps $\phi$): (1) **Inst-1:** Here the preference class $\mathcal{M}$ is characterized by a linear feature map $\phi : \mathcal{X} \mapsto \mathbb{R}^d$, where for any $x \in \mathcal{X} \subset \mathbb{R}^c$, context dimension $c \in \mathbb{N}$, s.t. $\phi(x) = \frac{Zx}{\|Zx\|}$, for some arbitrary choice of $Z \in \mathbb{R}^{d \times c}$. For our experiment, we set $c = 8$, and report the experiments for $d = 2, 5, 10, 15$. (2) **Inst-2:** Here the preference class $\mathcal{M}$ is characterized by a quadratic feature map $\phi : \mathcal{X} \mapsto \mathbb{R}^d$, where for any $x \in \mathcal{X} \subset \mathbb{R}^c$, $c \in \mathbb{N}$, s.t. $\phi(x) = (x_1, \ldots, x_c, x_1^2, \ldots, x_c^2, x_1 x_2, \ldots, x_{c-1} x_c, 1) \in R^{2c + \binom{c}{2} + 1}$. Note thus $d = 2c + \binom{c}{2} + 1$. We consider $c = 1, 2, 3, 4$ respectively, yielding $d = 3, 6, 10, 15$.

**Regret with increasing $d$.** As shown in the figure below, we see that for both the above instances, the contextual regret scales as $\tilde{O}(\sqrt{dT})$ corroborating Theorem 11. Note our decision space is fully continuous and with infinite arms, unlike the previous work of [30], which also used online regression oracles. Also, our preference relations are much more general that can be built on non-linear utilities (e.g., Inst 2), unlike the previous attempts of [31, 25, 19], which were also inefficient to implement additionally. Consequently, these works could only report experiments on finite decision spaces.

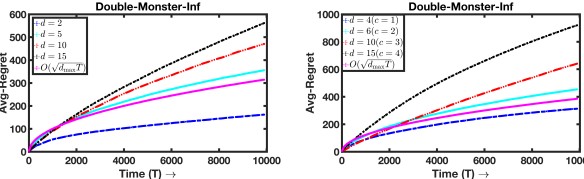

Figure 1: (Left) Inst-1 (linear transformation) and (Right) Inst-2 (quadratic transformation).

**Comparing with Non-Contextual Baselines.** We ran experiments with `Double-Monster-Inf` for finite-$K$ armed non-contextual MAB setting and compared with the following existing finite-arm Dueling-Bandit baselines: (i) RUCB [43], (ii) REX3 [18], (iii) [35], (iv) RMED [20]. Results show we outperformed almost every baseline and performed comparably with DTS, even though these baselines only apply to finite-amended non-contextual settings. We assumed 3 different types of scores $s$: - Base(20 arms): $s(1) = 1$, $s(i) = 0.7$ for $i \in \{2, ..., 10\}$, and $s(i) = 0.4$ for $i \in \{11, ..., 20\}$. - Hard (20 arms): $s(1) = 1$ and $s(i) = \theta_{i-1} - 0.05$ for $i \in \{2, ..., 20\}$. - Worst Case (WC, 40 arms): Considers a worst-case instance $s(1) = 1$ and $s(i) = 0.9$ for $i \in \{2, ..., 40\}$.

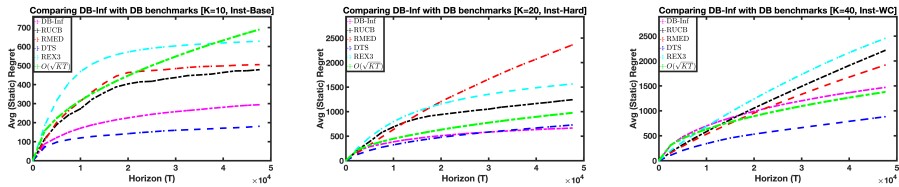

Figure 2: Comparing with Finite-armed Non-Contextual Baselines

**Runtime.** We also report the runtime comparison of different algorithms in Appendix E.

## 8 Perspectives

This work provides the first computationally efficient and near-optimal algorithm for contextual dueling bandits using offline oracles, resolving a key open problem from [30] and enabling scalable solutions for preference-based personalized learning in real-world applications. Additionally, we, for the first time, analyze the problem of for the continuous action space and propose a near-optimal $\tilde{O}(\sqrt{dT})$ regret algorithm for this setting. We hope this advancement will offer more practical and scalable solutions for contextual dueling bandits, contributing to more real-world human-centric prediction models, including LLMs, human-assisted robotics, recommender systems, etc.

**Future Work.** Moving forward it would be interesting to analyze the problem beyond pairwise preference, generalizing it to mulitset/ ranking preferences. Additionally, does the complexity of the performance limit alter when employing an offline oracle compared to an online oracle? Deploying the proposed algorithms in practical use-case scenarios, e.g., tuning LLMs, training autonomous vehicles, or AI-alignment in robotics, would also be an interesting empirical study. Another interesting direction would be to extend the proposed algorithmic framework to more general frameworks, like reinforcement learning or RLHF, and understand the corresponding theoretical guarantees.

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

# Supplementary: Efficient and Near-Optimal Algorithm for Contextual Dueling Bandits with Offline Regression Oracles

## A A Primer on Regression Oracles

In this section, we introduce the concepts of offline and online regression oracles to familiarize readers with their structural disparities and performance-related distinctions.

### A.1 Offline Regression Oracles [32]

Consider any abstract input space $\mathcal{Z}$ and output space $\mathcal{Y}$. Assume a given dataset $D_n := \{(z_i, y_i)\}_{i=1}^n$ consists of $n$ data points, each (input, output) pair $(z_i, y_i) \overset{iid}{\sim} \mathcal{D}$ being drawn iid from a fixed underlying distribution $\mathcal{D}$ on $\mathcal{Z} \times \mathcal{Y}$. Given a general function class $\mathcal{F} \subseteq \{f \mid f : \mathcal{Z} \mapsto \mathcal{Y}\}$, a general offline regression oracle associated with $\mathcal{F}$, denoted by $\texttt{OffReg}_\mathcal{F}$ is an algorithm, operates on the given dataset $D_n$ and outputs a mapping $\hat{f} : \mathcal{Z} \mapsto \mathcal{Y}$. In learning theory, the quality of $\hat{f}$ is measured by its "out-of-sample error," i.e., its expected error on *random* and *unseen* test data.

**Assumption 4** (Guarantee of Offline Regression Oracles)**.** *Consider a general function class $\mathcal{F} \subseteq \{f \mid f : \mathcal{Z} \mapsto \mathcal{Y}\}$. Then given a dataset $D_n := \{(z_i, y_i)\}_{i=1}^n$ that consists of $n$ data points, s.t. $(z_i, y_i) \overset{iid}{\sim} \mathcal{D}$, we assume that the output $\hat{f}_n \leftarrow \texttt{OffReg}_\mathcal{F}(D_n)$ of the offline regression oracle $\texttt{OffReg}$ satisfies that: For any $\delta \in (0, 1]$, with probability at least $1 - \delta$,*

$$\mathbf{E}_{(z,y)\sim\mathcal{D}}\left[\left(\hat{f}_n(z) - y\right)^2 - \inf_{f\in\mathcal{F}}\left(f(z) - y\right)^2\right] \leq \mathcal{E}_{\text{off},\mathcal{F}}(\delta, n),$$

*where $\mathcal{E}_{\text{off},\mathcal{M}}(\delta, n)$ denotes the square loss of the regression oracle when trained on $n$-iid instances.*

**Assumption 4 under Realizability.** Moreover, if we further assume realizability, in the sense that there exists $f^* \in \mathcal{F}$ such that $f^*(z) = \mathbb{E}_{(z,y)\sim\mathcal{D}}[y \mid z]$, then it is well-known that Assumption 4 further implies: For any $\delta \in (0, 1]$, with probability at least $1 - \delta$, we have

$$\mathbf{E}_{(z,y)\sim\mathcal{D}}\left[\left(\hat{f}_n(z) - f^\star(z)\right)^2\right] \leq \mathcal{E}_{\text{off},\mathcal{F}}(\delta, n). \tag{7}$$

The offline regression error bound $\mathcal{E}_{\text{off},\mathcal{F}}(n, \delta)$ is a function that decreases with $n$. Note, for our problem, we have $\mathcal{Z} = (\mathcal{X} \times [K]^2)$ to be the set of (context, action-pair) tuples such that $z_t = (x_t, a_t, b_t)$ with $x_t \in \mathcal{X}$ and $(a_t, b_t) \in [K]^2$. Our output space is $\mathcal{Y} = [-1, 1]$ as note $o_t \in \mathcal{Y}$.

### A.2 Online Regression Oracles

An online regression oracle [11, Chapter 3], is an algorithm, which we denote by $\texttt{OnReg}$, and which operates in the following online protocol on each round $t$: (1) it receives an abstract input $z_t \in \mathcal{Z}$ from some input space $\mathcal{Z}$, chosen adversarially by the environment; (2) it produces a real-valued prediction $\hat{y}_t \in \mathcal{Y} \subset \mathbb{R}$ where $\mathcal{Y}$ is some output space; (3) it observes the true response $y_t \in \mathcal{Y}$ and incurs loss $\ell(\hat{y}_t, y_t) := (\hat{y}_t - y_t)^2$. The goal of the oracle is to predict the outcomes as well as the best function in a given function class $\mathcal{F} := \{f : \mathcal{Z} \mapsto \mathbb{R}\}$, such that for every sequence of outcomes, the square loss regret is bounded.

More formally, at time $t$ and for input $z \in \mathcal{Z}$, the online regression oracle associated to the function class $\mathcal{F}$, denoted by $\texttt{OnReg}_\mathcal{F}$, can be seen as a mapping $\hat{y}_t(z) := \texttt{OnReg}_\mathcal{F}(z, \{z_\tau, y_\tau\}_{\tau=1}^{t-1})$. Note this corresponds to the prediction the algorithm would make at time $t$ if we passed in the input $z$, although this input may not be what is ultimately selected by the environment.

**Assumption 5.** *Online Regression Oracle Guarantee: Given any function class $\mathcal{F} \subseteq \{f \mid f : \mathcal{Z} \mapsto \mathcal{Y}\}$, the online regression oracle $\texttt{OnReg}_\mathcal{F}$ guarantees for every sequence $\{(z_t, y_t)\}_{t\in[n]}$, its regret is bounded as:*

$$\frac{1}{n}\left[\sum_{t=1}^n \left(\hat{y}_t(z_t) - y_t\right)^2 - \inf_{f\in\mathcal{F}}\sum_{t=1}^n\left(f(z_t) - y_t\right)^2\right] \leq \mathcal{E}_{\text{on},\mathcal{F}}(n),$$

*where $\mathcal{E}_{\text{on},\mathcal{F}}(n) = o(1)$ is a known upper bound and a decreasing function of $n$.*

**Assumption 5 under Realizability.** Moreover, if we further assume realizability, in the sense that there exists $f^\star \in \mathcal{F}$ such that $\forall t : f^\star(z_t) = \mathbb{E}[y_t \mid z_t]$, then one can show that the guarantee further implies:

$$\frac{1}{n}\left[ \sum_{t=1}^{n} \left(\hat{y}_t(z_t) - f^\star(z_t)\right)^2 \right] \leq \mathcal{E}_{\mathrm{on},\mathcal{F}}(n). \tag{8}$$

**Remark 12** (Some examples). *Online square loss regression is a well-studied problem, and efficient algorithms with provable regret guarantees are known for many specific function classes including finite classes where $|\mathcal{F}| < \infty$, finite and infinite dimensional linear classes, and others [16, 17]. For completeness, we provide formal definition for some specific classes and instantiations of the regression oracles in Appendix A.3.*

### A.3 Examples: Some Specific Regression Function Classes:

1. Finite regression function class $\mathcal{F}$ such that $|\mathcal{F}| < \infty$.
2. Class of linear predictors $\mathcal{F} := \{(x,a) \mapsto \langle \theta, x_a \rangle \mid \theta \in \mathbb{R}^d, \|\theta\| \leq 1\}$.
3. Class of generalized linear predictors $\mathcal{F} := \{(x,a) \mapsto \sigma(\langle \theta, x_a \rangle) \mid \theta \in \mathbb{R}^d, \|\theta\| \leq 1\}$ where $\sigma : \mathbb{R} \mapsto [0,1]$ is a fixed non-decreasing 1-Lipschitz link function.
4. Reproducing kernel hilbert space (RKHS) $\mathcal{F} := \{f \mid \|f\|_{\mathcal{H}} \leq 1, \mathcal{K}(x_a, x_a) \leq 1\}$.
5. Banach Spaces $\mathcal{F} := \{(x,a) \mapsto \langle \theta, x_a \rangle \mid \theta \in \mathcal{B}, \|\theta\| \leq 1\}$, where $(\mathcal{B}, \|\cdot\|)$ is a separable Banach space and $x$ belongs to the dual space $(\mathcal{B}, \|\cdot\|_*)$.

**Remark 13** (Online vs Offline Oracles). Offline regression is a batch-processing approach that involves analyzing the entire dataset at once to build a regression model. The advantage of offline oracles lies in that they can consider all the data available for training, the training data is generated iid from some underlying distribution and therefore, can often produce a more accurate model. On the contrary, $\mathtt{OnReg}_{\mathcal{F}}$ requires a much stronger guarantee compared to $\mathtt{OffReg}_{\mathcal{F}}$ since they are required to fit any arbitrary test instance, even adversarially generated (input,output) pair $(z_t, y_t)$, whereas an offline oracle only promises that when $(z_t, y_t) \sim \mathcal{D}$ is sampled from the same distribution $\mathcal{D}$ where it has been trained. Overall, the choice between offline and online regression oracles depends on the specific use case. If the entire dataset is available and accuracy is the primary concern, then offline regression may be the better approach. However, if real-time processing and adaptability are required, then online regression may be a better choice; although finding efficient online oracles for any arbitrary function class could be hard (e.g. see [23, Example 3], [22, Example 1]). It is also worth noting that an online oracle can always be converted to an offline one but not vice-versa since the former offers a much stronger regression guarantee compared to the latter.

## B Appendix for Section 4

### B.1 `Double-Monster` (Algorithm 1): Pseudocode of Algorithm for the Best-Response Regret

### B.2 Justification of existence of $\mathbf{p}$ in Eq. (2)

*Proof.* Let us start by recalling the Eq. (2) requires finding a $\mathbf{p} \in \Delta_{K \times K}$ such that:

$$\forall i, j \in [K], \mathbf{E}_{a \sim \mathbf{p}^\ell}\left[\widehat{\mathbf{M}}(x)[i,b]\right] + \mathbf{E}_{b \sim \mathbf{p}^r}\left[\widehat{\mathbf{M}}(x)[j,b]\right] + \frac{2}{\gamma}\frac{1}{p(i,j)} \leq \frac{5K^2}{\gamma}.$$

Towards proving this, let us fix the context $x \in \mathcal{X}$ and denote by $\mathbf{P} = \widehat{\mathbf{M}}(x)$. Then note we essentially need to solve the following minimax game (for any arbitrary preference matrix $\mathbf{P}$):

$$\min_{\mathbf{p} \in \Delta_{K \times K}} \max_{\mathbf{q}, \mathbf{q}' \in \Delta_K} \left[ \sum_{i,k} q(i) P[i,k] p^\ell(k) + \sum_{j,k} q'(j) P[j,k] p^r(k) + \frac{2}{\gamma} \sum_{i,j} \frac{q(i)q'(j)}{p(i,j)} \right].$$

**Algorithm 1 `Double-Monster`: Efficient Contextual Dueling Bandit with Offline Regressors**

1: **input** Epoch schedule $0 = \tau_0 < \tau_1 < \tau_2 < \cdots$. Confidence parameter $\delta$. Tuning parameter
$\gamma_1, \gamma_2, \ldots$.

2: Arm set: $[K]$. An instance of `OffReg` for function class $\mathcal{M}$

3: **for** epoch $m = 1, 2, \ldots$ **do**

4: Let $\gamma_m = \left( \dfrac{K}{\sqrt{\mathcal{E}_{\text{off},\mathcal{M}}(\delta/(2m^2), \tau_{m-1} - \tau_{m-2})}} \right)$ (for epoch 1, $\gamma_1 = 1$).

5: Compute $\widehat{\mathbf{M}}_m \leftarrow \text{OffReg}\left( \{(x_\tau, a_\tau, b_\tau), o_\tau\}_{\tau = \tau_{m-2}+1}^{\tau_{m-1}} \right)$, i.e.

$\widehat{\mathbf{M}}_m = \arg\min_{\mathbf{M} \in \mathcal{M}} \sum_{\tau = \tau_{m-2}+1}^{\tau_{m-1}} (\mathbf{M}(x_\tau)[a_\tau, b_\tau] - \tilde{o}_\tau)^2$ via the **offline least squares oracle**, where $\tilde{o}_\tau = 2o_\tau - 1$, for any $\tau \in [\tau_{m-2} + 1, \tau_{m-1}]$.

6: **for** round $t = \tau_{m-1} + 1, \cdots, \tau_m$ **do**

7: Observe context $x_t \in \mathcal{X}$.

8: Compute $\mathbf{p}_t \sim \text{Convex-Constraint-Solver}(x_t, \widehat{\mathbf{M}}_m, \gamma_m)$ s.t. $\mathbf{p}_t$ satisfies:

$$\forall i, j \in [K] : \mathbf{E}_{a \sim \mathbf{p}_t^\ell}\left[ \widehat{\mathbf{M}}_m(x)[i, a] \right] + \mathbf{E}_{b \sim \mathbf{p}_t^r}\left[ \widehat{\mathbf{M}}_m(x)[j, b] \right] + \frac{2}{\gamma_m} \frac{1}{p_t(i,j)} \leq \frac{5K^2}{\gamma_m}.$$

9: Sample $(a_t, b_t) \sim \mathbf{p}_t$ and observe preference feedback $o_t \sim \text{Ber}\left( \frac{\mathbf{M}^\star(x_t)[a_t, b_t]+1}{2} \right)$.

10: **end for**

11: **end for**

---

Let us denote the minmax value of the above game, associated to the preference matrix $\mathbf{P}$ by
$V(\mathbf{P}) := \min_{\mathbf{p} \in \Delta_{K \times K}} \max_{\mathbf{q}, \mathbf{q}' \in \Delta_K} \left[ \mathbf{q}^\top \mathbf{P} \mathbf{p}^\ell + \mathbf{q}'^\top \mathbf{P} \mathbf{p}^r + \frac{2}{\gamma} \sum_{i,j} \frac{q(i)q'(j)}{p(i,j)} \right]$.

Now fixing $\epsilon > 0$, for any $\mathbf{p} \in \Delta_{K \times K}$ we define the $\epsilon$-relaxation of $\mathbf{p}$ as $\mathbf{p}_{(\epsilon)} := (1 - \epsilon)\mathbf{p} + \epsilon \mathbf{1}/K^2$.
As $\mathbf{p}^{(\epsilon)}$ itself is a distribution, this upper bounds our objective while ensuring that the conditions for applying the Sion's minimax theorem are satisfied [3]. As such, we obtain

$$V(\mathbf{P}) \leq \min_{\mathbf{p} \in \Delta_{K \times K}} \max_{\mathbf{q}, \mathbf{q}' \in \Delta_K} \left[ \mathbf{q}^\top \mathbf{P} \mathbf{p}_{(\epsilon)}^\ell + \mathbf{q}'^\top \mathbf{P} \mathbf{p}_{(\epsilon)}^r + \frac{2}{\gamma} \sum_{i,j} \frac{q(i)q'(j)}{p_{(\epsilon)}(i,j)} \right]$$

$$= \max_{\mathbf{q}, \mathbf{q}' \in \Delta_K} \min_{\mathbf{p} \in \Delta_{K \times K}} \left[ \mathbf{q}^\top \mathbf{P} \mathbf{p}_{(\epsilon)}^\ell + \mathbf{q}'^\top \mathbf{P} \mathbf{p}_{(\epsilon)}^r + \frac{2}{\gamma} \sum_{i,j} \frac{q(i)q'(j)}{p_{(\epsilon)}(i,j)} \right].$$

Now let us define another distribution $\tilde{\mathbf{p}} \in \Delta_{K \times K}$ such that $\mathbf{qq}(i,j) = q(i)q'(j) \; \forall i, j \in [K]$. It is easy to note that $\mathbf{qq}^\ell = \mathbf{q}$ and $\mathbf{qq}^r = \mathbf{q}'$. Similarly, defining the $\epsilon$-approximation of $\mathbf{qq}$ as
$\mathbf{qq}_\epsilon := (1 - \epsilon)\mathbf{qq} + \epsilon \mathbf{1}/K^2$, we further note $\mathbf{qq}_\epsilon^\ell = (1 - \epsilon)\mathbf{q} + \epsilon \mathbf{1}/K$, and $\mathbf{qq}_\epsilon^r = (1 - \epsilon)\mathbf{q}' + \epsilon \mathbf{1}/K$.
Then replacing $\mathbf{p}$ by $\mathbf{qq}$ in the above expression, we can further upper bound $V(\mathbf{P})$ by:

$$V(\mathbf{P}) \leq \max_{\mathbf{q}, \mathbf{q}' \in \Delta_K} \left[ \sum_{i,k} q(i)P[i,k]qq_\epsilon^\ell(k) + \sum_{j,k} q'(j)P[j,k]qq_\epsilon^r(k) + \frac{2}{\gamma} \sum_{i,j} \frac{q(i)q'(j)}{qq_\epsilon(i,j)} \right]$$

$$= \max_{\mathbf{q}, \mathbf{q}' \in \Delta_K} \left[ (1 - \epsilon) \sum_{i,k} q(i)P[i,k]qq_\epsilon^\ell(k) + \frac{\epsilon}{K} \sum_{i,k} q(i)P[i,k] \right.$$

$$\left. + (1 - \epsilon) \sum_{j,k} q'(j)P[j,k]qq_\epsilon^r(k) + \frac{\epsilon}{K} \sum_{j,k} q'(j)P[j,k] + \frac{2}{\gamma} \sum_{i,j} \frac{q(i)q'(j)}{(1 - \epsilon)q(i)q'(j) + \epsilon/K^2} \right]$$

---

[3] Note that the expression is convex in $\mathbf{p}$ and quasi-concave in $\mathbf{qq}'$, as required, following [6].

$$\leq \max_{\mathbf{q}, \mathbf{q}' \in \Delta_K} \left[ (1-\epsilon) \sum_{i,k} q(i)P[i,k]q(k) + \frac{\epsilon}{K} \sum_i q(i) \right.$$

$$\left. + (1-\epsilon) \sum_{j,k} q'(j)P[j,k]q'(k) + \frac{\epsilon}{K} \sum_j q'(j) + \frac{2}{\gamma} \sum_{i,j} \frac{q(i)q'(j)}{(1-\epsilon)q(i)q'(j)} \right]$$

Here, the first inequality restricts the domain for $\mathbf{p}$ using the smoothing operator, the first equality is the minimax swap, and the second inequality follows by choosing $\mathbf{p} = \mathbf{q}$. The remaining three terms are bounded as follows: (i) the first term is zero since $\mathbf{P}$ is a preference matrix, (ii) the second term could trivially be upper bounded by $\epsilon$, and (iii) the third term is at most $4K^2/\gamma$ as long as $\epsilon \leq 1/2$. Setting $\epsilon = K^2/2\gamma$, we obtain the result, as long as $\gamma \geq K^3$. □

## C  Appendix for Section 5

### C.1  Proof of Lemmas

**Lemma 7** (Properties of decision policy $\boldsymbol{\pi}_m$ in Algorithm 1). *At any epoch $m$, $\forall \boldsymbol{\pi}, \boldsymbol{\pi}' \in \Pi$, the decision policy of epoch $m$ $\boldsymbol{\pi}_m$ satisfies:*

$$\widehat{\mathrm{BReg}}_m(\boldsymbol{\pi}_m^\ell) + \widehat{\mathrm{BReg}}_m(\boldsymbol{\pi}_m^r) \leq \frac{5K^2}{\gamma_m},$$

$$2\mathcal{V}_m(\boldsymbol{\pi}, \boldsymbol{\pi}') \leq 5K^2 + \gamma_m \widehat{\mathrm{Reg}}_m(\boldsymbol{\pi}_m^\ell, \boldsymbol{\pi}) + \gamma_m \widehat{\mathrm{Reg}}_m(\boldsymbol{\pi}_m^r, \boldsymbol{\pi}').$$

*Proof of Lemma 7.* Note by our algorithm selection rule, at any time $t$ in epoch $m$, our decision policy $\boldsymbol{\pi}_m : \mathcal{X} \mapsto \Delta_{K \times K}$ (see Definition 2) satisfies:

$$\forall \mathbf{q} \text{ and } \mathbf{q}' \in \Delta_K : \mathbf{q}^\top \widehat{\mathbf{M}}_m(x_t)\boldsymbol{\pi}_m^\ell + \mathbf{q}'^\top \widehat{\mathbf{M}}_m(x_t)\boldsymbol{\pi}_m^r + \frac{2}{\gamma_m} \mathbf{E}_{i \sim \mathbf{q}, j \sim \mathbf{q}'} \left[ \frac{1}{\pi_m(i,j \mid x)} \right] \leq \frac{5K^2}{\gamma_m},$$

but this implies that for any policy $\boldsymbol{\pi} \in \Pi$ and $\boldsymbol{\pi}' \in \Pi$,

$$\boldsymbol{\pi}(x_t)^\top \widehat{\mathbf{M}}_m(x_t)\boldsymbol{\pi}_m^\ell(x_t) + \boldsymbol{\pi}'(x_t)^\top \widehat{\mathbf{M}}_m(x_t)\boldsymbol{\pi}_m^r(x_t) + \frac{2}{\gamma_m} \sum_{i=1}^K \sum_{j=1}^K \frac{\pi(i|x_t)\pi'(j|x_t)}{\boldsymbol{\pi}_m(i,j|x_t)} \leq \frac{5K^2}{\gamma_m}.$$

Further taking expectation over $x_t \sim \mathcal{D}_\mathcal{X}$ along with Definition 3 and Definition 6, above inequality yields:

$$\widehat{\mathrm{Reg}}_m(\boldsymbol{\pi}, \boldsymbol{\pi}_m^\ell) + \widehat{\mathrm{Reg}}_m(\boldsymbol{\pi}', \boldsymbol{\pi}_m^r) + \frac{2\mathcal{V}_m(\boldsymbol{\pi}, \boldsymbol{\pi}')}{\gamma_m} \leq \frac{5K^2}{\gamma_m}. \tag{9}$$

This proves the first part of the proof since by definition setting $\boldsymbol{\pi} = \psi^\star[\boldsymbol{\pi}_m^\ell, \mathbf{M}^\star]$, $\boldsymbol{\pi}' = \psi^\star[\boldsymbol{\pi}_m^r, \mathbf{M}^\star]$ and the fact $\mathcal{V}_m(\boldsymbol{\pi}, \boldsymbol{\pi}') \geq 0$.

Additionally, due to the anti-symmetric property of preference matrix $P$, we have $\widehat{\mathrm{Reg}}_m(\boldsymbol{\pi}_m^\ell, \boldsymbol{\pi}) = -\widehat{\mathrm{Reg}}_m(\boldsymbol{\pi}, \boldsymbol{\pi}_m^\ell)$ and $\widehat{\mathrm{Reg}}_m(\boldsymbol{\pi}_m^r, \boldsymbol{\pi}') = -\widehat{\mathrm{Reg}}_m(\boldsymbol{\pi}', \boldsymbol{\pi}_m^r)$ (see Remark 3). Applying this which further to Eq. (9) we get:

$$2\mathcal{V}_m(\boldsymbol{\pi}, \boldsymbol{\pi}') \leq 5K^2 + \gamma_m \widehat{\mathrm{Reg}}_m(\boldsymbol{\pi}_m^\ell, \boldsymbol{\pi}) + \gamma_m \widehat{\mathrm{Reg}}_m(\boldsymbol{\pi}_m^r, \boldsymbol{\pi}').$$

This proves the second part of the proof. □

## C.2 Proof of Lemma 8

**Lemma 8.** *For any epoch $m$ and any two decision policies $\boldsymbol{\pi}$ and $\boldsymbol{\pi}'$,*

$$\mathrm{Reg}(\boldsymbol{\pi}', \boldsymbol{\pi}) - \widehat{\mathrm{Reg}}_m(\boldsymbol{\pi}', \boldsymbol{\pi}) \leq \frac{\mathcal{V}_{m-1}(\boldsymbol{\pi}', \boldsymbol{\pi})}{3\gamma_m} + \frac{3K^2}{4\gamma_m}.$$

*Proof of Lemma 8.* Let us analyze the difference between the true and empirical response regret at any time $t$:

$$\mathrm{Reg}(\boldsymbol{\pi}', \boldsymbol{\pi}) - \widehat{\mathrm{Reg}}_t(\boldsymbol{\pi}', \boldsymbol{\pi}) = \mathbf{E}_{x \sim \mathcal{D}}\left[\boldsymbol{\pi}'(x)^\top \mathbf{M}^\star(x)\boldsymbol{\pi}(x) - \boldsymbol{\pi}'(x)^\top \widehat{\mathbf{M}}_{m(t)}\boldsymbol{\pi}(x)\right]$$

$$= \mathbf{E}_{x \sim \mathcal{D}}\left[\boldsymbol{\pi}'(x)^\top \left(\mathbf{M}^\star(x) - \widehat{\mathbf{M}}_{m(t)}\right)\boldsymbol{\pi}(x)\right].$$

Let us consider any $x \in \mathcal{X}$, and denote by $\mathbf{q} = \boldsymbol{\pi}'(x)$, $\mathbf{p} = \boldsymbol{\pi}(x)$, $\mathbf{p}_{m-1} = \boldsymbol{\pi}_{m-1}(x)$, $m(t) = m$. Now the term inside the expectation can be further bounded as:

$$\mathbf{q}^\top \left(\mathbf{M}^\star(x) - \widehat{\mathbf{M}}_m(x)\right)\mathbf{p} = \sum_{a=1}^{K}\sum_{i=1}^{K} q(a)p(i)\left[\mathbf{M}^\star(x)[a,i] - \widehat{\mathbf{M}}_m(x)[a,i]\right]$$

$$\leq \sum_{a=1}^{K}\sum_{i=1}^{K} \frac{q(a)}{\gamma_m^{1/2}p_{m-1}(a)^{1/2}} \frac{p(i)}{p_{m-1}(i)^{1/2}} \gamma_m^{1/2} p_{m-1}(a)^{1/2} p_{m-1}(i)^{1/2}\left[\left|\mathbf{M}^\star(x)[a,i] - \widehat{\mathbf{M}}_m(x)[a,i]\right|\right],$$

$$\overset{(a)}{\leq} \frac{1}{3\gamma_m}\sum_{a=1}^{K}\sum_{i=1}^{K} \frac{q(a)^2 p^2(i)}{p_{m-1}(a)p_{m-1}(i)} + \frac{3\gamma_m}{4}\sum_{a=1}^{K}\sum_{i=1}^{K} p_{m-1}(a)p_{m-1}(i)(\mathbf{M}^\star(x)[a,i] - \widehat{\mathbf{M}}_m(x)[a,i])^2$$

$$\leq \frac{1}{3\gamma_m}\sum_{a=1}^{K}\sum_{i=1}^{K} \frac{q(a)p(i)}{p_{m-1}(a)p_{m-1}(i)} + \frac{3\gamma_m}{4}\sum_{a=1}^{K}\sum_{i=1}^{K} p_{m-1}(a)p_{m-1}(i)(\mathbf{M}^\star(x)[a,i] - \widehat{\mathbf{M}}_m(x)[a,i])^2$$

where (a) uses the Cauchy-Schwarz inequality followed by AM-GM inequality. Using similar ideas one can derive:

$$\mathbf{E}_{x \sim \mathcal{D}_\mathcal{X}}\left[\boldsymbol{\pi}'(x)^\top\left(\mathbf{M}^\star(x) - \widehat{\mathbf{M}}_m(x)\right)\boldsymbol{\pi}(x)\right]$$

$$\leq \frac{1}{3\gamma_m}\sum_{a=1}^{K}\sum_{i=1}^{K} \frac{\pi'(a \mid x)\pi(i \mid x)}{\pi_{m-1}(a \mid x)\pi_{m-1}(i \mid x)} + \frac{3\gamma_m}{4}\sum_{a=1}^{K}\sum_{i=1}^{K} \pi_{m-1}(a \mid x)\pi_{m-1}(i \mid x)(\mathbf{M}^\star(x)[a,i] - \widehat{\mathbf{M}}_m(x)[a,i])^2$$

$$= \frac{1}{3\gamma_m}\mathcal{V}_{m-1}(\boldsymbol{\pi}', \boldsymbol{\pi}) + \frac{3\gamma_m}{4}\mathcal{E}_{\mathrm{off},\mathcal{M}}\big(\delta/2m^2, \tau_{m-1} - \tau_{m-2}\big) = \frac{\mathcal{V}_{m-1}(\boldsymbol{\pi}', \boldsymbol{\pi})}{3\gamma_m} + \frac{3K^2}{4\gamma_m},$$

where the second last equation follows from Assumption 4 and the last equation from the choice of

$$\gamma_m = \left(\frac{K^2}{\mathcal{E}_{\mathrm{off},\mathcal{M}}\big(\delta/2m^2, \tau_{m-1} - \tau_{m-2}\big)}\right)^{1/2}. \qquad \square$$

## C.3 Proof of Lemma 10

Recall the notations introduced in Section 4.1, that will be used throughout the proof.

**Lemma 10** (Epochwise Recursion). *Let $\boldsymbol{\pi}$ be any policy in $\Pi^2$. Then for all epochs $m \in \mathbb{N}_+$:*

$$\mathrm{BReg}(\boldsymbol{\pi}^\ell) + \mathrm{BReg}(\boldsymbol{\pi}^r) \leq 2\widehat{\mathrm{BReg}}_m(\boldsymbol{\pi}^\ell) + 2\widehat{\mathrm{BReg}}_m(\boldsymbol{\pi}^r) + 11K^2/\gamma_m,$$

$$\widehat{\mathrm{BReg}}_m(\boldsymbol{\pi}^\ell) + \widehat{\mathrm{BReg}}_m(\boldsymbol{\pi}^r) \leq 2\mathrm{BReg}(\boldsymbol{\pi}^\ell) + 2\mathrm{BReg}(\boldsymbol{\pi}^r) + 11K^2/\gamma_m.$$

*Proof of Lemma 10.* Let us fix $c_0 = 11$ for the rest of the proof. We prove the claim via induction on $m$. We first consider the base case where $m = 1$ and $1 \leq t \leq \tau_1$. In this case, since $\gamma_1 = 1$,

$$\mathrm{BReg}(\boldsymbol{\pi}^\ell) + \mathrm{BReg}(\boldsymbol{\pi}^r) \leq 2 \leq c_0 K^2/\gamma_1, \quad \widehat{\mathrm{BReg}}_1(\boldsymbol{\pi}^\ell) + \widehat{\mathrm{BReg}}_1(\boldsymbol{\pi}^r) \leq 2 \leq c_0 K^2/\gamma_1$$

Thus the claim holds in the base case.

For the inductive step, let us fix some epoch $m > 1$. We assume that for all epochs $m' < m$, all $\boldsymbol{\pi} \in \Pi^2$,

$$\mathrm{BReg}(\boldsymbol{\pi}^\ell) + \mathrm{BReg}(\boldsymbol{\pi}^r) \leq 2\widehat{\mathrm{BReg}}_{m'}(\boldsymbol{\pi}^\ell) + 2\widehat{\mathrm{BReg}}_{m'}(\boldsymbol{\pi}^r) + c_0 K^2/\gamma_{m'}, \qquad (10)$$

$$\widehat{\mathrm{BReg}}_{m'}(\boldsymbol{\pi}^\ell) + \widehat{\mathrm{BReg}}_{m'}(\boldsymbol{\pi}^r) \leq 2\mathrm{BReg}(\boldsymbol{\pi}^\ell) + 2\mathrm{BReg}(\boldsymbol{\pi}^r) + c_0 K^2/\gamma_{m'}. \qquad (11)$$

**Part-1:** We will first show that for epoch $m$, and all $\boldsymbol{\pi} \in \Pi^2$,

$$\mathrm{BReg}(\boldsymbol{\pi}^\ell) + \mathrm{BReg}(\boldsymbol{\pi}^r) \leq 2\widehat{\mathrm{BReg}}_m(\boldsymbol{\pi}^\ell) + 2\widehat{\mathrm{BReg}}_m(\boldsymbol{\pi}^r) + c_0 K^2/\gamma_m.$$

For simplicity let us denote the policies $\boldsymbol{\psi}^\star[\boldsymbol{\pi}^\ell, \mathbf{M}^\star], \boldsymbol{\psi}^\star[\boldsymbol{\pi}^r, \mathbf{M}^\star]$ respectively by $\boldsymbol{\pi}^{\star,\ell}, \boldsymbol{\pi}^{\star,r}$, and $\boldsymbol{\psi}^\star[\boldsymbol{\pi}^\ell, \widehat{\mathbf{M}}_m], \boldsymbol{\psi}^\star[\boldsymbol{\pi}^r, \widehat{\mathbf{M}}_m]$, respectively by $\boldsymbol{\pi}_m^{\star,\ell}, \boldsymbol{\pi}_m^{\star,r}$. Then we start by noting that:

$$\mathrm{BReg}(\boldsymbol{\pi}^\ell) - \widehat{\mathrm{BReg}}_m(\boldsymbol{\pi}^\ell) = \mathrm{Reg}(\boldsymbol{\pi}^{\star,\ell}, \boldsymbol{\pi}^\ell) - \widehat{\mathrm{Reg}}_m(\boldsymbol{\pi}_m^{\star,\ell}, \boldsymbol{\pi}^\ell)$$

$$\leq \mathrm{Reg}(\boldsymbol{\pi}^{\star,\ell}, \boldsymbol{\pi}^\ell) - \widehat{\mathrm{Reg}}_m(\boldsymbol{\pi}^{\star,\ell}, \boldsymbol{\pi}^\ell), \qquad \text{(as } \widehat{\mathrm{Reg}}_m(\boldsymbol{\pi}_m^{\star,\ell}, \boldsymbol{\pi}^\ell) \geq \widehat{\mathrm{Reg}}_m(\boldsymbol{\pi}^{\star,\ell}, \boldsymbol{\pi}^\ell))$$

$$\overset{(a)}{\leq} \frac{\mathcal{V}_{m-1}(\boldsymbol{\pi}^{\star,\ell}, \boldsymbol{\pi}^\ell)}{3\gamma_m} + \frac{3K^2}{4\gamma_m},$$

where the first inequality is by the optimality of $\boldsymbol{\pi}_m^{\star,\ell}$ by definition and (a) follows from Lemma 8.

Similarly, we can show that

$$\mathrm{BReg}(\boldsymbol{\pi}^r) - \widehat{\mathrm{BReg}}_m(\boldsymbol{\pi}^r) \leq \frac{\mathcal{V}_{m-1}(\boldsymbol{\pi}^{\star,r}, \boldsymbol{\pi}^r)}{3\gamma_m} + \frac{3K^2}{4\gamma_m}.$$

Combining both we get:

$$\mathrm{BReg}(\boldsymbol{\pi}^\ell) + \mathrm{BReg}(\boldsymbol{\pi}^r) - [\widehat{\mathrm{BReg}}_m(\boldsymbol{\pi}^\ell) + \widehat{\mathrm{BReg}}_m(\boldsymbol{\pi}^r)] \leq \frac{\mathcal{V}_{m-1}(\boldsymbol{\pi}^{\star,\ell}, \boldsymbol{\pi}^\ell)}{3\gamma_m} + \frac{\mathcal{V}_{m-1}(\boldsymbol{\pi}^{\star,r}, \boldsymbol{\pi}^r)}{3\gamma_m} + \frac{3K^2}{2\gamma_m}. \qquad (12)$$

Now using Lemma 7 this implies for both $\boldsymbol{\pi}^\star$ and $\boldsymbol{\pi}^r$, we have:

$$2\mathcal{V}_{m-1}(\boldsymbol{\pi}^{\star,\ell}, \boldsymbol{\pi}^\ell) \leq 5K^2 + \gamma_{m-1}\widehat{\mathrm{Reg}}_{m-1}(\boldsymbol{\pi}_{m-1}^\ell, \boldsymbol{\pi}^{\star,\ell}) + \gamma_{m-1}\widehat{\mathrm{Reg}}_{m-1}(\boldsymbol{\pi}_{m-1}^r, \boldsymbol{\pi}^\ell), \qquad (13)$$

$$2\mathcal{V}_{m-1}(\boldsymbol{\pi}^{\star,r}, \boldsymbol{\pi}^r) \leq 5K^2 + \gamma_{m-1}\widehat{\mathrm{Reg}}_{m-1}(\boldsymbol{\pi}_{m-1}^\ell, \boldsymbol{\pi}^{\star,r}) + \gamma_{m-1}\widehat{\mathrm{Reg}}_{m-1}(\boldsymbol{\pi}_{m-1}^r, \boldsymbol{\pi}^r); \qquad (14)$$

Now note that Eq. (13) further implies:

$$\frac{2\mathcal{V}_{m-1}(\boldsymbol{\pi}^{\star,\ell}, \boldsymbol{\pi}^\ell)}{6\gamma_m} \leq \frac{5K^2 + \gamma_{m-1}\widehat{\mathrm{Reg}}_{m-1}(\boldsymbol{\pi}_{m-1}^\ell, \boldsymbol{\pi}^{\star,\ell}) + \gamma_{m-1}\widehat{\mathrm{Reg}}_{m-1}(\boldsymbol{\pi}_{m-1}^r, \boldsymbol{\pi}^\ell)}{6\gamma_m}$$

$$\leq \frac{5K^2 + \gamma_{m-1}\widehat{\mathrm{Reg}}_{m-1}(\boldsymbol{\psi}^\star[\boldsymbol{\pi}^{\star,\ell}, \widehat{\mathbf{M}}_{m-1}], \boldsymbol{\pi}^{\star,\ell}) + \gamma_{m-1}\widehat{\mathrm{Reg}}_{m-1}(\boldsymbol{\psi}^\star[\boldsymbol{\pi}^\ell, \widehat{\mathbf{M}}_{m-1}], \boldsymbol{\pi}^\ell)}{6\gamma_m}$$

$$\text{(by definition of } \boldsymbol{\psi}^\star[\boldsymbol{\pi}^{\star,\ell}, \widehat{\mathbf{M}}_{m-1}], \boldsymbol{\psi}^\star[\boldsymbol{\pi}^\ell, \widehat{\mathbf{M}}_{m-1}])$$

$$= \frac{5K^2 + \gamma_{m-1}\widehat{\mathrm{BReg}}_{m-1}(\boldsymbol{\pi}^{\star,\ell}) + \gamma_{m-1}\widehat{\mathrm{BReg}}_{m-1}(\boldsymbol{\pi}^\ell)}{6\gamma_m}$$

Similarly one can show that:

$$\frac{2\mathcal{V}_{m-1}(\boldsymbol{\pi}^{\star,r}, \boldsymbol{\pi}^r)}{6\gamma_m} \leq \frac{5K^2 + \gamma_{m-1}\widehat{\mathrm{Reg}}_{m-1}(\boldsymbol{\pi}_{m-1}^\ell, \boldsymbol{\pi}^{\star,r}) + \gamma_{m-1}\widehat{\mathrm{Reg}}_{m-1}(\boldsymbol{\pi}_{m-1}^r, \boldsymbol{\pi}^r)}{6\gamma_m}$$

$$\leq \frac{5K^2 + \gamma_{m-1}\widehat{\mathrm{Reg}}_{m-1}(\boldsymbol{\psi}^\star[\boldsymbol{\pi}^{\star,r}, \widehat{\mathbf{M}}_{m-1}], \boldsymbol{\pi}^{\star,r}) + \gamma_{m-1}\widehat{\mathrm{Reg}}_{m-1}(\boldsymbol{\psi}^\star[\boldsymbol{\pi}^r, \widehat{\mathbf{M}}_{m-1}], \boldsymbol{\pi}^r)}{6\gamma_m}$$

$$\text{(by definition of } \boldsymbol{\psi}^\star[\boldsymbol{\pi}^{\star,r}, \widehat{\mathbf{M}}_{m-1}], \boldsymbol{\psi}^\star[\boldsymbol{\pi}^r, \widehat{\mathbf{M}}_{m-1}])$$

$$= \frac{5K^2 + \gamma_{m-1}\widehat{\mathrm{BReg}}_{m-1}(\boldsymbol{\pi}^{\star,r}) + \gamma_{m-1}\widehat{\mathrm{BReg}}_{m-1}(\boldsymbol{\pi}^r)}{6\gamma_m}$$

Summing the results from the above two inequalities:

$$\frac{2\mathcal{V}_{m-1}(\boldsymbol{\pi}^{\star,\ell}, \boldsymbol{\pi}^{\ell}) + 2\mathcal{V}_{m-1}(\boldsymbol{\pi}^{\star,r}, \boldsymbol{\pi}^{r})}{6\gamma_m}$$

$$\leq \frac{10K^2 + \gamma_{m-1}\big(\widehat{\mathrm{BReg}}_{m-1}(\boldsymbol{\pi}^{\star,\ell}) + \widehat{\mathrm{BReg}}_{m-1}(\boldsymbol{\pi}^{\ell}) + \widehat{\mathrm{BReg}}_{m-1}(\boldsymbol{\pi}^{\star,r}) + \widehat{\mathrm{BReg}}_{m-1}(\boldsymbol{\pi}^{r})\big)}{6\gamma_m}$$

$$\overset{(a)}{\leq} \frac{2(5K^2 + c_0 K^2) + 2\gamma_{m-1}\big(\mathrm{BReg}(\boldsymbol{\pi}^{\star,\ell}) + \mathrm{BReg}(\boldsymbol{\pi}^{\star,r}) + \mathrm{BReg}(\boldsymbol{\pi}^{\ell}) + \mathrm{BReg}(\boldsymbol{\pi}^{r})\big)}{6\gamma_m}$$

$$\overset{(b)}{\leq} \frac{2(5K^2 + c_0 K^2) + 2\gamma_{m-1}\big(\mathrm{BReg}(\boldsymbol{\pi}^{\ell}) + \mathrm{BReg}(\boldsymbol{\pi}^{r})\big)}{6\gamma_m}$$

$$\leq \frac{2K^2(5 + c_0)}{6\gamma_m} + \frac{2\big(\mathrm{BReg}(\boldsymbol{\pi}^{\ell}) + \mathrm{BReg}(\boldsymbol{\pi}^{r})\big)}{6}$$

where $(a)$ follows from Eq. (10), and $(b)$ follows from the fact that $\mathrm{BReg}(\boldsymbol{\pi}^{\star,\ell}) + \mathrm{BReg}(\boldsymbol{\pi}^{\star,r}) = 0$ by Assumption 3. Combining the above two inequalities with (12):

$$\mathrm{BReg}(\boldsymbol{\pi}^{\ell}) + \mathrm{BReg}(\boldsymbol{\pi}^{r}) - [\widehat{\mathrm{BReg}}_m(\boldsymbol{\pi}^{\ell}) + \widehat{\mathrm{BReg}}_m(\boldsymbol{\pi}^{r})] \leq \frac{\mathcal{V}_{m-1}(\boldsymbol{\pi}^{\star,\ell}, \boldsymbol{\pi}^{\ell})}{3\gamma_m} + \frac{\mathcal{V}_{m-1}(\boldsymbol{\pi}^{\star,r}, \boldsymbol{\pi}^{r})}{3\gamma_m} + \frac{3K^2}{2\gamma_m}$$

$$\leq \frac{K^2(5 + c_0)}{3\gamma_m} + \frac{\big(\mathrm{BReg}(\boldsymbol{\pi}^{\ell}) + \mathrm{BReg}(\boldsymbol{\pi}^{r})\big)}{3} + \frac{3K^2}{2\gamma_m},$$

which implies

$$\frac{2}{3}[\mathrm{BReg}(\boldsymbol{\pi}^{\ell}) + \mathrm{BReg}(\boldsymbol{\pi}^{r})] - [\widehat{\mathrm{BReg}}_m(\boldsymbol{\pi}^{\ell}) + \widehat{\mathrm{BReg}}_m(\boldsymbol{\pi}^{r})] \leq \frac{K^2(5 + c_0)}{3\gamma_m} + \frac{3K^2}{2\gamma_m}.$$

Consequently, we get:

$$\mathrm{BReg}(\boldsymbol{\pi}^{\ell}) + \mathrm{BReg}(\boldsymbol{\pi}^{r}) \leq \frac{3}{2}[\widehat{\mathrm{BReg}}_m(\boldsymbol{\pi}^{\ell}) + \widehat{\mathrm{BReg}}_m(\boldsymbol{\pi}^{r})] + \frac{K^2(5 + c_0)}{2\gamma_m} + \frac{9K^2}{4\gamma_m},$$

finally leading to:

$$\mathrm{BReg}(\boldsymbol{\pi}^{\ell}) + \mathrm{BReg}(\boldsymbol{\pi}^{r}) \leq 2[\widehat{\mathrm{BReg}}_m(\boldsymbol{\pi}^{\ell}) + \widehat{\mathrm{BReg}}_m(\boldsymbol{\pi}^{r})] + \frac{c_0 K^2}{\gamma_m}, \tag{15}$$

for the choice of $c_0 = 11$ and recall that we assumed that $\boldsymbol{\pi}$ is any arbitrary policy. This concludes the first part of the proof.

The second part of the claim can be proved almost following the similar tricks. We add the analysis below for completeness.

**Part-2:** To prove the second part we start by noting that we can show:

$$\widehat{\mathrm{BReg}}(\boldsymbol{\pi}^{\ell}) + \widehat{\mathrm{BReg}}(\boldsymbol{\pi}^{r}) - [\mathrm{BReg}_m(\boldsymbol{\pi}^{\ell}) + \mathrm{BReg}_m(\boldsymbol{\pi}^{r})] \leq \frac{\mathcal{V}_{m-1}(\boldsymbol{\pi}_m^{\star,\ell}, \boldsymbol{\pi}^{\ell})}{3\gamma_m} + \frac{\mathcal{V}_{m-1}(\boldsymbol{\pi}_m^{\star,r}, \boldsymbol{\pi}^{r})}{3\gamma_m} + \frac{3K^2}{2\gamma_m}. \tag{16}$$

Following a similar analysis from above, we can obtain:

$$\frac{2\mathcal{V}_{m-1}(\boldsymbol{\pi}_m^{\star,\ell}, \boldsymbol{\pi}^{\ell}) + 2\mathcal{V}_{m-1}(\boldsymbol{\pi}_m^{\star,r}, \boldsymbol{\pi}^{r})}{6\gamma_m}$$

$$\leq \frac{10K^2 + \gamma_{m-1}\big(\widehat{\mathrm{BReg}}_{m-1}(\boldsymbol{\pi}_m^{\star,\ell}) + \widehat{\mathrm{BReg}}_{m-1}(\boldsymbol{\pi}^\ell) + \widehat{\mathrm{BReg}}_{m-1}(\boldsymbol{\pi}_m^{\star,r}) + \widehat{\mathrm{BReg}}_{m-1}(\boldsymbol{\pi}^r)\big)}{6\gamma_m}$$

$$\overset{(a)}{\leq} \frac{2(5K^2 + c_0 K^2) + 2\gamma_{m-1}\big([\mathrm{BReg}(\boldsymbol{\pi}_m^{\star,\ell}) + \mathrm{BReg}(\boldsymbol{\pi}_m^{\star,r})] + [\mathrm{BReg}(\boldsymbol{\pi}^\ell) + \mathrm{BReg}(\boldsymbol{\pi}^r)]\big)}{6\gamma_m}$$

$$\overset{(b)}{\leq} \frac{2(5K^2 + 2c_0 K^2) + 2\gamma_{m-1}\big(2[\widehat{\mathrm{BReg}}_m(\boldsymbol{\pi}_m^{\star,\ell}) + \widehat{\mathrm{BReg}}_m(\boldsymbol{\pi}_m^{\star,r})] + [\mathrm{BReg}(\boldsymbol{\pi}^\ell) + \mathrm{BReg}(\boldsymbol{\pi}^r)]\big)}{6\gamma_m}$$

$$\overset{(c)}{\leq} \frac{2(5K^2 + 2c_0 K^2) + 2\gamma_{m-1}\big(\mathrm{BReg}(\boldsymbol{\pi}^\ell) + \mathrm{BReg}(\boldsymbol{\pi}^r)\big)}{6\gamma_m}$$

$$\leq \frac{2K^2(5 + 2c_0)}{6\gamma_m} + \frac{2\big(\mathrm{BReg}(\boldsymbol{\pi}^\ell) + \mathrm{BReg}(\boldsymbol{\pi}^r)\big)}{6}$$

where $(a)$ follows from Eq. (11), $(b)$ follows since we have already proved the first part of the induction for epoch $m$ as concluded in Eq. (15), and $(c)$ follows from the fact that $\widehat{\mathrm{BReg}}_m(\boldsymbol{\pi}_m^{\star,\ell}) + \widehat{\mathrm{BReg}}_m(\boldsymbol{\pi}_m^{\star,r}) = 0$ by Assumption 3. Combining this with (16):

$$\widehat{\mathrm{BReg}}_m(\boldsymbol{\pi}^\ell) + \widehat{\mathrm{BReg}}_m(\boldsymbol{\pi}^r) - [\mathrm{BReg}(\boldsymbol{\pi}^\ell) + \mathrm{BReg}(\boldsymbol{\pi}^r)] \leq \frac{\mathcal{V}_{m-1}(\boldsymbol{\pi}_m^{\star,\ell}, \boldsymbol{\pi}^\ell)}{3\gamma_m} + \frac{\mathcal{V}_{m-1}(\boldsymbol{\pi}_m^{\star,r}, \boldsymbol{\pi}^r)}{3\gamma_m} + \frac{3K^2}{2\gamma_m}$$

$$\leq \frac{K^2(5 + 2c_0)}{3\gamma_m} + \frac{\big(\mathrm{BReg}(\boldsymbol{\pi}^\ell) + \mathrm{BReg}(\boldsymbol{\pi}^r)\big)}{3} + \frac{3K^2}{2\gamma_m},$$

which implies

$$\widehat{\mathrm{BReg}}_m(\boldsymbol{\pi}^\ell) + \widehat{\mathrm{BReg}}_m(\boldsymbol{\pi}^r) - \frac{4}{3}[\mathrm{BReg}(\boldsymbol{\pi}^\ell) + \mathrm{BReg}(\boldsymbol{\pi}^r)] \leq \frac{K^2(5 + 2c_0)}{3\gamma_m} + \frac{3K^2}{2\gamma_m}.$$

Consequently we have:

$$\widehat{\mathrm{BReg}}_m(\boldsymbol{\pi}^\ell) + \widehat{\mathrm{BReg}}_m(\boldsymbol{\pi}^r) \leq \frac{4}{3}[\mathrm{BReg}(\boldsymbol{\pi}^\ell) + \mathrm{BReg}(\boldsymbol{\pi}^r)] + \frac{K^2(5 + 2c_0)}{3\gamma_m} + \frac{3K^2}{2\gamma_m},$$

finally leading to:

$$\widehat{\mathrm{BReg}}_m(\boldsymbol{\pi}^\ell) + \widehat{\mathrm{BReg}}_m(\boldsymbol{\pi}^r) \leq 2[\widehat{\mathrm{BReg}}_m(\boldsymbol{\pi}^\ell) + \widehat{\mathrm{BReg}}_m(\boldsymbol{\pi}^r)] + \frac{c_0 K^2}{\gamma_m},$$

which is again satisfied for any choice of $c_0 = 11$, concluding the entire proof. $\qquad\square$

### C.4 Proof of Theorem 5

Given the result of Lemma 10, we are now ready to prove the final regret bound of Theorem 5:

**Theorem 5** (Regret Analysis of Algorithm 1). *Consider any function class $\mathcal{M}$ that satisfies Assumption 3. Then under realizability (Assumption 1), and with an epoch schedule $\tau_1, \ldots, \tau_m$ such that $\tau_m \geq 2^m$ for $m \leq \log T$, with probability at least $(1 - \delta)$ for any $\delta \in (0, 1)$ the best-response regret* $(\mathrm{BR\text{-}Reg}_T)$ *of* **Double-Monster** *after $T$ rounds is bounded by:*

$$O\left(K \sum_{m=2}^{m(T)} \sqrt{\mathcal{E}_{\mathrm{off},\mathcal{M}}(\delta/(2m^2), \tau_{m-1} - \tau_{m-2})(\tau_m - \tau_{m-1})}\right) \tag{5}$$

*Proof of Theorem 5.* Applying Lemma 10 over the phases $m = 1, \ldots, m(T)$ we get:

$$\mathrm{BR\text{-}Reg}_T = \frac{1}{2} \sum_{t=1}^{T} \mathbf{E}_{x_t \sim \mathcal{D}_{\mathcal{X}}}\left[\max_{\mathbf{q} \in \Delta_K}\left[\mathbb{E}_{a \sim \mathbf{q}}\left[\mathbb{E}_{(a_t, b_t) \sim \mathbf{p}_t}[\mathbf{M}^\star(x_t)[a, a_t] + \mathbf{M}^\star(x_t)[a, b_t]]\right]\right]\right]$$

$$= \frac{1}{2} \sum_{m=1}^{m(T)} \left[ \widehat{\mathrm{BReg}}_m(\boldsymbol{\pi}_m^\ell) + \widehat{\mathrm{BReg}}_m(\boldsymbol{\pi}_m^r) \right] (\tau_m - \tau_{m-1})$$

$$\overset{(a)}{\leq} \frac{1}{2} \sum_{m=1}^{m(T)} \left[ 2\mathrm{BReg}_m(\boldsymbol{\pi}_m^\ell) + 2\mathrm{BReg}_m(\boldsymbol{\pi}_m^r) + \frac{11K^2}{\gamma_m} \right] (\tau_m - \tau_{m-1})$$

$$\overset{(b)}{\leq} \frac{1}{2} \sum_{m=1}^{m(T)} \left[ \frac{21K^2}{\gamma_m} \right] (\tau_m - \tau_{m-1})$$

$$\leq 11K \sum_{m=2}^{m(T)} \sqrt[2]{\mathcal{E}_{\mathcal{F}, \delta/(2m^2)}(\tau_{m-1} - \tau_{m-2})} + \tau_1$$

where $(a)$ follows from Lemma 10, $(b)$ follows from Lemma 7 and the last inequality follows by the choice of $\gamma_m$ in Double-Monster (Algorithm 1). The proof is concluded noting $\tau_1 = O(1)$. $\qquad\square$

### C.5   Proof of Corollary 6 for Finite $\mathcal{M}$

*Proof.* In particular for the special case when $\mathcal{M}$ is finite, we know it is possible to design offline regression oracles (see [24, Table 1]) such that the offline loss that for any phase $m$ could be bounded by *by*

$$\mathcal{E}_{\mathcal{F}, \delta/(2m^2)}(\tau_{m-1} - \tau_{m-2}) \leq \frac{2\log(2|\mathcal{M}|\log(\tau_{m-1})m/\delta)}{\tau_{m-1}}$$

as follows from [32]. Now using the derivation of Theorem 5, we can specifically derive for $\delta = 1/T$:

$$\mathrm{BR\text{-}Reg}_T = \frac{1}{2} \sum_{t=1}^{T} \mathbf{E}_{x_t \sim \mathcal{D}_\mathcal{X}} \left[ \max_{\mathbf{q} \in \Delta_K} \left[ \mathbb{E}_{a \sim \mathbf{q}} \left[ \mathbb{E}_{(a_t, b_t) \sim \mathbf{p}_t} \left[ \mathbf{M}^\star(x_t)[a, a_t] + \mathbf{M}^\star(x_t)[a, b_t] \right] \right] \right] \right]$$

$$= \frac{1}{2} \sum_{m=1}^{m(T)} \left[ \mathrm{BReg}(\boldsymbol{\pi}_m^\ell) + \mathrm{BReg}(\boldsymbol{\pi}_m^r) \right] (\tau_m - \tau_{m-1})$$

$$\overset{(a)}{\leq} \frac{1}{2} \sum_{m=1}^{m(T)} \left[ 2\widehat{\mathrm{BReg}}_m(\boldsymbol{\pi}_m^\ell) + 2\widehat{\mathrm{BReg}}_m(\boldsymbol{\pi}_m^r) + \frac{11K^2}{\gamma_m} \right] (\tau_m - \tau_{m-1})$$

$$\overset{(b)}{\leq} \frac{1}{2} \sum_{m=1}^{m(T)} \left[ \frac{21K^2}{\gamma_m} \right] (\tau_m - \tau_{m-1})$$

$$\leq 11K \sum_{m=2}^{m(T)} \sqrt[2]{\mathcal{E}_{\mathcal{F}, \delta/(2m^2)}(\tau_{m-1} - \tau_{m-2})} + \tau_1$$

$$= 22K \sum_{m=2}^{m(T)} \sqrt[2]{\frac{\log(2|\mathcal{M}|\log(\tau_{m-1})T)}{\tau_{m-1}}} (\tau_m - \tau_{m-1}) + \tau_1 = O(K\sqrt{T\log(2|\mathcal{M}|T\log T)}).$$

$\qquad\square$

## D   Supplementary for Section 6

### D.1   Regret Analysis of Double-Monster-Inf: Proof Analysis of Theorem 11

As discussed in the algorithm description of Section 6, one of the key contributions of this work lies in the analysis of our proposed method which is the first attempt to address the contextual dueling bandit problem with offline regression. The overall proof structure adapts the same line of argument given in Section 5.2, however, the challenge was to reproduce the equivalent intermediate lemmas (precisely the results of Lemma 8, Lemma 7 and Lemma 10) for the continuous action space. We will discuss the proof of Theorem 11 in detail in the rest of this section. We will find it useful to define some notations for the ease of exposition.

### D.1.1 Additional Concepts towards proving Theorem 11

We will start with defining an important concept of Epsilon-net of $\mathcal{K}$. We will later see the usefulness of this concept while proving Lemma 15 as explained in the associated Remark 17.

**Definition 7** (Epsilon-Net of $\mathcal{K}$ (w.r.t Euclidean Norm)). *A (finite or countable) set $N_\varepsilon \subseteq \kappa$ is called an $\varepsilon$–net of $\mathcal{K}$ with respect to the Euclidean norm if*

$$\forall\, a \in \mathcal{K} \quad \exists\, \mathbf{b} \in N_\varepsilon \text{ such that } \|\mathbf{a} - \mathbf{b}\|_2 \leq \varepsilon.$$

Equivalently,

$$\mathcal{K} \subseteq \bigcup_{\mathbf{a} \in N_\varepsilon} B_2(\mathbf{a}, \varepsilon), \qquad \text{where } B_2(\mathbf{a}, \varepsilon) := \left\{\mathbf{b} \in \mathbb{R}^d \mid \|\mathbf{a} - \mathbf{b}\|_2 \leq \varepsilon \right\}.$$

**Definition 8** (Projection to $\varepsilon$-net). *Given any point $\mathbf{a} \in \mathcal{K}$, we denote the projection of $\mathbf{a}$ to its $\varepsilon$-net as $\mathcal{P}_{N_\varepsilon}(\mathbf{a}) := \arg\min_{\mathbf{b} \in N_\varepsilon} \|\mathbf{a} - \mathbf{b}\|_2$, where ties are broken arbitrarily.*

Note for any $\mathbf{a} \in \mathcal{K}$, $\|\mathbf{a} - \mathcal{P}_{N_\varepsilon}(\mathbf{a})\|_2 \leq \varepsilon$.

**Remark 14.** $\varepsilon \in (0, 1)$ *is a tunable parameter, and we will see in the proof of Theorem 11 below how to choose it optimally.*

**Definition 9** (Policy Class over $\mathcal{K}$). *A standard policy class $\boldsymbol{\Psi} := \{\psi \mid \psi : \mathcal{X} \mapsto \Delta(\mathcal{K})\}$ is a set containing all the set of all probability measures on $\mathcal{K}$. Further we also define by policy class $\boldsymbol{\Psi}^2 := \{\psi^2 \mid \psi^2 : \mathcal{X} \mapsto \Delta(\mathcal{K} \times \mathcal{K})\}$ be the set of all probability measures on $\mathcal{K} \times \mathcal{K}$.*

**Definition 10** (Policy Class over Epsilon-Net of $\mathcal{K}$). *A standard policy class $\Pi := \{\boldsymbol{\pi} \mid \boldsymbol{\pi} : \mathcal{X} \mapsto \Delta(N_\varepsilon)\}$ is a set containing all the probability distributions over $N_\varepsilon$. Further we also define by policy class $\Pi^2 := \{\boldsymbol{\pi}^2 \mid \boldsymbol{\pi}^2 : \mathcal{X} \mapsto \Delta(N_\varepsilon \times N_\varepsilon)\}$ be the set of all probability measures on $N_\varepsilon \times N_\varepsilon$.*

**Definition 11** (Decision policy of Epoch-$m$). *At any epoch $m$ and $x \in \mathcal{X}$, the decision policy of epoch $m$ is defined as $\boldsymbol{\psi}_m \in \boldsymbol{\Psi}^2$: $\boldsymbol{\psi}_m(x) \leftarrow \texttt{Cont-CvxConstraint-Solver}(x, \phi_m, \gamma_m), \forall x \in \mathcal{X}$.*

Note that at any time $t$ in epoch $m$, $\boldsymbol{\psi}_m(x_t) = \mathbf{p}_t \in \Delta(\mathcal{K} \times \mathcal{K})$ is a probability measure over the product space $\mathcal{K} \times \mathcal{K}$. Similar to Section 5, will further denote by $\boldsymbol{\psi}_m^\ell \in \boldsymbol{\Psi}$ and $\boldsymbol{\psi}_m^r \in \boldsymbol{\Psi}$ respectively the left and right marginal policies of $\boldsymbol{\pi}_m \in \boldsymbol{\Psi}^2$. i.e.:

$$\boldsymbol{\psi}_m^\ell(\mathbf{a} \mid x) = \mathbf{E}_{x \sim \mathcal{D}_\mathcal{X}}\left[\int_\mathcal{K} \boldsymbol{\psi}_m(\mathbf{a}, \mathbf{b} \mid x)d\mathbf{b}\right] \text{ and } \boldsymbol{\psi}_m^r(\mathbf{b} \mid x) = \mathbf{E}_{x \sim \mathcal{D}_\mathcal{X}}\left[\int_\mathcal{K} \boldsymbol{\psi}_m(\mathbf{a}, \mathbf{b} \mid x)d\mathbf{a}\right].$$

**Definition 12** (Approximate Decision policy of Epoch-$m$). *At any epoch $m$ and $x \in \mathcal{X}$, the decision policy of epoch $m$ is defined as $\boldsymbol{\pi}_m \in \Pi^2$, such that:*

$$\boldsymbol{\pi}_m(x) = \text{argmax}_{\mathbf{q} \in \Delta(N_\varepsilon \times \varepsilon)}\left[\langle \mathbf{ab_q}, \phi(x)\rangle + \tfrac{1}{\gamma}\log\det(\mathbf{H_q})\right]$$

Similarly we define $\boldsymbol{\pi}_m^\ell \in \Pi$ and $\boldsymbol{\pi}_m^r \in \Pi$ respectively to be the left and right marginal of $\boldsymbol{\pi}_m \in \Pi^2$.

**Definition 13** ($\phi$-Best-Response Policy). *Given any scoring function $\phi \in \Phi$, we define the $\phi$-best response policy $\boldsymbol{\psi}_\phi : \mathcal{X} \mapsto \mathcal{K}$ as:*

$$\boldsymbol{\psi}_\phi(x) := \arg\max_{\mathbf{a} \in \mathcal{K}}\langle \phi(x), \mathbf{a}\rangle, \quad \forall x \in \mathcal{X}$$

*which denotes the* best-response (a.k.a. highest-scoring) decision/action *for the scoring function $\phi$.*

**Definition 14** (Instantaneous Regret of a Policy). *For any arbitrary underlying preference mapping $\phi \in \Phi$, we denote the best-response regret of policy $\psi \in \boldsymbol{\Psi}$ as:*

$$\overline{\text{BReg}}_\phi(\boldsymbol{\psi}) := \mathbf{E}_{x \sim \mathcal{D}_\mathcal{X}}[\langle \boldsymbol{\psi}_\phi(x), \phi(x)\rangle - \langle \boldsymbol{\psi}(x), \phi(x)\rangle],$$

*where $\boldsymbol{\psi}_\phi(x) \in \mathcal{K}$ is as defined in Definition 13, and with a slight abuse of notation $\langle \boldsymbol{\psi}(x), \phi(x)\rangle := \mathbf{E}_{\mathbf{a} \sim \boldsymbol{\psi}(x)}[\langle \mathbf{a}, \phi(x)\rangle]$.*

Given the above definition, for the true scoring function $\phi^*$, we will denote $\overline{\text{BReg}}_{\phi^*}(\boldsymbol{\psi})$ simply by $\overline{\text{BReg}}(\boldsymbol{\psi})$.

**Definition 15** (Approximate $\phi$-Best-Response Policy). *Given any scoring function $\phi \in \Phi$, we define the approximate $\phi$-best response policy $\boldsymbol{\pi}_\phi : \mathcal{X} \mapsto N_\varepsilon$ as:*

$$\boldsymbol{\pi}_\phi(x) := \arg\max_{\mathbf{a} \in N_\varepsilon} \langle \phi(x), \mathbf{a} \rangle, \quad \forall x \in \mathcal{X}$$

*which denotes the* best-response (a.k.a. highest-scoring) decision/action *for the scoring function $\phi$ within $N_\varepsilon$, the $\varepsilon$-net of $\mathcal{K}$.*

**Definition 16** (Approximate Instantaneous Regret of a Policy). *For any arbitrary underlying preference mapping $\phi \in \Phi$, we denote the approximate best-response regret of policy $\boldsymbol{\pi} \in \Pi$ as:*

$$\mathrm{BReg}_\phi(\boldsymbol{\pi}) := \mathbf{E}_{x \sim \mathcal{D}_\mathcal{X}} [\langle \boldsymbol{\pi}_\phi(x), \phi(x) \rangle - \langle \boldsymbol{\pi}(x), \phi(x) \rangle],$$

*where $\boldsymbol{\pi}_\phi(x) \in N_\varepsilon$ is as defined in Definition 13, and again with a slight abuse of notation, we denote by $\langle \boldsymbol{\pi}(x), \phi(x) \rangle := \mathbf{E}_{\mathbf{a} \sim \boldsymbol{\pi}(x)}[\langle \mathbf{a}, \phi(x) \rangle]$.*

For the true scoring function $\phi^*$, we will denote $\mathrm{BReg}_{\phi^*}(\boldsymbol{\pi})$ simply by $\mathrm{BReg}(\boldsymbol{\pi})$, and for $\phi = \phi_m$, the estimated feature mapping at epoch $m$ (see Algorithm 2), we will denote $\mathrm{BReg}_{\phi_m}(\boldsymbol{\pi})$ by $\widehat{\mathrm{BReg}}_m(\boldsymbol{\pi})$. Further, we will also define the decision variance of any policy $\boldsymbol{\pi} \in \Pi$ defined as:

**Definition 17** (Decision variance of a policy). *For any two policies $\boldsymbol{\pi}_1, \boldsymbol{\pi}_2 \in \Pi$ the decision variance of $\boldsymbol{\pi}_1, \boldsymbol{\pi}_2$ with respect to another decision policy $\tilde{\boldsymbol{\pi}} \in \Pi^2$ is defined as:*

$$\mathcal{V}_{\tilde{\boldsymbol{\pi}}}(\boldsymbol{\pi}_1, \boldsymbol{\pi}_2) := \mathbf{E}_{x \sim \mathcal{D}_\mathcal{X}} \left[ \|\boldsymbol{\pi}_1(x) - \boldsymbol{\pi}_2(x)\|_{\mathbf{H}(\tilde{\boldsymbol{\pi}}(x))^{-1}} \right].$$

In particular, when $\tilde{\boldsymbol{\pi}} = \boldsymbol{\pi}_m$, where recall that $\boldsymbol{\pi}_m$ is the decision policy of epoch $m$, we will use the shorthand $\mathcal{V}_m(\boldsymbol{\pi}_1, \boldsymbol{\pi}_2)$ to denote $\mathcal{V}_{\tilde{\boldsymbol{\pi}}}(\boldsymbol{\pi}_1, \boldsymbol{\pi}_2)$. To understand the term from an intuitive level, roughly $\mathcal{V}_{\tilde{\boldsymbol{\pi}}}(\boldsymbol{\pi}_1, \boldsymbol{\pi}_2)$ indicates the selection variance of policy $\tilde{\boldsymbol{\pi}}$ on the random pairs of actions generated by the pair of policies $(\boldsymbol{\pi}_1, \boldsymbol{\pi}_2)$.

### D.1.2 Proof of Main Theorem 11

Given the above definitions and setting the preliminaries, we are now ready to explain the proof Theorem 11. Let us start with recalling Theorem 11 first:

**Theorem 11** (Regret Analysis of Algorithm 2). *Assume $\forall x \in \mathcal{X}$, $\|\phi^*(x)\| \leq D$ for some $D > 0$, and suppose we choose $c = \frac{5d}{64D^2\gamma_{m(T)}}$. Then under realizability (Assumption 1) of the function class $\Phi$, and with an epoch schedule $\tau_1, \ldots, \tau_m$ such that $\tau_m \geq 2^m$ for $m \leq \log T$, with probability at least $(1 - \delta)$ for any $\delta \in (0, 1)$ the best-response regret of Algorithm 2, $\mathrm{BR\text{-}Reg}_T^{(cont)}$, in $T$ rounds $\mathcal{K}$ is bounded by: $O\left( \sum_{m=2}^{m(T)} \sqrt{d\mathcal{E}_{\mathrm{off},\Phi^*}(\delta/(2m^2), \tau_{m-1} - \tau_{m-2})(\tau_m - \tau_{m-1})} \right).$*

*Proof of Theorem 11.* We start with a generalization of Lemma 7 for continuous decision spaces which establish the empirical performance of Algorithm 2, as given below:

**Lemma 15** (Properties of decision policy $\boldsymbol{\pi}_m$ in Algorithm 2). *At any epoch $m$, the decision policy of epoch $m$, $\boldsymbol{\pi}_m$, satisfies:*

$$\widehat{\mathrm{BReg}}_m(\boldsymbol{\pi}_m^\ell) + \widehat{\mathrm{BReg}}_m(\boldsymbol{\pi}_m^r) \leq \frac{d}{\gamma_m},$$

$$\mathcal{V}_m(\boldsymbol{\pi}, \boldsymbol{\pi}') \leq d + \gamma_m \big( \widehat{\mathrm{BReg}}_m(\boldsymbol{\pi}) + \widehat{\mathrm{BReg}}_m(\boldsymbol{\pi}') \big),$$

*for any two policies $\boldsymbol{\pi}$ and $\boldsymbol{\pi}' \in \Pi$.*

The next key step is to connect the empirical performance of any policy $\boldsymbol{\pi}$ to its true performance. Our next result carries a similar spirit of Lemma 8, however the proof analysis if very different as follows from our detailed proof analyses in Appendix D.2.

**Lemma 16** (Emp vs True Performance). *Let $\boldsymbol{\pi}$ be any policy in $\Pi^2$. Then for all epochs $m \in \mathbb{N}_+$:*

$$\mathrm{BReg}(\boldsymbol{\pi}^\ell) + \mathrm{BReg}(\boldsymbol{\pi}^r) \leq 2\widehat{\mathrm{BReg}}_m(\boldsymbol{\pi}^\ell) + 2\widehat{\mathrm{BReg}}_m(\boldsymbol{\pi}^r) + 6d/\gamma_m,$$

$$\widehat{\mathrm{BReg}}_m(\boldsymbol{\pi}^\ell) + \widehat{\mathrm{BReg}}_m(\boldsymbol{\pi}^r) \leq 2\mathrm{BReg}(\boldsymbol{\pi}^\ell) + 2\mathrm{BReg}(\boldsymbol{\pi}^r) + 6d/\gamma_m.$$

Now the final regret bound of Algorithm 2 now follows by combining the statements of Lemma 15 and Lemma 16, as shown given below:

Given the above results, the proof of Theorem 11 follows combining the results of Lemma 16 and Lemma 15. Applying Lemma 16 over the phases $m = 1, \ldots, m(T)$ we get:

$$
\text{BR-Reg}_T^{(cont)} = \sum_{t=1}^{T} \mathbf{E}_{x_t \sim \mathcal{D}_{\mathcal{X}}} \left[ \max_{\mathbf{a}^* \in \mathcal{K}} \langle \mathbf{a}^*, \phi^*(x_t) \rangle - \mathbf{E}_{(\mathbf{a}_t, \mathbf{b}_t) \sim p_t} \left[ \frac{\langle (\mathbf{a}_t + \mathbf{b}_t), \phi^*(x_t) \rangle}{2} \right] \right]
$$

$$
= \frac{1}{2} \sum_{m=1}^{m(T)} \left[ \overline{\text{BReg}}_m(\boldsymbol{\psi}_m^\ell) + \overline{\text{BReg}}_m(\boldsymbol{\psi}_m^r) \right] (\tau_m - \tau_{m-1})
$$

$$
= \frac{1}{2} \sum_{m=1}^{m(T)} \mathbf{E}_{x \sim \mathcal{D}} \left[ \langle \phi^*(x), \boldsymbol{\psi}_{\phi^*}(x) - \boldsymbol{\psi}_m^\ell(x) \rangle + \langle \phi^*(x), \boldsymbol{\psi}_{\phi^*}(x) - \boldsymbol{\psi}_m^r(x) \rangle \right] (\tau_m - \tau_{m-1})
$$

$$
\leq \frac{1}{2} \sum_{m=1}^{m(T)} \mathbf{E}_{x \sim \mathcal{D}} \left[ 2 \langle \phi^*(x), \boldsymbol{\psi}_{\phi^*}(x) \rangle - \langle \phi^*(x), \boldsymbol{\pi}_m^\ell(x) \rangle - \langle \phi^*(x), \boldsymbol{\pi}_m^r(x) \rangle \right] (\tau_m - \tau_{m-1})
$$

$$
\overset{(a)}{=} \frac{1}{2} \sum_{m=1}^{m(T)} \mathbf{E} \left[ 2 \langle \phi^*(x), \mathcal{P}_{N_\epsilon}(\boldsymbol{\psi}_{\phi^*}(x)) + \boldsymbol{\psi}_{\phi^*}(x) - \mathcal{P}_{N_\epsilon}(\boldsymbol{\psi}_{\phi^*}(x)) \rangle - \langle \phi^*(x), (\boldsymbol{\pi}_m^\ell(x) + \boldsymbol{\pi}_m^r(x)) \rangle \right] (\tau_m - \tau_{m-1})
$$

$$
\overset{(b)}{\leq} \frac{1}{2} \sum_{m=1}^{m(T)} \mathbf{E}_{x \sim \mathcal{D}} \left[ 2 \langle \phi^*(x), \mathcal{P}_{N_\epsilon}(\boldsymbol{\psi}_{\phi^*}(x)) \rangle + 2 \varepsilon D - \langle \phi^*(x), (\boldsymbol{\pi}_m^\ell(x) + \boldsymbol{\pi}_m^r(x)) \rangle \right] (\tau_m - \tau_{m-1})
$$

$$
\leq \frac{1}{2} \sum_{m=1}^{m(T)} \mathbf{E}_{x \sim \mathcal{D}} \left[ \max_{\mathbf{a} \in N_\varepsilon} 2 \langle \phi^*(x), \mathbf{a} \rangle - \langle \phi^*(x), (\boldsymbol{\pi}_m^\ell(x) + \boldsymbol{\pi}_m^r(x)) \rangle \right] (\tau_m - \tau_{m-1}) + \varepsilon D T
$$

$$
= \frac{1}{2} \sum_{m=1}^{m(T)} \mathbf{E}_{x \sim \mathcal{D}} \left[ 2 \langle \phi^*(x), \boldsymbol{\pi}_{\phi^*}(x) \rangle - \langle \phi^*(x), (\boldsymbol{\pi}_m^\ell(x) + \boldsymbol{\pi}_m^r(x)) \rangle \right] (\tau_m - \tau_{m-1}) + \varepsilon D T
$$

$$
= \frac{1}{2} \sum_{m=1}^{m(T)} \left[ \text{BReg}_m(\boldsymbol{\pi}_m^\ell) + \text{BReg}_m(\boldsymbol{\pi}_m^r) \right] (\tau_m - \tau_{m-1}) + \varepsilon D T
$$

$$
\overset{(c)}{\leq} \frac{1}{2} \sum_{m=1}^{m(T)} \left[ 2 \widehat{\text{BReg}}_m(\boldsymbol{\pi}_m^\ell) + 2 \widehat{\text{BReg}}_m(\boldsymbol{\pi}_m^r) + \frac{6d}{\gamma_m} \right] (\tau_m - \tau_{m-1}) + \varepsilon D T
$$

$$
\overset{(d)}{\leq} \frac{1}{2} \sum_{m=1}^{m(T)} \left[ \frac{8d}{\gamma_m} \right] (\tau_m - \tau_{m-1}) + \varepsilon D T
$$

$$
\leq 8 \sqrt{2d} \sum_{m=2}^{m(T)} \sqrt[2]{\mathcal{E}_{\Phi, \delta/(2m^2)}(\tau_{m-1} - \tau_{m-2})} + \tau_1 + \varepsilon D T
$$

where $(a)$ follows from Definition 8, $(b)$ applies Cauchy-Schwarz and the fact that $\|\phi^*(x)\| \leq D$, $\forall x \in \mathcal{X}$ by assumption, $(c)$ follows from Lemma 16, $(d)$ follows from Lemma 15 and the last inequality follows by the choice of $\gamma_m$ in `Double-Monster` (Algorithm 2). The proof is concluded noting $\tau_1 = 1$ (by parameter choice) and setting $\varepsilon = \frac{1}{DT}$.

This gives an overall roadmap of the proof of Theorem 11. We give the detailed proofs of the above lemmas in the following subsection. □

### D.2 Proof of Key Lemmas for Theorem 11

**Proof of Lemma 15**

**Lemma 15** (Properties of decision policy $\boldsymbol{\pi}_m$ in Algorithm 2). *At any epoch $m$, the decision policy of epoch $m$, $\boldsymbol{\pi}_m$, satisfies:*

$$\widehat{\mathrm{BReg}}_m(\boldsymbol{\pi}_m^\ell) + \widehat{\mathrm{BReg}}_m(\boldsymbol{\pi}_m^r) \leq \frac{d}{\gamma_m},$$

$$\mathcal{V}_m(\boldsymbol{\pi}, \boldsymbol{\pi}') \leq d + \gamma_m\big(\widehat{\mathrm{BReg}}_m(\boldsymbol{\pi}) + \widehat{\mathrm{BReg}}_m(\boldsymbol{\pi}')\big),$$

*for any two policies $\boldsymbol{\pi}$ and $\boldsymbol{\pi}' \in \Pi$.*

*Proof of Lemma 15.* Note that at any epoch $m$ and context $x \in \mathcal{X}$, we have:

$$\boldsymbol{\pi}_m(x) = \arg\min_{\mathbf{q} \in \Delta(N_\varepsilon \times N_\varepsilon)} \left[ \mathbf{E}_{(\mathbf{a},\mathbf{b}) \sim \mathbf{q}}\big[\langle \boldsymbol{\pi}_{\phi_m}(x), \phi_m(x)\rangle - \langle (\mathbf{a}+\mathbf{b})/2, \phi_m(x)\rangle\big] - \frac{1}{\gamma_m}\log\det(\mathbf{H_q}) \right]. \tag{17}$$

Fixing a $x \in \mathcal{X}$, let $\tilde{\mathbf{q}}^* \in \Delta(N_\varepsilon \times N_\varepsilon)$ be the optimal solution of the above optimization. Then consider the Lagrangian of the above optimization and setting the derivative with respect to $\tilde{q}^*(\mathbf{a}, \mathbf{b})$ to zero, we obtain logdet determinant:

$$\langle \boldsymbol{\pi}_{\phi_m}(x), \phi_m(x)\rangle - \langle(\mathbf{a}+\mathbf{b})/2, \phi_m(x)\rangle - \frac{1}{\gamma_m}\|\mathbf{a}-\mathbf{b}\|_{\mathbf{H}(\tilde{\mathbf{q}}^*)^{-1}}^2 - \lambda(\mathbf{a},\mathbf{b}) + \lambda = 0, \tag{18}$$

where $\lambda \in \mathbb{R}$ and $\lambda(\mathbf{a}, \mathbf{b}) \geq 0$, are the Lagrangian multipliers. The interesting thing to note is the third term in the above expression which we obtain using the known fact about the derivative of the determinant of any invertible (full rank) matrix $\mathbf{A}$ is given by

$$d(\det(\mathbf{A})) = \det(\mathbf{A})\,\mathrm{Tr}(\mathbf{A}^{-1}\,d\mathbf{A}),$$

where $Tr(\cdot)$ the trace function of the matrix. For any $\mathbf{q} \in \Delta(N_\varepsilon \times N_\varepsilon)$, using this we get:

$$\frac{d}{dq_{ab}}(\log\det(\mathbf{H_q})) = \frac{1}{\det(\mathbf{H_q})}\det(\mathbf{H_q})\,\mathrm{Tr}\left(\mathbf{H_q}^{-1}\frac{d}{dq_{ab}}\mathbf{H_q}\right) = \mathrm{Tr}\left(\mathbf{H_q}^{-1}(\mathbf{a}-\mathbf{b})(\mathbf{a}-\mathbf{b})^\top\right)$$

$$= \mathrm{Tr}\left((\mathbf{a}-\mathbf{b})^\top\mathbf{H_q}^{-1}(\mathbf{a}-\mathbf{b})\right) = \|\mathbf{a}-\mathbf{b}\|_{\mathbf{H}(\mathbf{q})^{-1}}^2,$$

where the last equality uses that the trace of a matrix product is invariant under cyclic permutations, i.e. $\mathrm{Tr}(\mathbf{ABC}) = \mathrm{Tr}(\mathbf{CAB}) = \mathrm{Tr}(\mathbf{BCA})$ given any $d \times d$ dimensional matrices $\mathbf{A}, \mathbf{B}, \mathbf{C}$.

Further, multiplying above by $\tilde{\mathbf{q}}^*(\mathbf{a}, \mathbf{b})$ and summing over all pairs $(\mathbf{a}, \mathbf{b})$, we get:

$$\sum_{\mathbf{a},\mathbf{b} \in (N_\varepsilon \times N_\varepsilon)} \tilde{q}^*(\mathbf{a},\mathbf{b})\left(\langle\boldsymbol{\pi}_{\phi_m}(x),\phi_m(x)\rangle - \langle(\mathbf{a}+\mathbf{b})/2,\phi_m(x)\rangle\right)$$

$$-\frac{1}{\gamma_m}\sum_{(\mathbf{a},\mathbf{b}) \in (N_\varepsilon \times N_\varepsilon)} \tilde{q}^*(\mathbf{a},\mathbf{b})\|\mathbf{a}-\mathbf{b}\|_{\mathbf{H}(\tilde{\mathbf{q}}^*)^{-1}}^2 - \sum_{(\mathbf{a},\mathbf{b}) \in (N_\varepsilon \times N_\varepsilon)} \tilde{q}^*(a,b)\lambda(\mathbf{a},\mathbf{b}) + \lambda = 0. \tag{19}$$

One important observation to note is

$$\sum_{(\mathbf{a},\mathbf{b}) \in (N_\varepsilon \times N_\varepsilon)} \tilde{\mathbf{q}}^*(\mathbf{a},\mathbf{b})\|\mathbf{a}-\mathbf{b}\|_{\mathbf{H}(\tilde{\mathbf{q}}^*)^{-1}}^2 = \sum_{(\mathbf{a},\mathbf{b}) \in (N_\varepsilon \times N_\varepsilon)} \tilde{\mathbf{q}}^*(\mathbf{a},\mathbf{b})(\mathbf{a}-\mathbf{b})^\top\mathbf{H}(\tilde{\mathbf{q}}^*)^{-1}(\mathbf{a}-\mathbf{b})$$

$$= \sum_{(\mathbf{a},\mathbf{b}) \in (N_\varepsilon \times N_\varepsilon)} \tilde{\mathbf{q}}^*(\mathbf{a},\mathbf{b})\,\mathrm{Tr}\left((\mathbf{a}-\mathbf{b})^\top\mathbf{H}(\tilde{\mathbf{q}}^*)^{-1}(\mathbf{a}-\mathbf{b})\right)$$

$$= \sum_{(\mathbf{a},\mathbf{b}) \in (N_\varepsilon \times N_\varepsilon)} \tilde{\mathbf{q}}^*(\mathbf{a},\mathbf{b})\,\mathrm{Tr}\left((\mathbf{a}-\mathbf{b})^\top\mathbf{H}(\tilde{\mathbf{q}}^*)^{-1}(\mathbf{a}-\mathbf{b})\right)$$

$$= \sum_{(\mathbf{a},\mathbf{b}) \in (N_\varepsilon \times N_\varepsilon)} \tilde{\mathbf{q}}^*(\mathbf{a},\mathbf{b})\,\mathrm{Tr}\left(\mathbf{H}(\tilde{\mathbf{q}}^*)^{-1}(\mathbf{a}-\mathbf{b})(\mathbf{a}-\mathbf{b})^\top\right)$$

$$= \text{Tr}\left(\sum_{(\mathbf{a},\mathbf{b})\in(N_\varepsilon\times N_\varepsilon)} \tilde{\mathbf{q}}^*(\mathbf{a},\mathbf{b})\mathbf{H}(\tilde{\mathbf{q}}^*)^{-1}(\mathbf{a}-\mathbf{b})(\mathbf{a}-\mathbf{b})^\top\right)$$

$$= \text{Tr}\left(\mathbf{H}(\tilde{\mathbf{q}}^*)^{-1}\sum_{(\mathbf{a},\mathbf{b})\in(N_\varepsilon\times N_\varepsilon)} \tilde{\mathbf{q}}^*(\mathbf{a},\mathbf{b})(\mathbf{a}-\mathbf{b})(\mathbf{a}-\mathbf{b})^\top\right) \leq d$$

where the third inequality again follows by applying the *cyclic permutation invariance property* of trace function as describe above. Further the last inequality can be justified through Lemma 19 and noting $\mathbf{H}(\tilde{\mathbf{q}}^*) = \sum_{(\mathbf{a},\mathbf{b})\in(N_\varepsilon\times N_\varepsilon)} \tilde{\mathbf{q}}^*(\mathbf{a},\mathbf{b})(\mathbf{a}-\mathbf{b})(\mathbf{a}-\mathbf{b})^\top + \lambda\mathbf{I}_d$.

Rearranging the terms in (19) and noting $\sum_{(\mathbf{a},\mathbf{b})\in(N_\varepsilon\times N_\varepsilon)} \tilde{\mathbf{q}}^*(\mathbf{a},\mathbf{b})\|\mathbf{a}-\mathbf{b}\|^2_{\mathbf{H}(\tilde{\mathbf{q}}^*)^{-1}} \leq d$, we have:

$$\sum_{(\mathbf{a},\mathbf{b})\in(N_\varepsilon\times N_\varepsilon)} \tilde{q}^*(\mathbf{a},\mathbf{b})\left(\langle\boldsymbol{\pi}_{\phi_m}(x),\phi_m(x)\rangle - \langle(\mathbf{a}+\mathbf{b})/2,\phi_m(x)\rangle\right) + \lambda$$

$$= \frac{1}{\gamma_m}\sum_{(\mathbf{a},\mathbf{b})\in(N_\varepsilon\times N_\varepsilon)} \tilde{q}^*(\mathbf{a},\mathbf{b})\|\mathbf{a}-\mathbf{b}\|^2_{\mathbf{H}(\tilde{\mathbf{q}}^*)^{-1}} \leq \frac{d}{\gamma_m}. \tag{20}$$

Note the first equality uses complementary slackness and set $\sum_{(\mathbf{a},\mathbf{b})\in(N_\varepsilon\times N_\varepsilon)} \tilde{q}^*(a,b)\lambda(\mathbf{a},\mathbf{b}) = 0$.

Now an interesting observation is that, setting $a = b = \boldsymbol{\pi}_{\phi_m}(x)$ in (18) we get: $\lambda = \lambda(a,b) \geq 0$. But that (20) simply yields:

$$\sum_{(\mathbf{a},\mathbf{b})\in(N_\varepsilon\times N_\varepsilon)} \tilde{q}^*(\mathbf{a},\mathbf{b})\left(\langle\boldsymbol{\pi}_{\phi_m}(x),\phi_m(x)\rangle - \langle(\mathbf{a}+\mathbf{b})/2,\phi_m(x)\rangle\right) \leq \frac{d}{\gamma_m}.$$

which proves the first claim.

On the other hand, (20) also implies:

$$\sum_{(\mathbf{a},\mathbf{b})\in(N_\varepsilon\times N_\varepsilon)} \tilde{q}^*(\mathbf{a},\mathbf{b})\left(\langle\boldsymbol{\pi}_{\phi_m}(x),\phi_m(x)\rangle - \langle(\mathbf{a}+\mathbf{b})/2,\phi_m(x)\rangle\right) + \lambda \leq \frac{d}{\gamma_m}$$

$$\text{i.e. } \lambda \leq \frac{d}{\gamma_m} - \sum_{(\mathbf{a},\mathbf{b})\in(N_\varepsilon\times N_\varepsilon)} \tilde{q}^*(\mathbf{a},\mathbf{b})\left(\langle\boldsymbol{\pi}_{\phi_m}(x),\phi_m(x)\rangle - \langle(\mathbf{a}+\mathbf{b})/2,\phi_m(x)\rangle\right),$$

which further implies

$$\lambda \leq \frac{d}{\gamma_m},$$

since $\sum_{(\mathbf{a},\mathbf{b})\in(N_\varepsilon\times N_\varepsilon)} \tilde{q}^*(\mathbf{a},\mathbf{b})\left(\langle\boldsymbol{\pi}_{\phi_m}(x),\phi_m(x)\rangle - \langle(\mathbf{a}+\mathbf{b})/2,\phi_m(x)\rangle\right) \geq 0$ by definition of $\boldsymbol{\pi}_{\phi_m}$. But since $\lambda(a,b) \geq 0 \ \forall a,b \in (N_\varepsilon \times N_\varepsilon)$, (18) further implies:

$$\langle\boldsymbol{\pi}_{\phi_m}(x),\phi_m(x)\rangle - \langle(\mathbf{a}+\mathbf{b})/2,\phi_m(x)\rangle - \frac{1}{\gamma_m}\|\mathbf{a}-\mathbf{b}\|^2_{\mathbf{H}(\tilde{\mathbf{q}}^*)^{-1}} + \lambda = \lambda(\mathbf{a},\mathbf{b}) \geq 0,$$

$$\text{i.e. } \frac{1}{\gamma_m}\|\mathbf{a}-\mathbf{b}\|^2_{\mathbf{H}(\tilde{\mathbf{q}}^*)^{-1}} \leq \lambda + \left(\langle\boldsymbol{\pi}_{\phi_m}(x),\phi_m(x)\rangle - \langle(\mathbf{a}+\mathbf{b})/2,\phi_m(x)\rangle\right)$$

$$\leq \frac{d}{\gamma_m} + \left(\langle\boldsymbol{\pi}_{\phi_m}(x),\phi_m(x)\rangle - \langle(\mathbf{a}+\mathbf{b})/2,\phi_m(x)\rangle\right)$$

Since above inequality holds true for any pair $(\mathbf{a},\mathbf{b}) \in (N_\varepsilon \times \varepsilon)$, the second claim of Lemma 15 now follows for any two policies $\boldsymbol{\pi}$ and $\boldsymbol{\pi}' \in \Pi$ by taking expectation over $x$ and replacing $\mathbf{a}$ by $\boldsymbol{\pi}(x)$ and $\mathbf{b}$ by $\boldsymbol{\pi}'(x)$. Precisely, the last inequality gives:

$$\mathbf{E}_{x\sim\mathcal{D}_\mathcal{X}}\left[\|\boldsymbol{\pi}(x)-\boldsymbol{\pi}'(x)\|^2_{\mathbf{H}(\tilde{\mathbf{q}}^*)^{-1}}\right] \leq \gamma_m\left(\langle\boldsymbol{\pi}_{\phi_m}(x),\phi_m(x)\rangle - \langle(\boldsymbol{\pi}(x)+\boldsymbol{\pi}'(x))/2,\phi_m(x)\rangle\right) + d$$

$$\implies \mathcal{V}_m(\boldsymbol{\pi},\boldsymbol{\pi}') \leq \gamma_m\left(\widehat{\text{BReg}}_m(\boldsymbol{\pi}) + \widehat{\text{BReg}}(\boldsymbol{\pi}')\right) + d, \tag{21}$$

where the last claim follows noting that $\tilde{\mathbf{q}}^*$ is the optimal solution of the initial optimization problem in Eq. (17) and hence we can always set $\boldsymbol{\pi}_m(x) = \tilde{\mathbf{q}}^*$ for the given context $x \in \mathcal{X}$. $\qquad\square$

**Remark 17** (A comment on correct application of KKT conditions). *It is crucial to observe that the proof of Lemma 15 relies on the $\varepsilon$-net construction precisely because it allows us to invoke the Karush–Kuhn–Tucker (KKT) conditions. To apply KKT, the optimization variable must live in a finite-dimensional space; here that variable is $\mathbf{q} \in \Delta(N_\varepsilon \times N_\varepsilon)$, whose dimensionality is finite by definition of the $\varepsilon$-net. In contrast, the original problem over $\Delta(\mathcal{K} \times \mathcal{K})$ is inherently infinite dimensional, so any Lagrange multipliers would themselves be infinite dimensional, and the standard KKT machinery would break down.*

*Importantly, introducing the $\varepsilon$-net is purely a proof device; it does not alter the algorithm or its outputs in any way. Clarifying this technical point corrects a gap that appears in several prior contextual bandit papers for continuous action spaces—for example, the proof of [17, Proposition 3] informally applies KKT conditions to an infinite-dimensional variable without addressing the attendant issues of uncountably many Lagrange multipliers! Our analysis closes that loophole by first projecting onto a finite $\varepsilon$-net and only then applying KKT. This requires significant modifications in the proof of regret analysis of Theorem 11 as detailed above.*

**Proof of Lemma 16**

We will state another important result before proving Lemma 16 that expresses the difference between the (approximate) true regret and empirical regret in terms of decision variance. The formal claim is as follows:

**Lemma 18.** *For any epoch $m$ and policy $\boldsymbol{\pi} \in \Pi$*

$$\mathrm{BReg}(\boldsymbol{\pi}) - \widehat{\mathrm{BReg}}_m(\boldsymbol{\pi}) \leq \frac{\mathcal{V}_{m-1}(\boldsymbol{\pi}_{\phi^*}, \boldsymbol{\pi})}{5\gamma_m} + \frac{5d}{16\gamma_m}$$

$$\widehat{\mathrm{BReg}}_m(\boldsymbol{\pi}) - \mathrm{BReg}(\boldsymbol{\pi}) \leq \frac{\mathcal{V}_{m-1}(\boldsymbol{\pi}_{\phi_m}, \boldsymbol{\pi})}{5\gamma_m} + \frac{5d}{16\gamma_m}$$

*Proof of Lemma 18.* Consider any epoch $m$, and let $\boldsymbol{\pi}_\phi$ denote the best (score-maximizing) policy for the scoring function $\phi$, for any $\phi \in \Phi$ (as defined in Definition 14). Then we start by noting that by definition,

$$\mathrm{BReg}(\boldsymbol{\pi}) - \widehat{\mathrm{BReg}}_m(\boldsymbol{\pi})$$

$$= \mathbf{E}_{x \sim \mathcal{D}_\mathcal{X}}[\langle \boldsymbol{\pi}_{\phi^*}(x), \phi^*(x) \rangle] - \mathbf{E}_{x \sim \mathcal{D}_\mathcal{X}}[\langle \boldsymbol{\pi}(x), \phi^*(x) \rangle] - \left( \mathbf{E}_{x \sim \mathcal{D}_\mathcal{X}}[\langle \boldsymbol{\pi}_{\phi_m}(x), \phi_m(x) \rangle] - \mathbf{E}_{x \sim \mathcal{D}_\mathcal{X}}[\langle \boldsymbol{\pi}(x), \phi_m(x) \rangle] \right)$$

$$\leq \mathbf{E}_{x \sim \mathcal{D}_\mathcal{X}}[\langle \boldsymbol{\pi}_{\phi^*}(x), \phi^*(x) \rangle] - \mathbf{E}_{x \sim \mathcal{D}_\mathcal{X}}[\langle \boldsymbol{\pi}(x), \phi^*(x) \rangle] - \left( \mathbf{E}_{x \sim \mathcal{D}_\mathcal{X}}[\langle \boldsymbol{\pi}_{\phi^*}(x), \phi_m(x) \rangle] - \mathbf{E}_{x \sim \mathcal{D}_\mathcal{X}}[\langle \boldsymbol{\pi}(x), \phi_m(x) \rangle] \right)$$

$$= \mathbf{E}_{x \sim \mathcal{D}_\mathcal{X}}\left[ \Big| \langle \big( \boldsymbol{\pi}_{\phi^*}(x) - \boldsymbol{\pi}(x) \big), \big( \phi^*(x) - \phi_m(x) \big) \rangle \Big| \right]$$

$$\overset{(a)}{\leq} \mathbf{E}_{x \sim \mathcal{D}_\mathcal{X}}\left[ \sqrt{2/5\gamma_m} \|\boldsymbol{\pi}_{\phi^*}(x) - \boldsymbol{\pi}(x)\|_{H(\boldsymbol{\pi}_{m-1}(x))^{-1}} \sqrt{5\gamma_m/2} \|\phi^*(x) - \phi_m(x)\|_{H(\boldsymbol{\pi}_{m-1}(x))} \right]$$

$$\overset{(b)}{\leq} \mathbf{E}_{x \sim \mathcal{D}_\mathcal{X}}\left[ \frac{1}{5\gamma_m} \|\boldsymbol{\pi}_{\phi^*}(x) - \boldsymbol{\pi}(x)\|^2_{H(\boldsymbol{\pi}_{m-1}(x))^{-1}} + \frac{5\gamma_m}{4} \|\phi^*(x) - \phi_m(x)\|^2_{H(\boldsymbol{\pi}_{m-1}(x))} \right]$$

$$\overset{(c)}{\leq} \frac{1}{5\gamma_m} \mathbf{E}_{x \sim \mathcal{D}_\mathcal{X}}\left[ \|\boldsymbol{\pi}_{\phi^*}(x) - \boldsymbol{\pi}(x)\|^2_{H(\boldsymbol{\pi}_{m-1}(x))^{-1}} \right] + \frac{5}{4}\gamma_m \mathcal{E}_{\mathrm{off},\mathcal{M}}(\delta/2m^2, \tau_{m-1} - \tau_{m-2})$$

$$\leq \frac{\mathcal{V}_{m-1}(\boldsymbol{\pi}_{\phi^*}, \boldsymbol{\pi})}{5\gamma_m} + \frac{5d}{16\gamma_m},$$

where (a) uses Holder's inequality, (b) applies Cauchy's Schwarz and AM-GM inequality. Inequality (c) follows noting that

$$\|\phi^*(x) - \phi_m(x)\|^2_{H(\boldsymbol{\pi}_{m-1}(x))} = \mathbf{E}_{(\mathbf{a},\mathbf{b}) \sim \boldsymbol{\pi}_{m-1}(x)}[\langle \phi^*(x) - \phi_m(x), (\mathbf{a} - \mathbf{b}) \rangle^2] + c\|\phi^*(x) - \phi_m(x)\|^2_2$$

$$\overset{(a)}{\leq} \sum_{(\mathbf{a},\mathbf{b}) \in (N_\varepsilon \times N_\varepsilon)} \boldsymbol{\pi}_{m-1}(x)[\mathbf{a}, \mathbf{b}]\langle \phi^*(x) - \phi_m(x), (\mathbf{a} - \mathbf{b}) \rangle^2 + 2cD^2$$

$$= \mathcal{E}_{\mathrm{off},\mathcal{M}}(\delta/2m^2, \tau_{m-1} - \tau_{m-2}) + \frac{5d}{32\gamma_m},$$

where $(a)$ applied Cauchy-Schwarz, and the last equality follows from Definition 17, our choice of $\gamma_m = \frac{1}{2\sqrt{2}} \left( \frac{d}{\mathcal{E}_{\text{off},\Phi^*}(\delta/2m^2, \tau_{m-1}-\tau_{m-2})} \right)^{1/2}$, and since we set $c = \frac{5d}{64\gamma_{m(T)}D^2} \leq \frac{5d}{64\gamma_m D^2}$. This completes the first part of the proof. The second part of the proof follows almost with the same argument as shown above, with the additional observation that:

$$\widehat{\text{BReg}}_m(\boldsymbol{\pi}) - \text{BReg}(\boldsymbol{\pi})$$
$$= \mathbf{E}_{x\sim\mathcal{D}_\mathcal{X}}[\langle \boldsymbol{\pi}_{\phi_m}(x), \phi_m(x)\rangle] - \mathbf{E}_{x\sim\mathcal{D}_\mathcal{X}}[\langle \boldsymbol{\pi}(x), \phi_m(x)\rangle] - \left( \mathbf{E}_{x\sim\mathcal{D}_\mathcal{X}}[\langle \boldsymbol{\pi}_{\phi^*}(x), \phi^*(x)\rangle] - \mathbf{E}_{x\sim\mathcal{D}_\mathcal{X}}[\langle \boldsymbol{\pi}(x), \phi^*(x)\rangle] \right)$$
$$\leq \mathbf{E}_{x\sim\mathcal{D}_\mathcal{X}}\left[ \left| \langle (\boldsymbol{\pi}_{\phi_m}(x) - \boldsymbol{\pi}(x)), (\phi_m(x) - \phi^*(x))\rangle \right| \right].$$

$\square$

Given the results of Lemma 18, we are now ready to proof Lemma 16.

**Lemma 16** (Emp vs True Performance). *Let $\boldsymbol{\pi}$ be any policy in $\Pi^2$. Then for all epochs $m \in \mathbb{N}_+$:*

$$\text{BReg}(\boldsymbol{\pi}^\ell) + \text{BReg}(\boldsymbol{\pi}^r) \leq 2\widehat{\text{BReg}}_m(\boldsymbol{\pi}^\ell) + 2\widehat{\text{BReg}}_m(\boldsymbol{\pi}^r) + 6d/\gamma_m,$$
$$\widehat{\text{BReg}}_m(\boldsymbol{\pi}^\ell) + \widehat{\text{BReg}}_m(\boldsymbol{\pi}^r) \leq 2\text{BReg}(\boldsymbol{\pi}^\ell) + 2\text{BReg}(\boldsymbol{\pi}^r) + 6d/\gamma_m.$$

*Proof of Lemma 16.* The proof follows exactly the same as the proof of Lemma 10.

Let us fix $c_0 = 6$ for the rest of the proof. We prove the claim via induction on $m$. We first consider the base case where $m = 1$ and $1 \leq t \leq \tau_1$. In this case, since $\gamma_1 = 1$,

$$\text{BReg}(\boldsymbol{\pi}^\ell) + \text{BReg}(\boldsymbol{\pi}^r) \leq 2 \leq c_0 d/\gamma_1, \quad \widehat{\text{BReg}}_1(\boldsymbol{\pi}^\ell) + \widehat{\text{BReg}}_1(\boldsymbol{\pi}^r) \leq 2 \leq c_0 d/\gamma_1$$

Thus the claim holds in the base case.

For the inductive step, let us fix some epoch $m > 1$. We assume that for all epochs $m' < m$, all $\boldsymbol{\pi} \in \Pi^2$,

$$\text{BReg}(\boldsymbol{\pi}^\ell) + \text{BReg}(\boldsymbol{\pi}^r) \leq 2\widehat{\text{BReg}}_{m'}(\boldsymbol{\pi}^\ell) + 2\widehat{\text{BReg}}_{m'}(\boldsymbol{\pi}^r) + c_0 d/\gamma_{m'}, \tag{22}$$

$$\widehat{\text{BReg}}_{m'}(\boldsymbol{\pi}^\ell) + \widehat{\text{BReg}}_{m'}(\boldsymbol{\pi}^r) \leq 2\text{BReg}(\boldsymbol{\pi}^\ell) + 2\text{BReg}(\boldsymbol{\pi}^r) + c_0 d/\gamma_{m'}. \tag{23}$$

**Part-1:** We will first show that for epoch $m$, and all $\boldsymbol{\pi} \in \Pi^2$,

$$\text{BReg}(\boldsymbol{\pi}^\ell) + \text{BReg}(\boldsymbol{\pi}^r) \leq 2\widehat{\text{BReg}}_m(\boldsymbol{\pi}^\ell) + 2\widehat{\text{BReg}}_m(\boldsymbol{\pi}^r) + c_0 d/\gamma_m.$$

We start by noting that using Lemma 18 we have:

$$\text{BReg}(\boldsymbol{\pi}^\ell) - \widehat{\text{BReg}}_m(\boldsymbol{\pi}^\ell) \leq \frac{\mathcal{V}_{m-1}(\boldsymbol{\pi}_{\phi^*}, \boldsymbol{\pi}^\ell)}{5\gamma_m} + \frac{5d}{16\gamma_m}.$$

Similarly, applying Lemma 18 we get that:

$$\text{BReg}(\boldsymbol{\pi}^r) - \widehat{\text{BReg}}_m(\boldsymbol{\pi}^r) \leq \frac{\mathcal{V}_{m-1}(\boldsymbol{\pi}_{\phi^*}, \boldsymbol{\pi}^r)}{5\gamma_m} + \frac{5d}{16\gamma_m}.$$

Combining the above two inequalities we get:

$$\text{BReg}(\boldsymbol{\pi}^\ell) + \text{BReg}(\boldsymbol{\pi}^r) - [\widehat{\text{BReg}}_m(\boldsymbol{\pi}^\ell) + \widehat{\text{BReg}}_m(\boldsymbol{\pi}^r)] \leq \frac{\mathcal{V}_{m-1}(\boldsymbol{\pi}_{\phi^*}, \boldsymbol{\pi}^\ell)}{5\gamma_m} + \frac{\mathcal{V}_{m-1}(\boldsymbol{\pi}_{\phi^*}, \boldsymbol{\pi}^r)}{5\gamma_m} + \frac{5d}{8\gamma_m}. \tag{24}$$

Now applying Lemma 15 we know that:

$$\mathcal{V}_{m-1}(\boldsymbol{\pi}_{\phi^*}, \boldsymbol{\pi}^\ell) \leq d + \gamma_{m-1}\widehat{\text{BReg}}_{m-1}(\boldsymbol{\pi}_{\phi^*}) + \gamma_{m-1}\widehat{\text{BReg}}_{m-1}(\boldsymbol{\pi}^\ell), \tag{25}$$

$$\mathcal{V}_{m-1}(\boldsymbol{\pi}_{\phi^*}, \boldsymbol{\pi}^r) \leq d + \gamma_{m-1}\widehat{\mathrm{BReg}}_{m-1}(\boldsymbol{\pi}_{\phi^*}) + \gamma_{m-1}\widehat{\mathrm{BReg}}_{m-1}(\boldsymbol{\pi}^r); \tag{26}$$

Summing the results from the above two inequalities, we get:

$$
\begin{aligned}
&\frac{\mathcal{V}_{m-1}(\boldsymbol{\pi}_{\phi^*}, \boldsymbol{\pi}^\ell) + \mathcal{V}_{m-1}(\boldsymbol{\pi}_{\phi^*}, \boldsymbol{\pi}^r)}{5\gamma_m} \\
&\leq \frac{2d + \gamma_{m-1}\big(\widehat{\mathrm{BReg}}_{m-1}(\boldsymbol{\pi}_{\phi^*}) + \widehat{\mathrm{BReg}}_{m-1}(\boldsymbol{\pi}^\ell) + \widehat{\mathrm{BReg}}_{m-1}(\boldsymbol{\pi}_{\phi^*}) + \widehat{\mathrm{BReg}}_{m-1}(\boldsymbol{\pi}^r)\big)}{5\gamma_m} \\
&\overset{(a)}{\leq} \frac{2(d + c_0 d) + 2\gamma_{m-1}\big(\mathrm{BReg}(2\boldsymbol{\pi}_{\phi^*}) + \mathrm{BReg}(\boldsymbol{\pi}^\ell) + \mathrm{BReg}(\boldsymbol{\pi}^r)\big)}{5\gamma_m} \\
&\overset{(b)}{\leq} \frac{2(d + c_0 d) + 2\gamma_{m-1}\big(\mathrm{BReg}(\boldsymbol{\pi}^\ell) + \mathrm{BReg}(\boldsymbol{\pi}^r)\big)}{5\gamma_m} \\
&\leq \frac{2d(1 + c_0)}{5\gamma_m} + \frac{2\big(\mathrm{BReg}(\boldsymbol{\pi}^\ell) + \mathrm{BReg}(\boldsymbol{\pi}^r)\big)}{5}
\end{aligned}
$$

where $(a)$ follows from Eq. (23), and $(b)$ follows from the fact that $\mathrm{BReg}(\boldsymbol{\pi}_{\phi^*}) = 0$ by Definition 13. The last inequality follows since by choice $\gamma_{m-1} \leq \gamma_m$. Combining the above two inequalities with (24) we get:

$$
\begin{aligned}
\mathrm{BReg}(\boldsymbol{\pi}^\ell) + \mathrm{BReg}(\boldsymbol{\pi}^r) - [\widehat{\mathrm{BReg}}_m(\boldsymbol{\pi}^\ell) + \widehat{\mathrm{BReg}}_m(\boldsymbol{\pi}^r)] &\leq \frac{\mathcal{V}_{m-1}(\boldsymbol{\pi}_{\phi^*}, \boldsymbol{\pi}^\ell)}{5\gamma_m} + \frac{\mathcal{V}_{m-1}(\boldsymbol{\pi}_{\phi^*}, \boldsymbol{\pi}^r)}{5\gamma_m} + \frac{5d}{8\gamma_m} \\
&\leq \frac{2d(1 + c_0)}{5\gamma_m} + \frac{2\big(\mathrm{BReg}(\boldsymbol{\pi}^\ell) + \mathrm{BReg}(\boldsymbol{\pi}^r)\big)}{5} + \frac{5d}{8\gamma_m},
\end{aligned}
$$

which implies

$$\frac{3}{5}[\mathrm{BReg}(\boldsymbol{\pi}^\ell) + \mathrm{BReg}(\boldsymbol{\pi}^r)] - [\widehat{\mathrm{BReg}}_m(\boldsymbol{\pi}^\ell) + \widehat{\mathrm{BReg}}_m(\boldsymbol{\pi}^r)] \leq \frac{2d(1 + c_0)}{5\gamma_m} + \frac{5d}{8\gamma_m}.$$

Consequently, we get:

$$\mathrm{BReg}(\boldsymbol{\pi}^\ell) + \mathrm{BReg}(\boldsymbol{\pi}^r) \leq \frac{5}{3}[\widehat{\mathrm{BReg}}_m(\boldsymbol{\pi}^\ell) + \widehat{\mathrm{BReg}}_m(\boldsymbol{\pi}^r)] + \frac{2d(1 + c_0)}{3\gamma_m} + \frac{25d}{24\gamma_m},$$

finally leading to:

$$\mathrm{BReg}(\boldsymbol{\pi}^\ell) + \mathrm{BReg}(\boldsymbol{\pi}^r) \leq 2[\widehat{\mathrm{BReg}}_m(\boldsymbol{\pi}^\ell) + \widehat{\mathrm{BReg}}_m(\boldsymbol{\pi}^r)] + \frac{c_0 K^2}{\gamma_m}, \tag{27}$$

for the choice of $c_0 = 6$ and recall that we assumed that $\boldsymbol{\pi}$ is any arbitrary policy. This concludes the first part of the proof.

The second part of the claim can be proved almost following the similar tricks. We add the analysis below for completeness.

**Part-2:** To prove the second part, we start by noting that, similar to (24), we can show:

$$\widehat{\mathrm{BReg}}(\boldsymbol{\pi}^\ell) + \widehat{\mathrm{BReg}}(\boldsymbol{\pi}^r) - [\mathrm{BReg}_m(\boldsymbol{\pi}^\ell) + \mathrm{BReg}_m(\boldsymbol{\pi}^r)] \leq \frac{\mathcal{V}_{m-1}(\boldsymbol{\pi}_{\phi_m}, \boldsymbol{\pi}^\ell)}{5\gamma_m} + \frac{\mathcal{V}_{m-1}(\boldsymbol{\pi}_{\phi_m}, \boldsymbol{\pi}^r)}{5\gamma_m} + \frac{5d}{8\gamma_m}. \tag{28}$$

Following a similar analysis from above, we can obtain:

$$\frac{\mathcal{V}_{m-1}(\boldsymbol{\pi}_{\phi_m}, \boldsymbol{\pi}^\ell) + \mathcal{V}_{m-1}(\boldsymbol{\pi}_{\phi_m}, \boldsymbol{\pi}^r)}{5\gamma_m}$$

$$\leq \frac{2d + \gamma_{m-1}\big(\widehat{\mathrm{BReg}}_{m-1}(\boldsymbol{\pi}_{\phi_m}) + \widehat{\mathrm{BReg}}_{m-1}(\boldsymbol{\pi}^\ell) + \widehat{\mathrm{BReg}}_{m-1}(\boldsymbol{\pi}_{\phi_m}) + \widehat{\mathrm{BReg}}_{m-1}(\boldsymbol{\pi}^r)\big)}{5\gamma_m}$$

$$\overset{(a)}{\leq} \frac{2(d + c_0 d) + 2\gamma_{m-1}\big([\mathrm{BReg}(\boldsymbol{\pi}_{\phi_m}) + \mathrm{BReg}(\boldsymbol{\pi}_{\phi_m})] + [\mathrm{BReg}(\boldsymbol{\pi}^\ell) + \mathrm{BReg}(\boldsymbol{\pi}^r)]\big)}{5\gamma_m}$$

$$\overset{(b)}{\leq} \frac{2(d + 2c_0 d) + 2\gamma_{m-1}\big(4[\widehat{\mathrm{BReg}}_m(\boldsymbol{\pi}_{\phi_m})] + [\mathrm{BReg}(\boldsymbol{\pi}^\ell) + \mathrm{BReg}(\boldsymbol{\pi}^r)]\big)}{5\gamma_m}$$

$$\overset{(c)}{\leq} \frac{2(d + 2c_0 d) + 2\gamma_{m-1}\big(\mathrm{BReg}(\boldsymbol{\pi}^\ell) + \mathrm{BReg}(\boldsymbol{\pi}^r)\big)}{5\gamma_m}$$

$$\leq \frac{2d(1 + 2c_0)}{5\gamma_m} + \frac{2\big(\mathrm{BReg}(\boldsymbol{\pi}^\ell) + \mathrm{BReg}(\boldsymbol{\pi}^r)\big)}{5}$$

where $(a)$ follows from Eq. (23), $(b)$ follows since we have already proved the first part of the induction for epoch $m$ as concluded in Eq. (27), and $(c)$ follows from the fact that $\widehat{\mathrm{BReg}}_m(\boldsymbol{\pi}_{\phi_m}) = 0$ by Assumption 3. Combining this with (28):

$$\widehat{\mathrm{BReg}}_m(\boldsymbol{\pi}^\ell) + \widehat{\mathrm{BReg}}_m(\boldsymbol{\pi}^r) - [\mathrm{BReg}(\boldsymbol{\pi}^\ell) + \mathrm{BReg}(\boldsymbol{\pi}^r)] \leq \frac{\mathcal{V}_{m-1}(\boldsymbol{\pi}_{\phi_m}, \boldsymbol{\pi}^\ell)}{5\gamma_m} + \frac{\mathcal{V}_{m-1}(\boldsymbol{\pi}_{\phi_m}, \boldsymbol{\pi}^r)}{5\gamma_m} + \frac{5d}{8\gamma_m}$$

$$\leq \frac{2d(1 + 2c_0)}{5\gamma_m} + \frac{2\big(\mathrm{BReg}(\boldsymbol{\pi}^\ell) + \mathrm{BReg}(\boldsymbol{\pi}^r)\big)}{5} + \frac{5d}{8\gamma_m},$$

which implies

$$\widehat{\mathrm{BReg}}_m(\boldsymbol{\pi}^\ell) + \widehat{\mathrm{BReg}}_m(\boldsymbol{\pi}^r) - \frac{7}{5}[\mathrm{BReg}(\boldsymbol{\pi}^\ell) + \mathrm{BReg}(\boldsymbol{\pi}^r)] \leq \frac{2d(1 + 2c_0)}{5\gamma_m} + \frac{5d}{8\gamma_m}.$$

Consequently, we have:

$$\widehat{\mathrm{BReg}}_m(\boldsymbol{\pi}^\ell) + \widehat{\mathrm{BReg}}_m(\boldsymbol{\pi}^r) \leq \frac{7}{5}[\mathrm{BReg}(\boldsymbol{\pi}^\ell) + \mathrm{BReg}(\boldsymbol{\pi}^r)] + \frac{2d(1 + 2c_0)}{5\gamma_m} + \frac{5d}{8\gamma_m},$$

finally leading to:

$$\widehat{\mathrm{BReg}}_m(\boldsymbol{\pi}^\ell) + \widehat{\mathrm{BReg}}_m(\boldsymbol{\pi}^r) \leq 2[\widehat{\mathrm{BReg}}_m(\boldsymbol{\pi}^\ell) + \widehat{\mathrm{BReg}}_m(\boldsymbol{\pi}^r)] + \frac{c_0 d}{\gamma_m},$$

which is again satisfied for any choice of $c_0 = 6$, concluding the entire proof.

$\square$

### D.3 Additional useful results for proving Theorem 11

**Lemma 19.** *Let* $\mathbf{A}$ *be any* $d$-*dimensional postive semi-definite matrix and* $\tilde{\mathbf{A}}' = \mathbf{A} + c\mathbf{I}_d$ *for some positive constant* $c > 0$. *Then*
$$\mathrm{Tr}(\tilde{\mathbf{A}}^{-1}\mathbf{A}) \leq d.$$

*Proof of Lemma 19.* Consider the eigen-decomposition of $\mathbf{A} = \mathbf{U}^\top \boldsymbol{\Sigma} \mathbf{U} = \sum_{i=1}^d \sigma_i \mathbf{u}_i \mathbf{u}_i^\top$, where $\sigma_i \geq 0$ is the $i$-th eigenvalue of $\mathbf{A}$ ($\boldsymbol{\Sigma}$ is the diagonal matrix with $\Sigma(i, i) = \sigma_i$), and $\mathbf{U} = [\mathbf{u}_1 \ldots \mathbf{u}_d]$ is orthogonal matrix with its columns, $\mathbf{u}_1, \ldots, \mathbf{u}_d$ being orthogonal vectors.

By definition of $\mathbf{A}$, it then follows that the eigen-decomposition of $\tilde{\mathbf{A}} = \sum_{i=1}^d (\sigma_i + c)\mathbf{u}_i \mathbf{u}_i^\top$ and hence $\tilde{\mathbf{A}}^{-1} = \sum_{i=1}^d \frac{1}{\sigma_i + c} \mathbf{u} \mathbf{u}^\top$.

Combining the above insights, we get: $\tilde{\mathbf{A}}^{-1}\mathbf{A} = \sum_{i=1}^d \frac{\sigma_i}{\sigma_i + c} \mathbf{u}_i \mathbf{u}_i^\top$ represents the eigen-decomposition of $\tilde{\mathbf{A}}^{-1}\mathbf{A}$. This further given $\mathrm{Tr}(\tilde{\mathbf{A}}^{-1}\mathbf{A}) \sum_{i=1}^d \frac{\sigma_i}{\sigma_i + c} \leq d$ since the trace of a matrix is the sum of its eigenvalues. $\square$

# E Supplementary for Section 7: Additional Experiments

## E.1 Running Time Performance

We also report the running time performance of `Double-Monster` on Inst-1:

| d | runtime (sec) |
|----|---------------|
| 4 | 24.481216 |
| 6 | 117.084477 |
| 10 | 176.244782 |
| 15 | 501.716698 |

T = 10,000 (Inst-1)

We here report the (averaged) runtimes of the above executions (in seconds) with increasing $d$ to check its runtime efficiency of `Double-Monster-Inf`. We report this for **Inst-1** for $T = 10,000$. Despite large $T$ and $d$ the complete runs of the algorithm are finished within a few minutes, which demonstrates its computational efficiency. The experiments are run on a standard MacBook Pro (36GB RAM).

# NeurIPS Checklist:

