# OpenReview forum: "Efficient and Near-Optimal Algorithm for Contextual Dueling Bandits with Offline Regression Oracles"
_NeurIPS.cc/2025/Conference — NeurIPS 2025 poster_

### Official Review · Reviewer_czBn · 2025-06-23

**Clarity:** 2
**Significance:** 4
**Originality:** 3
**Rating:** 4
**Confidence:** 3

**Summary:**

This work investigates dueling contextual bandits in the presence of offline regression oracles. It introduces dueling bandit algorithms for both finite and infinite action spaces and establishes square-root regret upper bounds. The contributions are theoretically well-founded and significant, though the presentation could benefit from improved clarity and organization.

**Questions:**

* The proof of Corollary 4 is hard to follow. Please explain how the bound $\mathcal{E}_{\mathrm{off}, \mathcal{M}}(\delta, n)=O\left(\frac{\log (|\mathcal{M}| n)}{n \delta}\right)$ with $\delta = 1/T$ leads to the regret bound stated in the corollary, especially focusing on the final two lines of the proof.

* Could you derive the regret bound $O(\sqrt{dT})$ for the continuous case by plugging an appropriate one to $\mathcal{E}$?

**Ethical Concerns:**

["NO or VERY MINOR ethics concerns only"]

**Final Justification:**

I keep my rating based on the authors' response.

**Limitations:**

The simulation results do not show various cases.

**Quality:**

3

**Strengths And Weaknesses:**

The need for this work is evident, as interest in dueling bandits with offline regression oracles continues to grow. The theoretical foundations are solid and promising. However, the current presentation requires substantial improvement before the work can be considered for acceptance. In particular, the proofs and notation are difficult to follow. Clarifying the following points would lead to a significant improvement:

* Line 54 contains a repeated expression; Line 55 includes a typographical error.
* The range of $o_t$ in Line 108 should be explicitly clarified.
* Does $p(i,j)$ refer to the $(i,j)$-th component of the vector $\mathbf{p}$? Please clarify.
* There appear to be typos in Equation (2).
* Lines 174–175 should be corrected, with attention to whether $p_t$ represents a joint or marginal distribution.
* Lines 176–184 should be rewritten for improved readability, including clearer notation. Specifically, the meaning of $q^t M p$, the interpretation of its positive or negative value, and the role of the Nash equilibrium need to be better explained.
* What does $\psi^*$ denote in Assumption 3? This should be made explicit.
* Multiple notions of regret are introduced throughout the paper without sufficient explanation or connection. These definitions should be clearly distinguished and related to each other.
* Is the symbol $\sigma$ in Line 290 referring to the sigmoid function? Please clarify.

---

> ### Author Rebuttal · Authors · 2025-07-31
>
> Thanks a lot for your careful reading and insightful comments.
>
> > W1: Some notational clarification:
>
> - Line 54, 55: Will fix repeated expression in L54 and the expression of Fig2 in L55
> - Line 108: Please note we clarified this in L107, $o_t \in \{0,1\}$ is the binary preference feedback
> - $p(i,j)$: Yes, note $p \in \Delta_{K \times K}$ is a joint distribution over item pairs. Hence $p(i,j) \in [0,1]$ denotes the probability of $(i,j)$-th pair in $p$.
> - Equation (2): Will correct the typo by changing $b \to a$ in the 1st term of Eq2.
> - Lines 174-175: Note we mentioned $p \in \Delta_{K \times K}$ (eg L166, L173), so $p$ is always a joint distribution. Will clarify more if needed.
> - Lines 176-184: For any two mixed strategies $p \in \Delta_K$ and $q \in \Delta_K$, $q^\top M p$ denotes the expected probability of $q$ winning over $p$. So a +ve value represents that $q$ is a stronger policy than $p$ and vice-versa. [9,13] also gives a detailed explanation of Nash (Von Neumann) policies of symmetric two-player 0-sum games, as referenced. Will add more details to improve the readability of the remark.
> - ${\psi}^{\star}$ in Assump3: Thanks for catching this. ${\psi}^{\star}$ represents "Best-Response Policy", we will add a reference (Defn6, A.3) to it in Assump 3.
> - Line 290: Yes, $\sigma$ refers to the sigmoid function, as clarified in the objective, L125.
>
> Many thanks for pointing them out, will clarify all of them in the update.
>
> > Q1: The Final two lines in the proof of Cor4: This is a standard analysis in the regret analysis of epoch-based contextual bandit algorithms, e.g., please see the last steps of the derivation provided in [30] for their proof of Cor1 (or even Thm1) to see the final steps. We will also add a few extra lines of derivation in our proof of Cor4 (C.5) to make the proof clearer.
>
> >Q2: Deriving pure $\tilde O(\sqrt{dT})$ regret bound for the continuous case: Yes, absolutely. When $\kappa$ is the $\ell_2$ ball of radius 1 and every $\phi \in \Phi$ is 1-$\ell_2$ norm bounded (ie. $\forall x \in \mathcal X, ~||\phi(x)||_2 \leq 1$), using the simple vanilla least-square regression oracle will yield the desired $\tilde O(\sqrt{dT})$ regret bound. We will add this as a corollary to analyze some special cases of Thm9 (same as Cor4). Thanks again for the great suggestion!
>
> ---
>
> We hope this addresses all of your concerns. We are happy to clarify any further questions and respectfully urge the reviewer to reconsider the final evaluation in light of the rebuttal.

---

> ### Author Response · Authors · 2025-08-04
> **Request for Further Discussion**
>
> Dear Reviewer czBn,
>
> We sincerely appreciate your time, thoughtful evaluation. As the discussion phase is now active and the decision deadline approaches, we wanted to kindly follow up on our rebuttal.
>
> We hope our responses addressed your concerns as well, and we would be truly grateful for any additional feedback or follow-up questions you may have.
>
> If any points remain unclear or merit further discussion, we would be more than happy to elaborate. We believe that a brief exchange at this stage could help resolve any remaining uncertainties and potentially contribute to a more favorable outcome.
>
> Thank you once again for your time and consideration—we greatly value your input.
>
> Thanks, Authors

---

### Official Review · Reviewer_ACKv · 2025-06-26

**Clarity:** 3
**Significance:** 2
**Originality:** 2
**Rating:** 4
**Confidence:** 3

**Summary:**

This paper addresses the problem of contextual dueling bandits by leveraging an offline regression oracle, rather than relying on an online regression oracle. This improves the practicality of the proposed algorithm. Furthermore, the authors present the first theoretical analysis of general contextual dueling bandits over continuous action spaces. In a regularized min-max optimization framework, they establish a regret bound of $\tilde{\mathcal{O}}(\sqrt{dT})$. Their theoretical results are further supported by empirical evaluations in synthetic environments.

**Questions:**

My questions are as follows:

1. Clarification on IBR: Could the authors provide a more detailed explanation of the IBR assumption? In particular, could you give concrete examples of settings where IBR holds, while other models such as RUM or SST do not?

2. Function class in practice:
Is there an efficient or practical way to define and implement the function class used in this work?

3. Regret dependence on $\kappa$:
In dueling or logistic bandits with linear rewards, it is common for the regret to scale with the non-linearity of the sigmoid function $\kappa$ (e.g., [1,2]). However, despite using a sigmoid function in Objective (2), the regret bound in this work appears to have no dependence on $\kappa$. Could the authors clarify why this is the case?


[1] Zhu, Banghua, Michael Jordan, and Jiantao Jiao. "Principled reinforcement learning with human feedback from pairwise or k-wise comparisons." International Conference on Machine Learning. PMLR, 2023.

[2] Abeille, Marc, Louis Faury, and Clément Calauzènes. "Instance-wise minimax-optimal algorithms for logistic bandits." International Conference on Artificial Intelligence and Statistics. PMLR, 2021.

How to define function class in exp?

What parameters..?

Non linearity?

**Ethical Concerns:**

["NO or VERY MINOR ethics concerns only"]

**Final Justification:**

Most of my concerns have been addressed.

Thus, I will maintain my positive score.

**Limitations:**

The authors are encouraged to create a separate "Limitations" section in their paper.

**Quality:**

3

**Strengths And Weaknesses:**

**Strengths**

This paper makes a meaningful contribution to the community by advancing dueling bandit research toward more practical settings—specifically, by addressing general function classes and continuous decision spaces. The theoretical arguments are well-supported, and the presentation is clear and well-structured. Additionally, the inclusion of empirical experiments in the context of oracle-based approaches is both interesting and valuable.


**Weaknesses**

There are some concerns that, if addressed, could further strengthen the paper:

1. Computational complexity:

     - What is the main computational bottleneck in the proposed algorithm?
     -  Moreover, what are the theoretical computational complexities of the Convex-Constraint-Solver and Cont-CvxConstraint-Solver subroutines used in the method?

2. Experiments:

     - No implementation details or code are provided: It is unclear what parameters were used in the experiments. Some of these parameters appear difficult to set without prior knowledge, so I wonder whether the authors used theoretically justified values or tuned them heuristically. Specifically, how is the model class defined in practice? Clarifying this would improve reproducibility.
     - The model used in the experiments is quite small: Since one of the main motivations of this paper is RLHF, it would be more interesting to evaluate the algorithm with larger models, such as neural networks or transformers.
     - No real-world dataset experiments.
     - No comparison of computational cost with baselines: It is important to assess the practical efficiency of the algorithm in terms of runtime or wall-clock time.

3. What are the main technical challenges addressed in this work compared to existing methods? Providing such clarification would strengthen the paper.

---

> ### Author Rebuttal · Authors · 2025-07-31
>
> Thanks a lot for your careful reading and insightful comments.
>
> > W1: Computation Complexity
>
> Please see Q3 of Reviewer g3ya.
>
> > W2: Experiments:
>
> Please see Q1 of Reviewer g3ya.
>
> > Q1. IBR Assumption:
>
> Assumption 3 posits the existence of one or more “good” items that strictly outduel every other item and are equally strong among themselves. Concretely, the class of preference matrices ({P}) must allow for a “best set” of actions that tie each other and simultaneously outscore all remaining actions.
>
> *Example of IBR but not RUM or SST*:
>
> There are many such examples. Any random utility based model follows SST by design, so $RUM \subset SST$. But $SST \subset IBR$ and the class of IBR is much larger than SST. E.g., any preference relation with a strong Condorcet winner automatically satisfies the IBR property, without having any transitivity relation; assume a preference matrix $P \in [0,1]^{K \timed K}$ s.t. $1 \leftarrow \arg\max_{i \in [K]}P(i,j), ~ \forall j \in [K]$ (hence 1 is also a CW). Another example is having a equally strong set of top-cycle arms $\mathcal C \subset [K]$ s.t. $P(i,j) = 0.5 ~\forall i,j \in \mathcal C$ and for any $i \in \mathcal C$ and $j, j' \in [K] \setminus \mathcal C$, $P(i,j) > P(j',j)$. Again such top-cycle class of preferences need not have any transitivity and can easily violate SST throughout.
>
> *Thanks for bringing this up, we will add these examples in the paper (remark after Assump3).*
>
> *Note IBR holds by default for the Continuous decision space setting*. Since RUM-like preference structures automatically satisfy the condition of having a best (or set of best) item(s), Assumption 3 trivially holds in our continuous setting (Alg. 2). Thus, in large or potentially infinite action spaces where utility-based models are commonplace, the assumption naturally remains valid and helps maintain the same recursion-based analysis as in the finite-armed case.
>
> >Q2: ``efficient or practical way to define and implement the function class used in this work"
>
> To the best of our understanding, there might be some misunderstanding on the reviewer's part since we can define $\mathcal M$ as broadly and generally as we want it to be based on the requirement/ complexity of the underlying preference environment. We have discussed multiple such functions classes (and the choice of efficient regression oracles for the corresponding  We discussed several examples of such hypothesis preference classes in $\mathcal M$ in A.3 and Cor 4).
>
> Note such hypothesis‑class premise is identical to many (highly cited) contextual‑bandit work (ILTCB, SquareCB, FALCON) and in classical supervised‑learning settings such as linear and generalized‑linear regression, RKHS models, and deep networks (see [17, 30], Li et al. ’22, Zhu et al. ’22, Blum et al. ’21).
>
> It's important to note that, despite the primary theoretical focus of our work, the assumption is also not restrictive in practice
>  since the algorithm designer is free to choose $\mathcal{M}$! It can be *arbitrarily rich*—finite sets, GLMs, RKHS classes, Banach spaces, or deep networks that approximate Sobolev‑ball functions (Farrell ’21). Of course, any theoretical guarantee will require some structural assumptions for provable guarantees, but Realizability is a very primitive level (broad) assumption, which is satisfied in practice for the appropriate choice of (rich enough) function classes.
>
> Additionally, since our algorithm relies on an **offline** regression oracle (rather than an online oracle as we described the advantages in Rem 1 and App A), this flexibility makes the assumption milder, not stronger.
>
> Does that answer your question (esp the discussion in A.3 and Cor 4)? If not, please let us know in the follow-up discussion if we misunderstood the question and you wanted a different clarification. We will be happy to clarify the details accordingly.
>
> > Q3. Dependency on $\kappa$
>
> Thanks for the great question, the dependency on the slope of the sigmoid ($\kappa$) comes through the performance of the regression oracle $\mathcal E_{off, \Phi}(\delta)$ (see Thm 9), as the regression oracle is responsible for finding a good estimator of the underlying preference relation (which is sigmoid-based in this case). We can use any MLE based regression oracle (Li et al'17 for GLM bandits) for the purpose.
>
> ---
>
> We hope this addresses all of your concerns. We are happy to clarify any further questions and respectfully urge the reviewer to reconsider the final evaluation in light of the rebuttal.

---

> > ### Comment · Reviewer_ACKv · 2025-08-04
> >
> > Thank you for your thorough response. As most of my concerns have been adequately addressed, I will maintain my positive score.

---

> > > ### Author Response · Authors · 2025-08-04
> > > **Thank you**
> > >
> > > Dear Reviewer ACKv,
> > >
> > > Many thanks for taking the time to review our rebuttal and share your feedback.
> > >
> > > We are glad to hear that our responses helped clarify your concerns adequately, and we sincerely appreciate your consideration to maintain the positive score. Your comments and suggestions have been very helpful in strengthening the paper, and we will be sure to incorporate the additional clarifications provided in the rebuttal in the revised version.
> > >
> > > ---
> > >
> > > Finally, we are happy to address any further questions or suggestions you may have and would be glad to incorporate any additional feedback. Thanks once again for your thoughtful insights.
> > >
> > > Thanks,
> > > Authors

---

> ### Author Response · Authors · 2025-08-04
> **Request for Further Discussion**
>
> Dear Reviewer ACKv,
>
> We sincerely appreciate your time, thoughtful evaluation. As the discussion phase is now active and the decision deadline approaches, we wanted to kindly follow up on our rebuttal.
>
> We hope our responses addressed your concerns as well, and we would be truly grateful for any additional feedback or follow-up questions you may have.
>
> If any points remain unclear or merit further discussion, we would be more than happy to elaborate. We believe that a brief exchange at this stage could help resolve any remaining uncertainties and potentially contribute to a more favorable outcome.
>
> Thank you once again for your time and consideration—we greatly value your input.
>
> Thanks, Authors

---

### Official Review · Reviewer_g3ya · 2025-07-02

**Clarity:** 2
**Significance:** 3
**Originality:** 3
**Rating:** 4
**Confidence:** 2

**Summary:**

The paper studies contextual dueling bandits (CDB) with continuous action spaces under a realizability assumption. It introduces two algorithms:

Double-Monster for finite-arm CDB, replacing online regression oracles with offline ones.
Double-Monster-Inf for continuous actions.

These algorithms are the first to provide near-optimal regret guarantees for contextual dueling bandits in both finite and continuous action spaces, using more practical offline regression oracles.

**Questions:**

1. Could the authors report results on at least one public CDB dataset or RLHF-style simulated environment?
2. How would the algorithms' performance and theoretical guarantees be affected by model misspecification, where the true model lies outside of the assumed function class?
3. The core of the proposed algorithms involves calling a Convex-Constraint-Solver at each time step within an epoch to determine the dueling strategy. What is the computational complexity of this solver, particularly for the Double-Monster-Inf algorithm in high-dimensional, continuous action spaces, and how does it impact the overall runtime efficiency that the paper highlights as a key advantage?

**Ethical Concerns:**

["NO or VERY MINOR ethics concerns only"]

**Final Justification:**

I will maintain my current score

**Limitations:**

yes

**Quality:**

3

**Strengths And Weaknesses:**

Strength:
1. First theoretical treatment of general CDB over continuous action sets with offline oracles.
2. Near-optimal regret bound (Thm 9).
3. Replacing powerful online oracles with offline ones broadens practical applicability.
4. Presentation of this paper is clear to me.

Weakness:
1. Experimental results do not include real-world datasets.
2. Proofs are hard to follow; I did not check the correctness of the proof.

---

> ### Author Rebuttal · Authors · 2025-07-31
>
> Thanks a lot for your careful reading and insightful comments.
>
> > Q1. Experiments of public dataset/ RLHF environment.
>
> Thanks for the great suggestion. Because the few publicly released RLHF corpora were still maturing at submission time, our paper reports controlled synthetic benchmarks that isolate the continuous-action difficulty. Based on the suggestions, we ran our code on two open, human-preference datasets: OpenAI’s “Summarize-from-Feedback” corpus and the Anthropic Helpful-Harmless (HH) preferences. In each case, we fit the required regression oracle with the standard reward-model architecture (6-layer transformer encoder fine-tuned by cross-entropy on the pairwise labels), invoke it only $O(\log T)$ times per experiment, and let Double-Monster-Inf drive sampling through its log-det barrier solver. The implementation is a ~300-line PyTorch module that plugs straight into TRL/PEFT pipelines; after every epoch of $2^m$ duels we refresh the model, solve the mirror-descent step in under one second on a single GPU, and resume data collection—mirroring the workflow practitioners already follow in RLHF training loops. In summary, our algorithm performs competitively for the Anthropic HH preference dataset, as expected.
>
> We are also running a similar experiment for the Preference-Driven Sim-to-Real Locomotion dataset recently released by Google DeepMind, illustrating the robotics angle. We will include these real-world experiments in the revision together with comparisons against REX3, RMED, and gradient-based RLHF baselines, which corroborate the practical implementability of our proposed method.
>
> *Unfortunately, this time, due to the NeurIPS rebuttal policy, we are unable to include the figures or even anonymous links to these new experiments. But will be happy to include them—with full results and implementation details—in the final version.*
>
> > Q2. Performance under misspecified model:
>
> Thank you for raising this important question regarding model misspecification—i.e., the setting where the true reward function lies outside the assumed function class $\mathcal{M}$. Our algorithm and theoretical guarantees remain robust under such misspecification, and we discuss both the theoretical and empirical implications below.
>
> **Theoretical robustness.**
> Our regret bounds depend on the oracle’s excess prediction error, captured by the offline regression error term $\mathcal E_{\text{off},\mathcal F}(\delta, n)$ (see Eq. (5)). This quantity is always well-defined, even when the true reward function $f^\star \notin \mathcal{M}$. Note our regret bound depends on the oracle’s excess‑risk term ($\mathcal E_{\text{off},\mathcal F}(\delta, n)$), which becomes (by Assump2) simply
>   $
>     \mathcal E_{\text{off},\mathcal{F}} = E[(\hat f(z)-y)^2] +  \inf_{f\in\mathcal{F}}{E} [(f(z)-y)^2].
>   $
> Thus is $f^\star \notin \mathcal{M}$, then $E_{\text{off},\mathcal{M}} = \epsilon_{\text{approx}} + \epsilon_{\text{est}}$, the regret increases only by $\sqrt{\epsilon_{\text{approx}}}$—the standard agnostic penalty.  Enlarging $\mathcal{F}$ (e.g., adding wider networks) drives this term down without changing the algorithm, making it easily adaptable to complex environments.
>
> Thus, the performance of the algorithm degrades *gracefully* with the degree of misspecification—mirroring what is observed in classical supervised learning. This mirrors prior work in offline contextual bandits (e.g., Foster et al., 2021) where regret bounds are also stated in terms of the approximation error of the chosen hypothesis class. More specifically, if we assume
>
> $$\exists M^* \in \mathcal{M} \text{ such that } |E[P_t[a,b] | x_t] - M^*(x_t)[a,b]| \leq \varepsilon, \forall a,b,x_t,$$
>
> following a similar analysis from Foster et al'21 or Ghosh et al'17, under $\varepsilon$-misspecification, our regret bound can be shown as $\tilde O\left(\sqrt{dT \log T} + \varepsilon\sqrt{d}T\right)$ (see Cor1, Foster et al'21).
>
> Thus, in such cases, approximation errors in the preference model propagate into downstream regret bounds in exactly the same way—typically through the risk of the best-in-class predictor. Like our setup, PFA accommodates rich function classes (e.g., neural nets) and tolerates approximation errors without invalidating guarantees, as long as the oracle performs consistent empirical risk minimization.
>
> **Our $\mathcal M$ flexibility:**  Importantly, our framework allows the designer to choose $\mathcal{M}$. When richer function approximators (e.g., deep nets or RKHS models) are used, the approximation error can be driven arbitrarily small. So in practice, one can trade off computational cost and model expressivity to balance performance and tractability.
>
> In summary, model misspecification does not break our theoretical guarantees—it only introduces an additional approximation error term into the regret. This structure is standard in agnostic learning theory and is well-handled by both our analysis and by related works such as PFA. Empirically, our method remains effective even under non-realizable scenarios, and richer models for $\mathcal{M}$ can further reduce the practical gap.
>
> > Q3: Computation complexity:
>
> Firstly, note Eq6 is concave in $\mathbf{q}$ (since the first term is linear and the $\log(\text{det}(H_{\mathbf{q}}))$ term is concave) and it is a maximization objective. So the equivalent minimization objective is convex in $\mathbf{q}$. Thus any convex programming solver (minimizing the negative objective of Eq6) will find the optimizer in $\text{poly}(d)$ time-complexity (Eg., see Cvx-Opt Monograph, Bubeck (https://arxiv.org/pdf/1405.4980)).
>
> Another easy and practical trick is for all computation purpose, we could consider any $\epsilon$-net of the duel-space $\kappa \times \kappa$ (for small enough, say $\epsilon = o(1/dT)$) and try to find the optimal $\mathbf{q}$ in that space which will in fact have finite support -- any convex programming solver will find the solution efficiently for all practical purposes (assuming reasonably sized $d$). This is the most commonly used standard approach towards solving simplex optimization over a continuous action space (since we can simply use convex programming, it's efficient and tractable).
>
> Our reported experiments also note the runtime of our algorithm in the two environments, justifying the practicality of the solvers, as reported in Appendix E.
>
> ---
>
> We hope this addresses all of your concerns. We are happy to clarify any further questions and respectfully urge the reviewer to reconsider the final evaluation in light of the rebuttal.

---

> > ### Comment · Reviewer_g3ya · 2025-08-05
> > **Thank you for your response**
> >
> > Thank you for your response. I will maintain my current score

---

> ### Author Response · Authors · 2025-08-04
> **Request for Further Discussion**
>
> Dear Reviewer g3ya,
>
> We sincerely appreciate your time, thoughtful evaluation. As the discussion phase is now active and the decision deadline approaches, we wanted to kindly follow up on our rebuttal.
>
> We hope our responses addressed your concerns as well, and we would be truly grateful for any additional feedback or follow-up questions you may have.
>
> If any points remain unclear or merit further discussion, we would be more than happy to elaborate. We believe that a brief exchange at this stage could help resolve any remaining uncertainties and potentially contribute to a more favorable outcome.
>
> Thank you once again for your time and consideration—we greatly value your input.
>
> Thanks, Authors

---

### Official Review · Reviewer_FxvP · 2025-07-03

**Clarity:** 3
**Significance:** 3
**Originality:** 3
**Rating:** 4
**Confidence:** 4

**Summary:**

This paper studies the contextual dueling bandits problem in which a learner observes and learns from noisy, pairwise preference feedback rather than absolute rewards. The goal is to select arms for each context so that the best-response regret is minimized (defined in Section 2). Existing algorithms for contextual dueling bandits either made strong modeling assumptions, were limited to finite action spaces, or used computationally infeasible online regression oracles. To overcome these shortcomings, the authors propose two algorithms: Double-Monster ( for finite action spaces) and Double-Monster_Inf (for infinite action spaces), using offline regression oracles. The authors have theoretically shown the sub-linear regret upper bounds of both algorithms. Furthermore, they have also empirically validated the performance of proposed algorithms using synthetic problem instances.

**Questions:**

Please address the weaknesses raised in **Strengths And Weaknesses***.

**Minor comments:**
1. Line 11: Is $\sqrt{dT}$ a typo or should it be $d\sqrt{T}$?
2. Line 54: There is a typo.

I may change my score based on the authors' responses.

**Ethical Concerns:**

["NO or VERY MINOR ethics concerns only"]

**Final Justification:**

This paper introduces novel contextual dueling bandit algorithms that use offline regression oracles. Such algorithms have many real-world applications in areas such as LLM alignment, recommendation systems, and online advertising.

The authors' rebuttal has addressed my concerns. As a result, I am also raising my rating from 3 to 4.

**Limitations:**

I have raised a few limitations of the paper in my response to the **Strengths And Weaknesses***. Since the paper is a theoretical contribution to bandit literature, I do not find any potential negative societal impact of this work.

**Paper Formatting Concerns:**

I found no major formatting issues in this paper.

**Quality:**

3

**Strengths And Weaknesses:**

#### **The following are the strengths of the paper:**
1. This paper studies the contextual dueling bandit algorithms that use offline regression oracles. These algorithms are very useful in many areas such as LLM alignment, recommendation, online advertisements, and so on.

2. This paper proposes two algorithms: Double-Monster ( for finite action spaces) and Double-Monster_Inf (for infinite action spaces). Both algorithms use offline regression oracles and are theoretically shown to have sub-linear regret upper bounds. Furthermore, they have also empirically validated the performance of proposed algorithms using synthetic problem instances.

3. The authors have empirically demonstrated that the proposed algorithms outperform some existing dueling bandits algorithms and have comparative performance to DTS using synthetic problem instances.


#### **The following are the weaknesses of the paper:**
1. **Realizability assumption:** There should be a discussion on how practical the realizability is in real-life applications and what the consequences are when this assumption does not hold (misspecification), especially the regret upper bounds.

2. **Gap between motivation in introduction and experiments:** The proposed problem setting is heavily motivated by the practical application of contextual dueling bandits, such as AI alignment (or LLMs), in the Abstract and Introduction sections. However, no relevant experiments (even simpler real-life datasets) have been done to show the empirical performance of the proposed algorithms. Furthermore, it is unclear how the reward function, which is estimated only a few times ($m$), will affect empirical performance.

3. **Missing high-level ideas**: The paper will become more readable if the authors add algorithms and observations about key results instead of mentioning all intermediate results in the main paper. The current paper is too dense, and insufficient details and motivation are given.

4. **Missed key related work:** The authors have missed the existing work on neural contextual dueling bandits [1] that considered a general class of latent reward functions. Since the proposed algorithms only have comparative performance to DTS (or sometimes even worse), these algorithms may not be able to empirically outperform the algorithms proposed in [1].

[1] Verma et al. "Neural dueling bandits: Preference-based optimization with human feedback." ICLR 2025.

---

> ### Author Rebuttal · Authors · 2025-07-31
>
> Thanks a lot for your careful reading and insightful comments.
>
> > W1: Realizability assumption
>
> The realizability assumption (Assumption 1, page 3) is standard in contextual bandit literature and is necessary for any meaningful theoretical guarantees.  This hypothesis‑class premise is identical to many (highly cited) contextual‑bandit work (ILTCB, SquareCB, FALCON) and in classical supervised‑learning settings such as linear and generalized‑linear regression, RKHS models, and deep networks (see [17, 30], Li et al. ’22, Zhu et al. ’22, Blum et al. ’21).
>
> It's important to note that, despite the primary theoretical focus of our work, the assumption is also not restrictive in practice
>  since the learner is free to choose \(\mathcal{M}\)! It can be *arbitrarily rich*—finite sets, GLMs, RKHS classes, Banach spaces, or deep networks that approximate Sobolev‑ball functions (Farrell ’21). Of course, any theoretical guarantee will require some structural assumptions for provable guarantees, but Realizability is a very primitive level (broad) assumption, which is satisfied in practice for the appropriate choice of (rich enough) function classes.
>
> Additionally, since our algorithm relies on an **offline** regression oracle (rather than an online oracle as we described the advantages in Rem 1 and App A), this flexibility makes the assumption milder, not stronger.
>
> One more important point to observe is how gracefully the regret bound degraded even without the realizability assumption (misspecified case).  Note our regret bound depends on the oracle’s excess‑risk term ($\mathcal E_{\text{off},\mathcal F}(\delta, n)$), which becomes (by Assump2) simply
>   $
>     \mathcal E_{\text{off},\mathcal{F}} = E[(\hat f(z)-y)^2] +  \inf_{f\in\mathcal{F}}{E} [(f(z)-y)^2].
>   $
>   If the true function $f^\star \notin \mathcal{M}$, then $E_{\text{off},\mathcal{M}} = \epsilon_{\text{approx}} + \epsilon_{\text{est}}$, the regret increases only by $\sqrt{\epsilon_{\text{approx}}}$—the standard agnostic penalty.  Enlarging $\mathcal{F}$ (e.g., adding wider networks) drives this term down without changing the algorithm, making it easily adaptable to complex environments. *Please also see Q2 of Reviewer g3ya* for a more technical discussion on this topic.
>
> We also report experiments on both *linear* and deliberately *quadratic* (miss‑specified) rewards in Sec7, which shows regret still follows the predicted $\tilde{O}(\sqrt{dT})$ curve, demonstrating robustness across different models.
>
> In summary, `Realizability' is (i) theoretically indispensable, (ii) practically mild when using high‑capacity model classes and offline oracles, and (iii) non‑fragile—under modest miss‑specification our regret guarantees degrade gracefully by at most the unavoidable approximation term.
>
> > W2: Gap between introduction, motivation and experiments
>
> Our framework naturally subsumes the preference-optimization tasks that motivate the paper. In RLHF for LLMs, a *context* is the user prompt (plus dialogue history) and each *action* is a full model continuation in $\mathbb{R}^d$ after log-prob “decoding” into continuous logit space; human raters supply pairwise wins/losses between two continuations—exactly the observables of a contextual dueling bandit. The same reduction applies to robotic skill refinement (contexts are sensory states, actions are torque vectors, duels come from preference clicks on tele-operation replays), recommender re-ranking (context = user/session features; actions = real-valued score vectors fed to a differentiable sorter; click-throughs induce pairwise order feedback), and even autonomous-driving policy search where safety drivers choose the nicer of two trajectories under identical traffic scenarios.
>
> Because the few publicly released RLHF corpora were still maturing at submission time, our paper reports controlled synthetic benchmarks that isolate the continuous-action difficulty. Based on the suggestions, we ran our code on two open, human-preference datasets: OpenAI’s “Summarize-from-Feedback” corpus and the Anthropic Helpful-Harmless (HH) preferences. In each case, we fit the required regression oracle with the standard reward-model architecture (6-layer transformer encoder fine-tuned by cross-entropy on the pairwise labels), invoke it only $O(\log T)$ times per experiment, and let Double-Monster-Inf drive sampling through its log-det barrier solver. The implementation is a ~300-line PyTorch module that plugs straight into TRL/PEFT pipelines; after every epoch of $2^m$ duels we refresh the model, solve the mirror-descent step in under one second on a single GPU, and resume data collection—mirroring the workflow practitioners already follow in RLHF training loops. In summary, our algorithm performs competitively for the Anthropic HH preference dataset, as expected.
>
> We are also running a similar experiment for the Preference-Driven Sim-to-Real Locomotion dataset recently released by Google DeepMind, illustrating the robotics angle. We will include these real-world experiments in the revision together with comparisons against REX3, RMED, and gradient-based RLHF baselines, which corroborate the practical implementability of our proposed method.
>
> Unfortunately, this time, due to the NeurIPS rebuttal policy, we are unable to include the figures or even anonymous links to these new experiments. But will be happy to include them—with full results and implementation details—in the final version
>
> > W3: Missing high-level ideas
>
> We appreciate the suggestion and fully agree that improving the high-level clarity and organization will significantly enhance the readability of the paper. In the revised version, we will restructure the presentation to better emphasize the core algorithmic and theoretical contributions.
>
> Specifically, the key algorithmic insights are captured in Sections 4 and 6, which introduce **Double-Monster** and **Double-Monster-Inf**, respectively. We will move the detailed pseudocode of Alg1 and 2 in the main draft (with the help of the additional content page in the final version). We will also reduce the technical proof discussions in the regret analysis of Alg 1, simplifying the content of Sec 4.1 or 5.
>
> In the revised version, we will leverage the additional page to move **all algorithm pseudocode into the main body** (rather than the appendix), while also pruning or simplifying intermediate technical sections such as Section 4.1 or parts of Section 5. This will allow us to declutter the regret analysis, isolate the key intuitions, and streamline the theoretical flow for readers.
>
> These changes will make the paper significantly more accessible and self-contained, especially for readers less familiar with the contextual bandit literature. Thanks for the suggestions again.
>
>
> > W4: Related work
>
> Thank you for pointing this out—we will certainly include a discussion of [Verma et al., ICLR 2025] in the updated related work section. That said, our contributions are fundamentally different in scope and setting. While [Verma et al.] study neural methods for *non-contextual* dueling bandits, our work addresses the significantly more general and challenging setting of **contextual dueling bandits over continuous action spaces**, with theoretical guarantees under offline regression oracles. Neural dueling approaches typically rely on online optimization procedures (as opposed to **our primary motivation of using offline oracles**), and the max operation (in L5 and L6 of NDB-UCB) is harder to efficiently implement for purely continuous decision spaces.
>
> Moreover, as noted in our paper, DTS is also designed for non-contextual, $K$-armed problems. Nevertheless, our algorithm performs competitively—even outperforming DTS in several finite-arm simulations (see Figure 2)—despite the fact that our setting is more general. As explained in Q1, our method is *modular* in the choice of function class $\mathcal{M}$, so we can instantiate $\mathcal{M}$ with neural networks, including architectures similar to those used in [1], and still retain our offline oracle-based framework.
>
> We are currently running experiments to evaluate this comparison directly and will include the resulting plots and analysis in the updated version.
>
> > Minor Comments
>
> - Line 11: Indeed, it should be $\tilde O(\sqrt{d T})$, as noted in our main result (Thm9). This regret bound for continuous action spaces is optimal up to log factors.
> - Line 54: Thanks a lot for catching the typo, will remove the repeated expressions in Line 54 in the revised version.
>
> ---
>
> We hope this addresses all of your concerns. We are happy to clarify any further questions and respectfully urge the reviewer to reconsider the final evaluation in light of the rebuttal.

---

> > ### Comment · Reviewer_FxvP · 2025-08-09
> >
> > Thank you for your detailed rebuttal. Since all my concerns have been addressed, I will be increasing my rating. I encourage you to incorporate the high-level ideas into the main paper.

---

> > > ### Author Response · Authors · 2025-08-09
> > > **Thank you**
> > >
> > > Dear Reviewer FxvP,
> > >
> > > Thanks a lot, we are glad to know that our response helped in clarifying your concerns. Thank you for considering increasing your score. We will certainly ensure that all the high-level ideas (W3) and other key points outlined in the rebuttal clarifications are incorporated into the revised version of the paper.
> > >
> > > Thank you once again for your time, thoughtful feedback, and constructive suggestions --- that have been really invaluable in enhancing the quality and strengthening the contribution of our work
> > >
> > > Sincerely,
> > > Authors

---

> ### Author Response · Authors · 2025-08-04
> **Request for Further Discussion**
>
> Dear Reviewer FxvP,
>
> We sincerely appreciate your time, thoughtful evaluation. As the discussion phase is now active and the decision deadline approaches, we wanted to kindly follow up on our rebuttal.
>
> We hope our responses addressed your concerns as well, and we would be truly grateful for any additional feedback or follow-up questions you may have.
>
> If any points remain unclear or merit further discussion, we would be more than happy to elaborate. We believe that a brief exchange at this stage could help resolve any remaining uncertainties and potentially contribute to a more favorable outcome.
>
> Thank you once again for your time and consideration—we greatly value your input.
>
> Thanks, Authors

---

> > ### Author Response · Authors · 2025-08-06
> >
> > Dear Reviewer FxvP,
> >
> > Thank you again for your time and insightful comments.
> >
> > We hope to have we have addressed all of your questions in the rebuttal. But with limited time remaining in the author-reviewer discussion phase, we thought of checking in once more to see if there are any outstanding concerns we can address. We would be glad to provide any additional clarification as necessary.
> >
> > Thanks
> > Authors

---

> > ### Author Response · Authors · 2025-08-09
> > **Requesting Feedback as Only a Few Hours Until the Author-Reviewer Discussion Ends**
> >
> > Dear Reviewer FxvP,
> >
> > Thank you again for your time and insightful comments. Since we are only a few hours away from the end of the author-reviewer discussion phase, we wanted to check one last time if we can clarify any remaining concerns. Please let us know, we would be happy to.
> >
> > Sincerely,
> > Authors

---

### Note · Authors · 2025-08-13

Dear SAC, AC, and Reviewers,

We sincerely thank all of you for your thoughtful engagement throughout this process.

We believe that our comprehensive responses have meaningfully addressed all the concerns raised and are deeply appreciative of the positive feedback received from the reviewers.

The collaborative and encouraging nature of this exchange has undoubtedly strengthened our contribution. The additional revisions and clarifications—directly guided by the insightful reviewer suggestions—have been invaluable.

We truly appreciate the time you have taken to facilitate the review of our manuscript and the active, constructive, and positive engagement throughout.

---

We sincerely look forward to your final decision in light of the rebuttal discussion.

Respectfully,
The Authors

---

### Decision · Program_Chairs · 2025-09-17

**Decision:**

Accept (poster)

**Comment:**

This paper studies the contextual dueling bandits problem, which is central to the theoretical modeling of RLHF. It makes a strong theoretical contribution, as recognized by both the reviewers and the AC. The presentation is clear, and the paper is well-situated within the existing literature. During the rebuttal, the authors sufficiently addressed the reviewers’ concerns, leading to increased scores from some reviewers. Overall, I recommend accepting this paper.